# Aberrant NSUN1 activity connects m$^5$C-RNA modification to TDP-43 neurotoxicity in ALS/FTD

Melissa Parra-Torres[1] , Kumara Dissanayake[1], James A Gray[1] , Alistair J Langlands[2] , Ridvan Kucuk[1], Marek Gierlinski[3] , Claire Troakes[4,5], Andrew King[6] , Leeanne McGurk[1]

In amyotrophic lateral sclerosis (ALS) and frontotemporal dementia (FTD), the nuclear RNA-binding protein TDP-43 mislocalises to the cytoplasm and forms insoluble aggregates, but the mechanisms controlling this remain unclear. We define a native TDP-43 interactome in human SH-SY5Y cells and identify proteins linked to the 5-methylcytosine (m$^5$C) RNA modification as highly enriched. Using a Drosophila model of TDP-43 pathology, we show that aberrant activity of m$^5$C-RNA methyltransferases Nsun1 drives TDP-43–induced m$^5$C-RNA hypermethylation, whereas Nsun1 down-regulation alleviates TDP-43–induced degeneration, lifespan deficits, and cytoplasmic accumulation. In human cells, TDP-43 selectively interacts with NSUN1 isoform 3 independently of RNA. Furthermore, NSUN1 is nucleolar and TDP-43 is largely nucleoplasmic, yet they interact in both compartments, suggesting functional roles beyond their predominant localisations. In ALS/FTD postmortem frontal cortex, NSUN1 isoform 3 persists, whereas the shorter isoform is reduced, suggesting that a pool of NSUN1 capable of contributing to pathological TDP-43 interactions remains in disease. These findings suggest that TDP-43 neurotoxicity is coupled to NSUN1 activation and m$^5$C-RNA methylation, revealing a potential therapeutic axis in ALS/FTD.

## Introduction

The transactive response (TAR) DNA-binding protein of 43 kD (TDP-43) is a predominantly nuclear protein that regulates RNA transcription, splicing, transport, and stability by binding to thousands of target RNAs (Polymenidou et al, 2011; Sephton et al, 2011; Tollervey et al, 2011; Lagier-Tourenne et al, 2012; Colombrita et al, 2015; Ling et al, 2015; Rot et al, 2017; Tank et al, 2018; Donde et al, 2019). It preferentially binds single-stranded RNA at UG-dinucleotide motifs, with enrichment in intronic regions where it suppresses cryptic exon inclusion (Buratti & Baralle, 2001; Tollervey et al, 2011; Ling et al, 2015; Tan et al, 2016; Klim et al, 2019; Melamed et al, 2019; Brown et al, 2022; Ma et al, 2022). Loss of TDP-43 is embryonically lethal in mice, underscoring its essential function (Kraemer et al, 2010). In nearly all cases of amyotrophic lateral sclerosis (ALS), TDP-43 mislocalises to the cytoplasm and forms insoluble aggregates in neurons and glia; similar pathology occurs in ~45% of frontotemporal dementia (FTD), limbic-predominant age-related TDP-43 encephalopathy (LATE) and in Alzheimer's disease co-morbid with LATE-NC (LATE-neuropathological changes) (Arai et al, 2006; Neumann et al, 2006; Mackenzie et al, 2007; Josephs et al, 2014; Nelson et al, 2019). Aggregation is accompanied by nuclear depletion and loss of TDP-43 function, leading to widespread RNA processing defects such as aberrant cryptic exon inclusion. These alterations disrupt reading frames and protein expression, driving neurotoxicity in disease (Ling et al, 2015; Klim et al, 2019; Brown et al, 2022; Ma et al, 2022). Despite its central role, the mechanisms underlying TDP-43 nuclear depletion in ALS and related disorders remain unclear.

Under normal conditions, ~10% of TDP-43 continuously shuttles between the nucleus and cytoplasm, maintained by a balance of importin $\alpha/\beta$–mediated nuclear import and passive export (Ayala et al, 2008; Winton et al, 2008; Nishimura et al, 2010; Bentmann et al, 2012; Ederle et al, 2018; Duan et al, 2022). In the cytoplasm, TDP-43 contributes to mRNA transport, microRNA biogenesis, and protein translation (Buratti et al, 2010; Fallini et al, 2012; Russo et al, 2017). Multiple mechanisms can disrupt this nuclear–cytoplasmic equilibrium. Examples include impaired nuclear import/export pathways, defects in the nuclear pore complex, or alterations to the TDP-43 nuclear localisation signal (NLS) through cleavage or post-translational modification (Igaz et al, 2009; Nishimura et al, 2010; Bentmann et al, 2012; Hutten et al, 2020; Coyne et al, 2021; Garcia Morato et al, 2022). Cellular stressors, including ER and oxidative

[1]Division of Molecular, Cell and Developmental Biology, School of Life Sciences, University of Dundee, Dundee, UK   [2]National Phenotypic Screening Centre, School of Life Sciences, University of Dundee, Dundee, UK   [3]Data Analysis Group, Division of Computational Biology, School of Life Sciences, University of Dundee, Dundee, UK   [4]Department of Basic and Clinical Neuroscience, London Neurodegenerative Diseases Brain Bank, Institute of Psychiatry, Psychology and Neuroscience, King's College London, London, UK   [5]London Neurodegenerative Diseases Brain Bank, SGDP Centre, PO65, Institute of Psychiatry, Psychology and Neuroscience, King's College London, London, UK   [6]King's College Hospital NHS Foundation Trust, Academic Neuroscience Centre, London, UK

Correspondence: LMcgurk001@dundee.ac.uk

stress, further drive TDP-43 mislocalisation and promote aggregate formation (Molliex et al, 2015; McGurk et al, 2018a; Gasset-Rosa et al, 2019; Mann et al, 2019; Hans et al, 2020; Streit et al, 2022). RNA also acts as a critical regulator: transcription enhances nuclear retention, whereas post-transcriptional N6-methyladenosine ($m^6A$) methylation facilitates cytoplasmic accumulation (Ayala et al, 2008; Ederle et al, 2018; Duan et al, 2022; McMillan et al, 2023; Sun et al, 2023; Nguyen et al, 2025). RNA modifications such as $m^6A$ methylation are emerging as key players in neurodegeneration, but how they intersect with stress pathways to promote TDP-43 pathology remains unknown.

There are over 100 types of RNA modifications, deposited by writer proteins, interpreted by reader proteins, and removed by erasers (Zaccara et al, 2019; Delaunay et al, 2024). Among the most abundant are $m^6A$ and N1-methyladenosine ($m^1A$), both of which enhance TDP-43 binding to RNA, with $m^1A$ additionally driving cytoplasmic accumulation (Wei et al, 1976; Wei & Moss, 1977; Dominissini et al, 2016; Li et al, 2016; McMillan et al, 2023; Sun et al, 2023; Nguyen et al, 2025). These findings highlight RNA modifications as emerging regulators of TDP-43 localisation and function, yet whether other modifications—such as 5-methylcytosine ($m^5C$)—contribute to pathology remains unclear. In this study, we generated a native TDP-43 interactome from human neuronal-like cells (SH-SY5Y) and found that proteins binding to RNA modifications, including $m^6A$ readers, are highly enriched. Notably, proteins that recognize $m^5C$ in RNA were the top enriched molecular function. Functional screening in *Drosophila melanogaster* (Drosophila) revealed that TDP-43 up-regulation increases RNA $m^5C$ levels via the writer Nsun1, and that Nsun1 is required for TDP-43–induced neurodegeneration. In human cells, TDP-43 selectively interacts with the longer NSUN1 isoform (isoform 3), forming a protein complex in both the nucleolus and nucleoplasm. Analysis of ALS/FTD postmortem cortex revealed that although the shorter NSUN1 isoform is significantly reduced, isoform 3 persists, suggesting that the TDP-43–interacting isoform remains available in disease and may enable ongoing pathological interactions. Together, these findings uncover a new mechanism linking TDP-43 to $m^5C$-RNA methylation and identify NSUN1 as a critical mediator of TDP-43–driven neurotoxicity.

# Results

## 5-methylcytosine in RNA is an enriched pathway that interacts with TDP-43

Previously, we and others have used mass spectrometry to identify protein interactors that were important for TDP-43 function and toxicity (Freibaum et al, 2010; Kawaguchi et al, 2020; François-Moutal et al, 2022). However, these studies involved the use of up-regulated and epitope-tagged TDP-43; thus, we set out to gain an understanding of the native TDP-43 interactome. Under non-denaturing conditions, TDP-43, versus an IgG control, was immunoprecipitated from SH-SY5Y cells (Fig 1A), which are human cells of a neuronal origin. Co-immunoprecipitating proteins were identified by liquid chromatography with tandem mass spectrometry (LC-MS/MS). Of the 1,053 identified proteins, only 26 were

enriched in the IgG control, indicating that our TDP-43 co-immunoprecipitation was highly selective for TDP-43 (Table S1). Of the 1,027 proteins that selectively co-immunoprecipitated with TDP-43, 782 co-immunoprecipitated with TDP-43 in all three repeats—these proteins were classified as high-confidence interactors (Table S1). STRING interaction analysis was used to generate a high-confidence interaction network (interaction score > 0.9), which revealed four main clusters enriched for proteins involved in ribosome biogenesis, RNA splicing/transport/degradation, DNA replication and repair, and protein transport (Figs 1B and S1, S2, S3, S4, and S5). These data are consistent with known TDP-43 functions and a previous TDP-43–GFP interactome study that revealed enrichment for RNA biogenesis mechanisms (Freibaum et al, 2010). As expected, TDP-43 interacted with nuclear, cytoplasmic, and nucleolar proteins (Fig 1D). Gene ontology (GO) analysis of molecular function revealed that proteins that bind to methylated cytosine (5-methylcytosine/$m^5C$) in RNA were the top enriched pathway (Fig 1C and Table S1). Reader proteins of the $m^6A$-RNA modification were also amongst the top enriched pathways (Fig 1C). Collectively, this TDP-43 native interactome suggests that RNA modifications broadly function with TDP-43, with cytosine methylation in RNA emerging as a novel pathway linked to TDP-43 biology.

## The 5-methylcytosine-RNA methyltransferases NSUN1 regulates TDP-43–associated neurotoxicity in Drosophila

Cytosines in RNA are methylated by the s-adenosylmethionine (sam)–dependent $m^5C$-RNA methyltransferases (Fig 2A), which include the seven NOL1/NOP2/SUN domain (NSUN) enzymes (NSUN1-7) and the DNA methyltransferase DNMt2a. Each $m^5C$-RNA methyltransferase methylates specific types of RNAs, for example, NSUN2 methylates mRNA, NSUN2/3/6 methylates tRNA, NSUN1/4/5 methylates rRNA, and DNMt2 methylates tRNA (Bohnsack et al, 2019). Thus, we set out to define which of the $m^5C$-RNA methyltransferase(s) functionally interact with TDP-43, hypothesizing that this would reveal the specific RNA methylation pathway and RNA type engaged by TDP-43 through its association with modified RNAs. To address this, we first identified the *D. melanogaster* (Drosophila) homologues of the human $m^5C$-RNA methyltransferases using BLAST analysis and then performed a genetic modifier screen in a Drosophila model that recapitulates TDP-43–associated toxicity (Elden et al, 2010). This analysis revealed that Drosophila has six of the eight $m^5C$-RNA methyltransferases—*Nsun1* (CG8545), *Nsun2* (CG6133), *Nsun4* (CG4749), *Nsun5* (CG42358), *Nsun6* (CG11109), and *Mt2* (CG10692) (Fig 2B and C), indicating that this enzyme family is conserved in Drosophila. We obtained Drosophila siRNA strains for *Nsun1*, *Nsun2*, *Nsun5*, *Nsun6*, and *Mt2* and a loss of function strain for *Nsun4*. To confirm that the silencing strategy for each gene was effective, the siRNAs were expressed ubiquitously with either *daughterless-geneswitch* (*DaGS*)-*GAL4* (for *Nsun1*) or *Daughterless* (*Da*)-*GAL4* (for *Nsun2*, *Nsun4*, *Nsun5*, *Nsun6*, and *Mt2*). This analysis showed that all strains to the $m^5C$-RNA methyltransferases significantly reduced the expression of their respective $m^5C$-RNA–methyltransferase targets (Fig 2D and E). The *Nsun1* gene is in the first intron of *Dgt5* and both genes are expressed from opposing strands

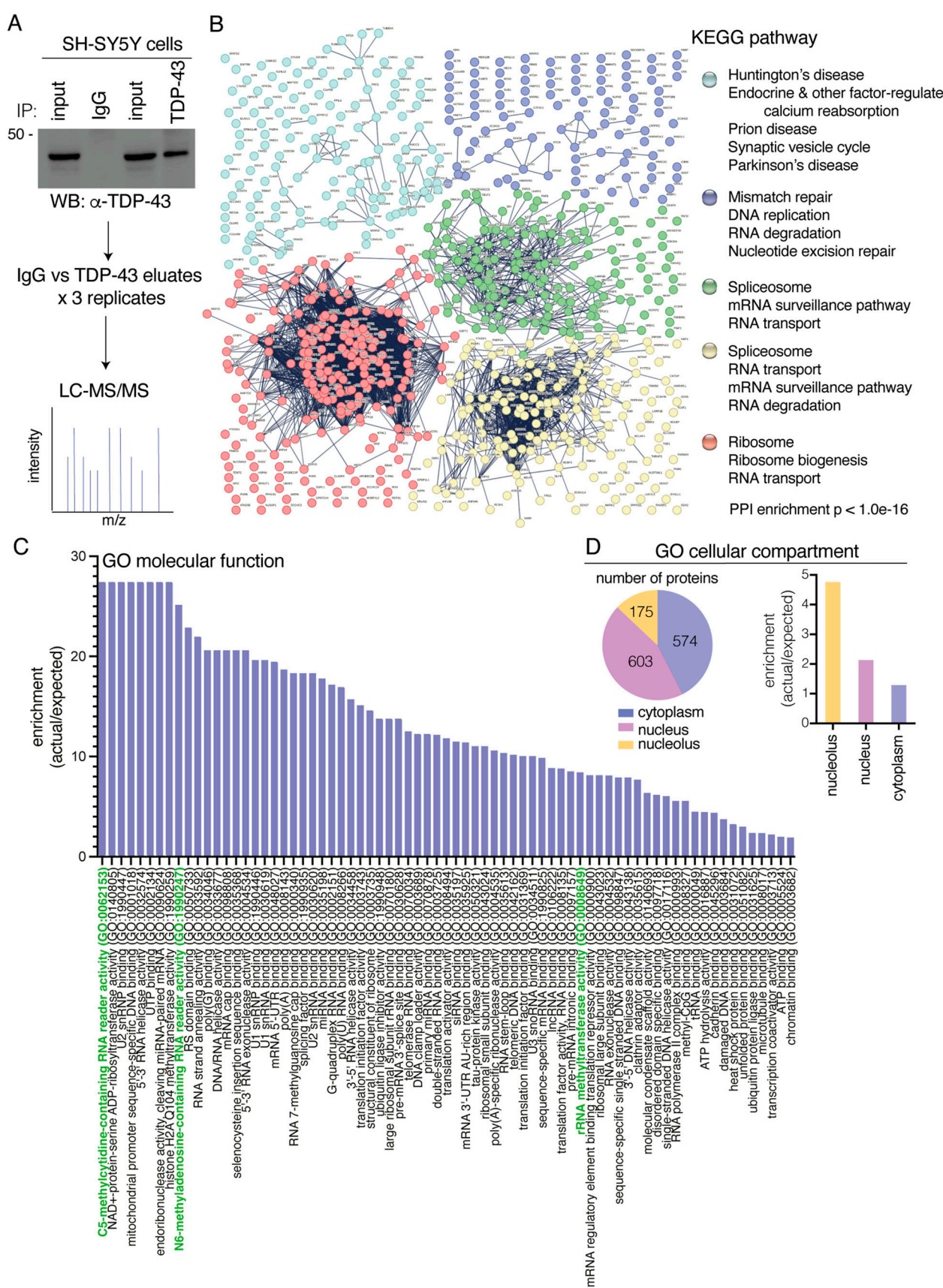

(Fig S6A). To exclude potential off-target effects of the siRNA to *Nsun1* on *Dgt5* expression, we measured *Dgt5* mRNA levels upon *Nsun1* silencing (Fig S6A and B). This confirmed that the siRNA was specific to *Nsun1* and did not alter *Dgt5* expression (Fig S6A and B).

To identify which of the conserved m5C-RNA methyltransferases function with TDP-43, we used a Drosophila strain that recapitulates TDP-43–induced toxicity (Elden et al, 2010). The selective expression of TDP-43 in the eye using the *glass multiple repeat-GAL4* (*gmr-GAL4*) driver causes degeneration of the external eye and internal retina (Fig 3A). In combination with the strains that down-regulate the m5C-RNA methyltransferases, we observed that TDP-43–induced degeneration of the external eye was significantly reduced by down-regulation of *Nsun1*, *Nsun2*, *Nsun5*, and *Nsun6* (Fig 3A and B), but only *Nsun1* down-regulation mitigated TDP-43–induced degeneration of the internal eye (Fig 3A and C). Extending this to a disease-causing mutant, TDP-43-Q331K (Elden et al, 2010), *Nsun1* knockdown significantly suppressed eye degeneration (Fig S6C–E), indicating that the protective effect of *Nsun1* loss persists in the context of pathogenic TDP-43. Importantly, *Nsun1* knockdown did not alter TDP-43 protein levels or those of a control protein (β-galactosidase) (Fig S6F–I), demonstrating that the rescue of toxicity is not because of reduced TDP-43 abundance or GAL4 protein levels (β-galactosidase). Real-time PCR further showed that *Nsun1* mRNA levels were unchanged by TDP-43 expression in the eye, suggesting that suppression of toxicity is not linked to restoration of *Nsun1* expression. Notably, expression of *Nsun1* was significantly reduced by siRNA-mediated knockdown in the eye (Fig 3D), although measuring transcript levels in whole heads can underestimate the extent of eye-specific silencing because of signal dilution from other tissues (e.g., brain and exoskeleton). Collectively, these findings identify *Nsun1* as the key modifier of TDP-43 toxicity in vivo, consistent with our native TDP-43 interactome, which revealed interactions with NSUN1 and NSUN5 (Table S1).

We next addressed whether *Nsun1* down-regulation was generally neuroprotective or whether it was selective for TDP-43 toxicity. To test this, *Nsun1* expression was down-regulated in three Drosophila strains that recapitulate spinocerebellar ataxia type 1 (SCA1), spinocerebellar ataxia type 3 (SCA3) and ALS/FTD characterised by the $(G_4C_2)$-hexanucleotide repeat expansion in C9orf72 (UAS-CAG$_{78}$; UAS-*ATXN1*-CAG$_{82}$; and UAS-[$G_4C_2$]$_{48}$, respectively). Unlike the protective effect on TDP-43 toxicity, *Nsun1* down-regulation exacerbated degeneration in UAS-*ATXN1*-CAG$_{82}$ and UAS-($G_4C_2$)$_{48}$, when having little to no impact on eye degeneration caused by UAS-CAG$_{78}$ (Fig S6J). *Nsun1* down-regulation on its own had no effect on the Drosophila eye compared with the normal control (Fig S6J), indicating that the enhanced degeneration in UAS-*ATXN1*-CAG$_{82}$ and UAS-($G_4C_2$)$_{48}$ by *Nsun1* down-regulation is

not simply additive. Together, these data demonstrate that the protective effect of *Nsun1* down-regulation is selective for TDP-43 toxicity in Drosophila.

## *Nsun1* is essential and, when down-regulated, mitigates TDP-43–associated neuronal toxicity

Expression of TDP-43 in the Drosophila in all neurons of the central nervous system with the *elav-GAL4* driver recapitulates aspects of human disease including age-dependent cytoplasmic accumulation of TDP-43 and shortened lifespan (Elden et al, 2010; Kim et al, 2014). To assess the effect of *Nsun1* down-regulation on TDP-43–associated neuronal toxicity, we silenced *Nsun1* in all neurons using *elav-GAL4*, and this resulted in developmental lethality (Fig S6K). Similar developmental dependence was observed when *Nsun1* was silenced in glia, motor neurons, and/or muscle, indicating that *Nsun1* is broadly essential during development (Fig S6K). To bypass this developmental requirement, we used the conditional *elav-geneswitch-GAL4* (*elavGS-GAL4*) strain to express TDP-43 specifically in adult neurons, either together with *Nsun1* siRNA (si.*Nsun1*) or a control siRNA (si.mCherry). Unlike development, *Nsun1* down-regulation in adult neurons had no significant effect on lifespan compared with controls (Fig 4A). Expression of human TDP-43 in adult neurons induced early death, which was significantly mitigated by *Nsun1* down-regulation (Fig 4A). Consistent with our data in the Drosophila eye (Fig S6F and G), *Nsun1* silencing had no effect on total TDP-43 protein levels in adult neurons (Fig 4B and C). These data indicate that targeting *Nsun1* in adults selectively suppresses TDP-43–mediated neuronal toxicity without affecting normal neuronal function.

To assess how *Nsun1* down-regulation affects TDP-43 accumulation in adult neurons, we measured TDP-43 protein levels in nuclear, cytosolic, and insoluble fractions from Drosophila head tissue. As expected, expression of TDP-43 with *elavGS-GAL4* resulted in detectable protein in all three compartments (Fig 4D and E). Concomitant *Nsun1* silencing, however, increased nuclear TDP-43 levels (0.2 ± 0.08 [SD] versus 0.6 ± 0.1 [SD], control versus si.*nsun1*, respectively), decreased cytoplasmic TDP-43 (0.82 ± 0.1 [SD] versus 0.45 ± 0.03 [SD], control versus si.*Nsun1*, respectively), and had no effect on the relative levels of insoluble TDP-43 (0.60 ± 0.57 [SD] versus 0.53 ± 0.19 [SD], control versus si.*Nsun1*, respectively) (Fig 4D and E). For the insoluble fraction, Lamin C was used as a loading control because it was present in this fraction; however, we cannot rule out that this may not fully reflect true loading as TDP-43 may impact Lamin C, so the insoluble fraction data should be interpreted with caution. As *Nsun1* silencing had no effect on total TDP-43 protein levels in adult neurons (Fig 4B and C), the observed redistribution is independent of changes in overall

**Figure 1. Native TDP-43 interactome in SH-SY5Y cells shows enrichment of 5-methylcytidine-containing RNA reader proteins as the top enriched molecular function.**
**(A)** The proteins that co-immunoprecipitated with either endogenous TDP-43 or the IgG control were subjected to LC-MS/MS and MaxQuant analysis. The experiment was performed in triplicate. **(B)** A total of 1,027 proteins selectively co-immunoprecipitated with TDP-43. Only proteins that co-immunoprecipitated with TDP-43 and not IgG in all three repeats were included in the analysis (783 proteins). Depicted is a full string network built upon experimental and text mining data. Only interactions with confidence score of >0.9 are shown and with a maximum of five interactors in the first shell. The network was clustered using a kmeans clustering of five. The KEGG pathways associated with each cluster are listed. Each cluster is presented in supplemental as larger figures (Figs S1, S2, S3, S4, and S5). **(C)** The gene ontology (GO) analysis shows the enriched molecular functions. With pathways linked to RNA methylation highlighted in green. **(D)** TDP-43 interacts with proteins that have functions in the nucleus, cytoplasm, and nucleolus.
Source data are available for this figure.

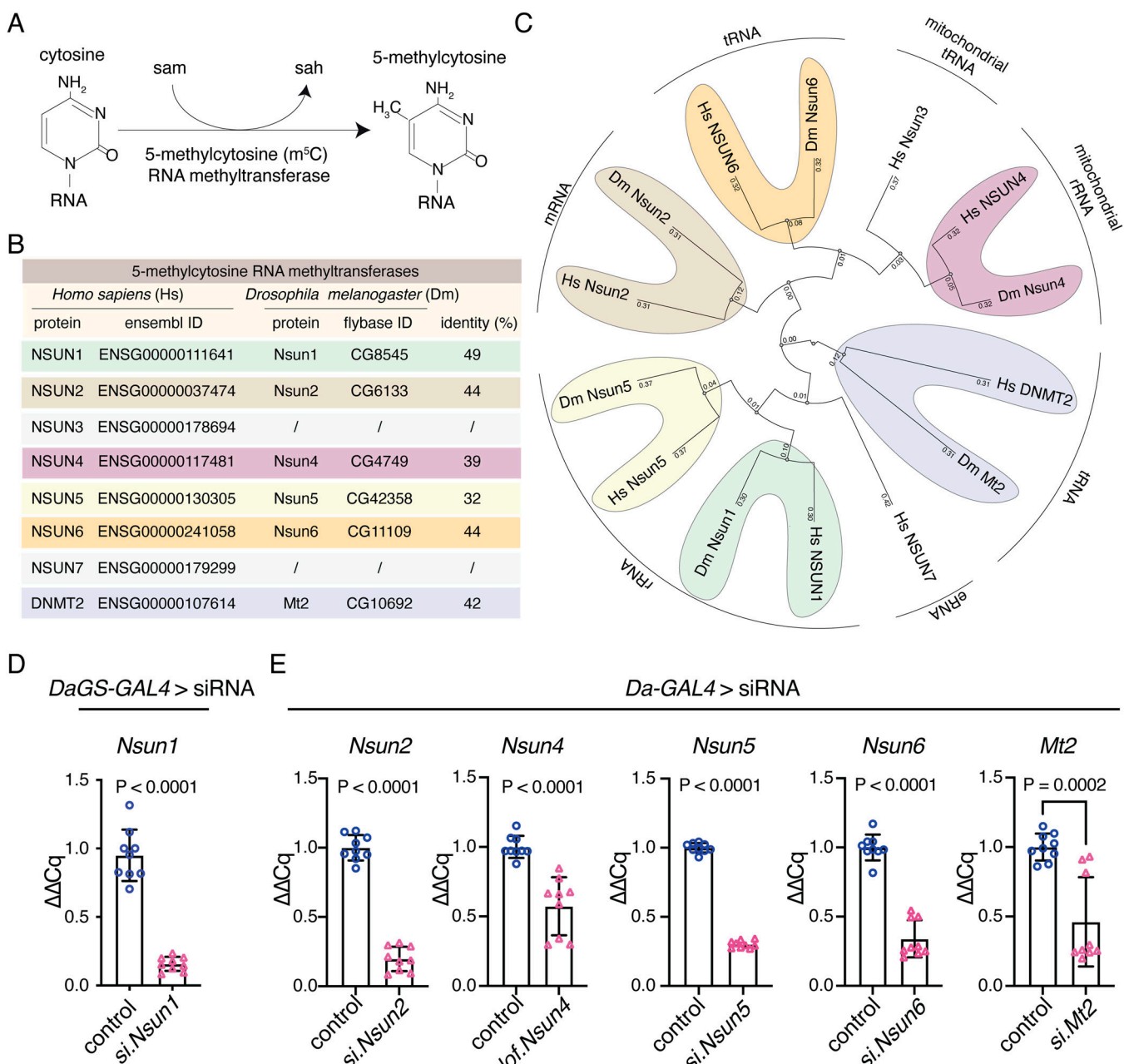

**Figure 2. The 5-methylcytosine RNA methyltransferases are conserved in Drosophila and are effectively silenced in the animal using genetic approaches.**
**(A)** The 5-methylcytosine (m⁵C) RNA methyltransferases add a methyl group at the carbon 5 position of cytosine using S-adenosyl methionine (sam) as a methyl donor producing S-adenosylhomocysteine (sah) as a byproduct. **(B)** The m⁵C-RNA methyltransferase proteins in humans and their closest homologue in *Drosophila melanogaster* (Drosophila) were identified using the NCBI BLAST online alignment tool (https://blast.ncbi.nlm.nih.gov/Blast.cgi). **(C)** Phylogenetic tree of the m⁵C-RNA methyltransferases in humans and Drosophila. The dendogram was generated using clustal omega (https://www.ebi.ac.uk/Tools/msa/clustalo/) and reconstructed using PRESTO (http://www.atgc-montpellier.fr/presto). The main RNA subtype for each enzyme is indicated. **(D)** *Nsun1* down-regulation by siRNA (si.*Nsun1*) in adult head and thorax tissue with *Daughterless-geneswitch (GS)-GAL4* (*DaGS-GAL4*) led to a significant reduction in *Nsun1* mRNA versus the control (si.mCherry). All data are relative to Tubulin. The graph is the mean (±SD) of three biological replicates; each data point is a technical replicate, three from each biological repeat, and a *t* test. **(E)** Expression of siRNAs to *Nsun2*, *Nsun5*, *Nsun6*, and by 50% loss of function of *Nsun4* causes a significant reduction in the respective genes versus the control (si.mCherry). Data are relative to Tubulin. The graph is the mean (±SD) of three biological replicates, each data point is a technical replicate, three from each biological repeat, and a *t* test, Genotypes: **(D)** Control: *y, sc, v, sev/w^1118; Daughterless-GAL4-geneswitch/+; si.mCherry^35785/+*, and si.*Nsun1: y, sc, v, sev/w^1118; Daughterless-GAL4-geneswitch/si.Nsun1 ^TRiP.HMC04440; +/+.* **(E)** Control: *w^1118; +/+; Daughterless-GAL4/si.mCherry^35785*, si.*Nsun2*: si.*Nsun2 ^TRiP.HMJ24019/+; Daughterless-GAL4/+, Nsun4: w^1118; cn[1] l(2)10685 [10685]/+; Daughterless-GAL4/+, Nsun5: w^1118; +/+; Daughterless-GAL4/si.Nsun5^TRiP.HMS00438, Nsun6: w^1118; +/+; Daughterless-GAL4/si.Nsun6^TRiP.HMC04118, Mt2: w^1118; si.mt2^TRiP.HMS0166/+; Daughterless-GAL4/+.*

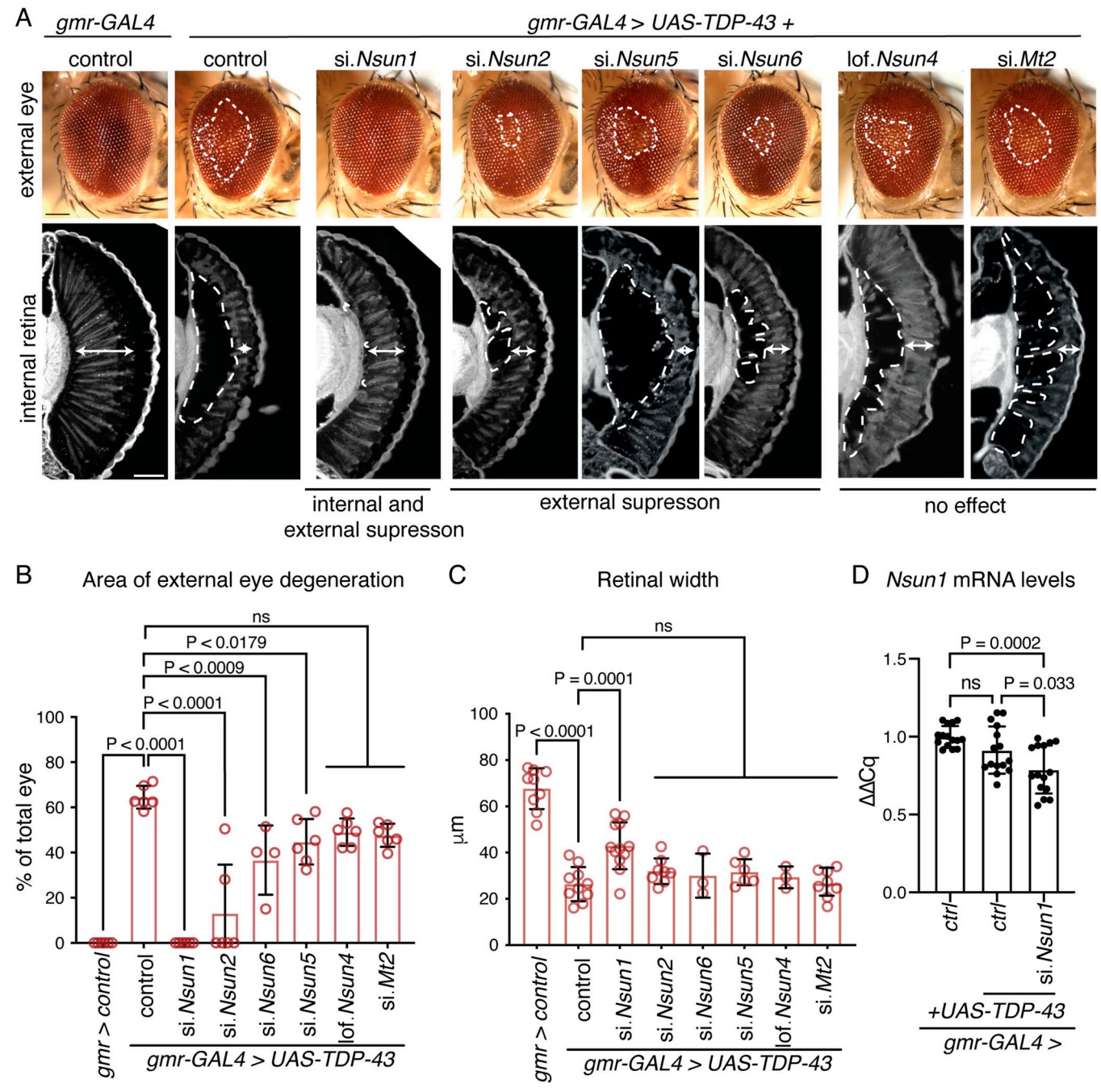

**Figure 3. Down-regulation of the 5-methylcytosine RNA methyltransferase *Nsun1* mitigates TDP-43 neurotoxicity in the Drosophila eye.**
**(A)** Expression of human TDP-43 leads to degeneration of the external eye (area in white hatched line, upper panel) and reduced retinal width (white arrow, lower panel) compared with the normal control. Reduction in *Nsun1*, *Nsun2* or *Nsun6* reduced TDP-43–induced degeneration of the external eye (upper panel), whereas only reduction in *Nsun1* mitigates TDP-43–induced degeneration of the internal retina (white arrow lower panel). Scale bars: upper panel is 25 μm, lower panel is 35 μm. **(B)** Reduction in *Nsun1*, *Nsun2*, *Nsun5*, and *Nsun6* by siRNA significantly reduces TDP-43–induced degeneration in the external eye. Each datapoint represents one animal and spans three independent biological repeats. Graph is the mean ± SD, one-way ANOVA ($P < 0.0001$), and a Tukey's test; ns, not significant. **(C)** Reduction *Nsun1* significantly mitigates TDP-43–induced reduction in retinal depth. Each datapoint represents one animal from three independent repeats. Graph is the mean ± SD, one-way ANOVA ($P < 0.0001$), and a Tukey's test; ns, not significant. **(D)** *Nsun1* expression levels are unalerted by TDP-43 expression. Expression of the siRNA to *Nsun1* is reduced in the eye. TDP-43 was expressed in the Drosophila eye with *gmr-GAL4* with either a control (si.mCherry) or a siRNA to *Nsun1* (si.*Nsun1*) and both were compared with the normal control (si.mCherry alone). All data are relative to Tubulin. RNA was isolated from the entire head; thus, not all cells will express the siRNA to *Nsun1*, and therefore, not all cells will have reduced *Nsun1* levels. Graph is the mean of five biological repeats, one-way ANOVA, and a Tukey's test; ns, not significant. Each data point is a technical replicate, three from each biological repeat. Genotypes: gmr-GAL4 control: +/+; gmr-GAL4 (YH3)/si.mCherry[35785], gmr-GAL4 + TDP-43 + control: TDP-43 (37M)/+; gmr-GAL4 (YH3)/si.mCherry[35785], gmr-GAL4 + TDP-43 + si.*Nsun1*: TDP-43 (37M)/si.*Nsun1*[TRiP.HMC04440]; gmr-GAL4 (YH3)/+, gmr-GAL4 + TDP-43 + si.*Nsun2*: TDP-43 (37M)/si.*Nsun2*[TRiP.HMJ24019]; gmr-GAL4 (YH3)/+, gmr-GAL4 + TDP-43 + lof.*Nsun4*: TDP-43 (37M)/cn[1] l(2)10685[10685]; gmr-GAL4 (YH3)/+, gmr-GAL4 + TDP-43 + si.*Nsun5*: TDP-43 (37M)/+; gmr-GAL4 (YH3)/si.*Nsun5* [TRiP.HMS00438], gmr-GAL4 + TDP-43 + si.*Nsun6*: TDP-43 (37M)/si.*Nsun6*[TRiP.HMC04118]; gmr-GAL4 (YH3)/+, and gmr-GAL4 + TDP-43 + si.*Mt2*: TDP-43 (37M)/si.*Mt2*[TRiP.HMS01667]; gmr-gal4 (YH3)/+.

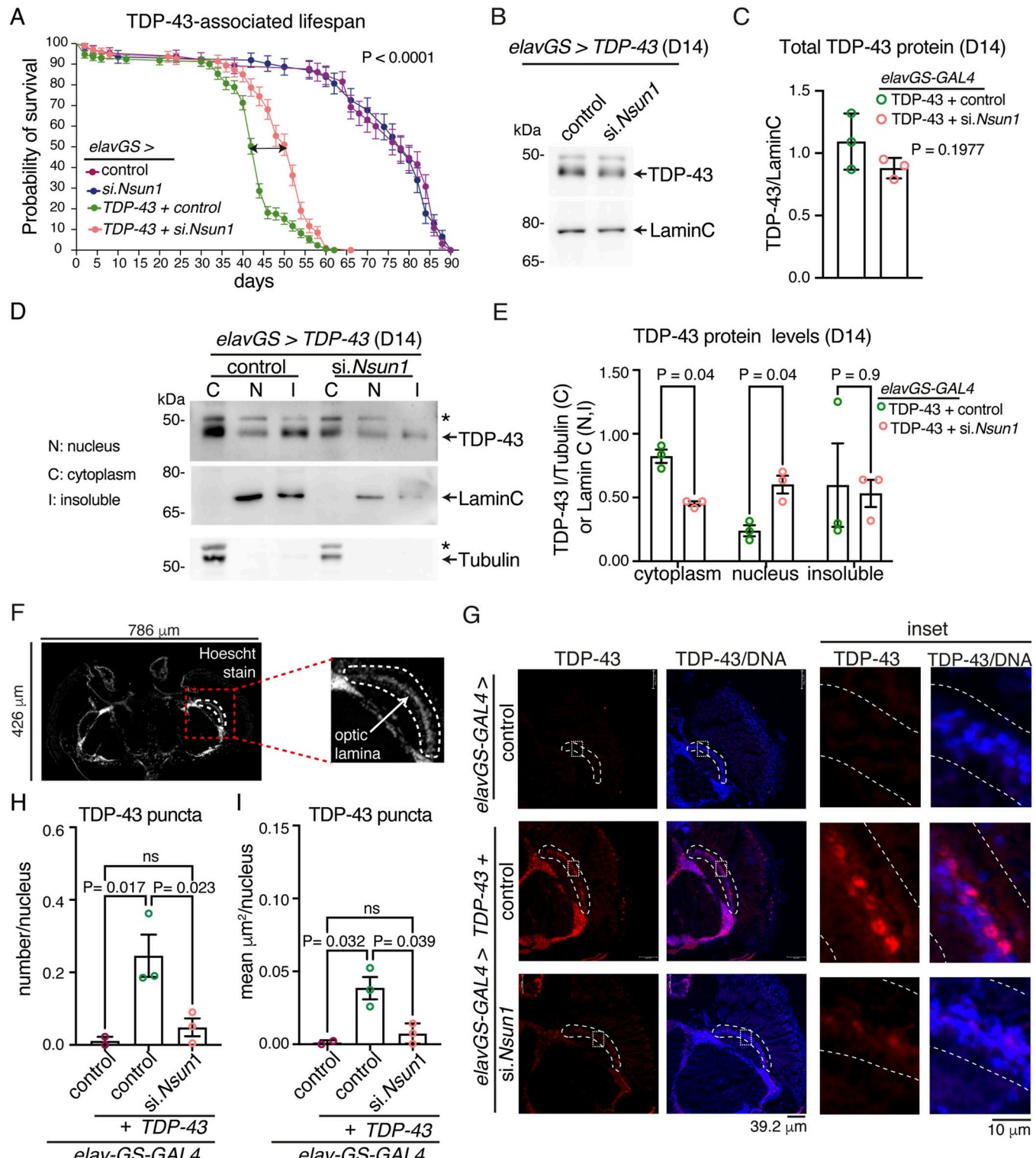

**Figure 4. Down-regulation of the 5-methylcytosine RNA methyltransferase *Nsun1* mitigates TDP-43 neurotoxicity and cytoplasmic accumulation in the Drosophila nervous system.**

**(A)** Reduction in *Nsun1* (si.*Nsun1*) mitigates lifespan deficits caused by expression of TDP-43 in adult neurons. Drosophila were fed 100 μl of 2 μg/ml RU486 topically on the food, >100 males per genotype were followed for each genotype, and a log-rank (Mantel-Cox) test was used for significance. **(B, C)** Reduction in *Nsun1* (si.*Nsun1*) does not alter total TDP-43 protein levels in adult neurons. Graph is the mean (±SD) of three biological repeats and a *t* test. **(D)** Reduction in *Nsun1* (si.*Nsun1*) reduces neuronal TDP-43 protein levels in the cytoplasm, increases TDP-43 protein levels in the nucleus and has no effect on insoluble TDP-43 levels. Top panel immunoblotted for TDP-43, middle panel for Lamin C and bottom panel for Tubulin. **(E)** TDP-43 protein levels quantified relative to the appropriate control (Lamin C for the nucleus, Tubulin for the cytoplasm and Lamin C for insoluble protein). Graph is the mean (±SD) of three biological repeats and multiple *t* tests; ns, not significant. **(F)** Micrograph of an example cryosection of an adult Drosophila head stained with Hoechst. The optic lamina is indicated as was the region selected for TDP-43 puncta

TDP-43 levels. Consistent with these biochemical findings, microscopic analysis of the adult brain revealed large cytoplasmic TDP-43 puncta in neurons expressing TDP-43 with *elavGS-GAL4* (Fig 4F and G). Strikingly, *Nsun1* down-regulation significantly reduced both the number and size of TDP-43 cytoplasmic puncta (Fig 4G–I). Collectively, these results demonstrate that *Nsun1* down-regulation in adult neurons promotes nuclear retention of TDP-43, reduces cytoplasmic accumulation, and alleviates TDP-43–induced lifespan deficits.

## TDP-43 activates NSUN1 methylation of RNA in Drosophila

NSUN1 catalyses methylation of a single, highly conserved cytosine in the 28S ribosomal RNA (rRNA) across species, including humans, *Caenorhabditis elegans*, and *Saccharomyces cerevisiae* (Hong et al, 1997; Janin et al, 2019; Liao et al, 2022). In addition to its enzymatic activity, NSUN1 also regulates rRNA processing independently of methylation (Hong et al, 1997; Hong et al, 2001; Liao et al, 2022). Given this, we hypothesized that TDP-43 might influence these rRNA-related functions of Nsun1 in Drosophila (Fig 5A). To assess the impact of TDP-43 on Nsun1 enzymatic activity, we measured $m^5C$ levels in total RNA from Drosophila head tissue expressing either control or TDP-43, with or without siRNA-mediated knockdown of *Nsun1* in the eye. Total RNA from Drosophila head tissue was DNase-treated, spotted onto a nylon membrane, and immunoblotted using an antibody specific for $m^5C$. This revealed that TDP-43 expression caused a dramatic and significant increase in $m^5C$ levels in total RNA compared with the normal control ($P = 0.034$) (Fig 5B and C). Notably, Drosophila has negligible $m^5C$ in genomic DNA (Gowher et al, 2000; Deshmukh et al, 2018), thus the signal arises from RNA rather than contaminating DNA. Importantly, co-expression of the siRNA to *Nsun1* with TDP-43 significantly reduced $m^5C$ in total RNA (Fig 5B and C), demonstrating that TDP-43–induced cytosine methylation is dependent on the Nsun1 protein.

To measure methylation of the Nsun1-target cytosine, we bioinformatically aligned the Drosophila 28S rRNA sequence to the corresponding homologues in *C. elegans*, *S. cerevisiae*, and human (Hong et al, 1997, 2001; Liao et al, 2022), and this showed that the Nsun1-target cytosine is conserved in Drosophila and is at position C3402 of the 28S rRNA (Fig 5D). To measure $m^5C$ levels of C3402, total RNA isolated from adult Drosophila was treated with sodium bisulphite, reverse transcribed, and the region surrounding C3402 was PCR amplified. Sanger sequencing of the subcloned PCR product showed that the conserved Nsun1-target cytosine (C3402) is methylated in adult Drosophila (Fig 5D). To determine how TDP-43 expression influenced methylation at this

site, $m^5C$ levels of C3402 were quantified by Illumina sequencing of the PCR product, comparing control, TDP-43, and TDP-43 and si.*Nsun1*. This showed that the Nsun1-target cytosine in Drosophila was constitutively methylated under control conditions (Fig 5E) and was unaltered by TDP-43 expression with or without *Nsun1* down-regulation (Fig 5E). These data suggest that the TDP-43–induced increase in cytosine methylation occurs at noncanonical sites rather than at the canonical Nsun1-target cytosine. To further assess whether TDP-43 or Nsun1 affects rRNA biogenesis, we measured the levels of the 47S rRNA precursor and the 18S, 5.8S, and 28S rRNA subunits using quantitate PCR. No change in the levels of 47S rRNA precursor or the 18S, 5.8S, and 28S rRNA subunits was observed (Fig 5F and G), indicating that TDP-43 and Nsun1 do not disrupt rRNA processing and that the observed increase in $m^5C$ arises from enhanced modification rather than altered rRNA abundance. Collectively, these data demonstrate that TDP-43 expression activates noncanonical Nsun1 cytosine methylation in RNA, and this is independent of the canonical Nsun1 target and rRNA processing activities, implicating noncanonical cytosines and/or RNA species as the likely substrates mediating TDP-43–dependent methylation.

## TDP-43 selectively interacts with isoform 3 of NSUN1 independently of RNA

To investigate the molecular basis of the TDP-43–NSUN1 interaction, we extended our studies to human cells. Human NSUN1 exists as four isoforms, all of which share a central, highly conserved methyltransferase domain flanked by N-terminal and C-terminal regions (Fig 6A). The N-terminal region contains both a nuclear localisation sequence and a nucleolar localisation sequence, the latter of which also functions as an arginine-rich RNA-binding domain (Gustafson et al, 1998). To establish an effective co-immunoprecipitation approach, we compared two NSUN1 antibodies with distinct epitopes: one targeting the N-terminal region, which spans sequence differences among isoforms 1, 2, and 3, and a second targeting the central methyltransferase domain (Fig 6A). The N-terminal antibody recognized two major splice variants—isoforms 1 and 2 (89 and 89.3 kD, respectively)—and the less abundant isoform 3 (93 kD) (Fig 6B). By contrast, the methyltransferase-domain antibody detected only a single NSUN1 isoform (Fig 6C). These findings were expected, as the N-terminal antibody targets a region that differs among NSUN1 splice variants, enabling recognition of multiple isoforms.

quantification. **(G)** Micrographs of Drosophila head cryosections immunolabelled for TDP-43 and counterstained with Hoechst. Hatched white line indicates the optic lamina and region quantified. Hatched white box indicates the area magnified in inset. Expression of TDP-43 in adult neurons leads to the formation of cytoplasmic TDP-43–lablelled puncta versus the negative control (*elavGS > si.mCherry*). Co-expression of the siRNA to *Nsun1* (si.*Nsun1*) reduces TDP-43 puncta. **(H)** The number of TDP-43–labelled puncta in the optic lamina was reduced by *Nsun1* reduction. Graph is the mean (±SD) of all animals over three biological repeats with one-way ANOVA and a Tukey's; ns is not significant. **(I)** TDP-43–positive puncta in the optic lamina were reduced in size by reduction in *Nsun1*. Graph is the mean (±SD) of all animals over three biological repeats (left), with one-way ANOVA and a Tukey's, ns is not significant. Genotypes are: **(A, B, C, D, E)** elavGS > control: *w\*; +/+; elavGS/si.mCherry$^{35785}$*, elavGS > si.*Nsun1*: *w\*; +/si.Nsun1$^{TRiP.HMJ24019}$; elavGS/+*, elavGS > TDP-43 + control: *w\*; +/+; elavGS, UAS-TDP-43-52S/si.mCherry$^{35785}$*, elavGS > TDP-43 + si.Nsun1: *w\*; +/si.Nsun1$^{TRiP.HMJ24019}$; elavGS, UAS-TDP-43-52S/+*. **(G, H, I)** elavGS > control: *w\*; +/+; elavGS/si.mCherry$^{35785}$*, elavGS > TDP-43 + control: *w\*; UAS-TDP-43-5X/+; elavGS/si.mCherry$^{35785}$*. elavGS > 5X-TDP-43 + si.*Nsun1*: *w\*; UAS-TDP-43-5X/si.Nsun1$^{TRiP.HMJ24019}$; elavGS/+*.
Source data are available for this figure.

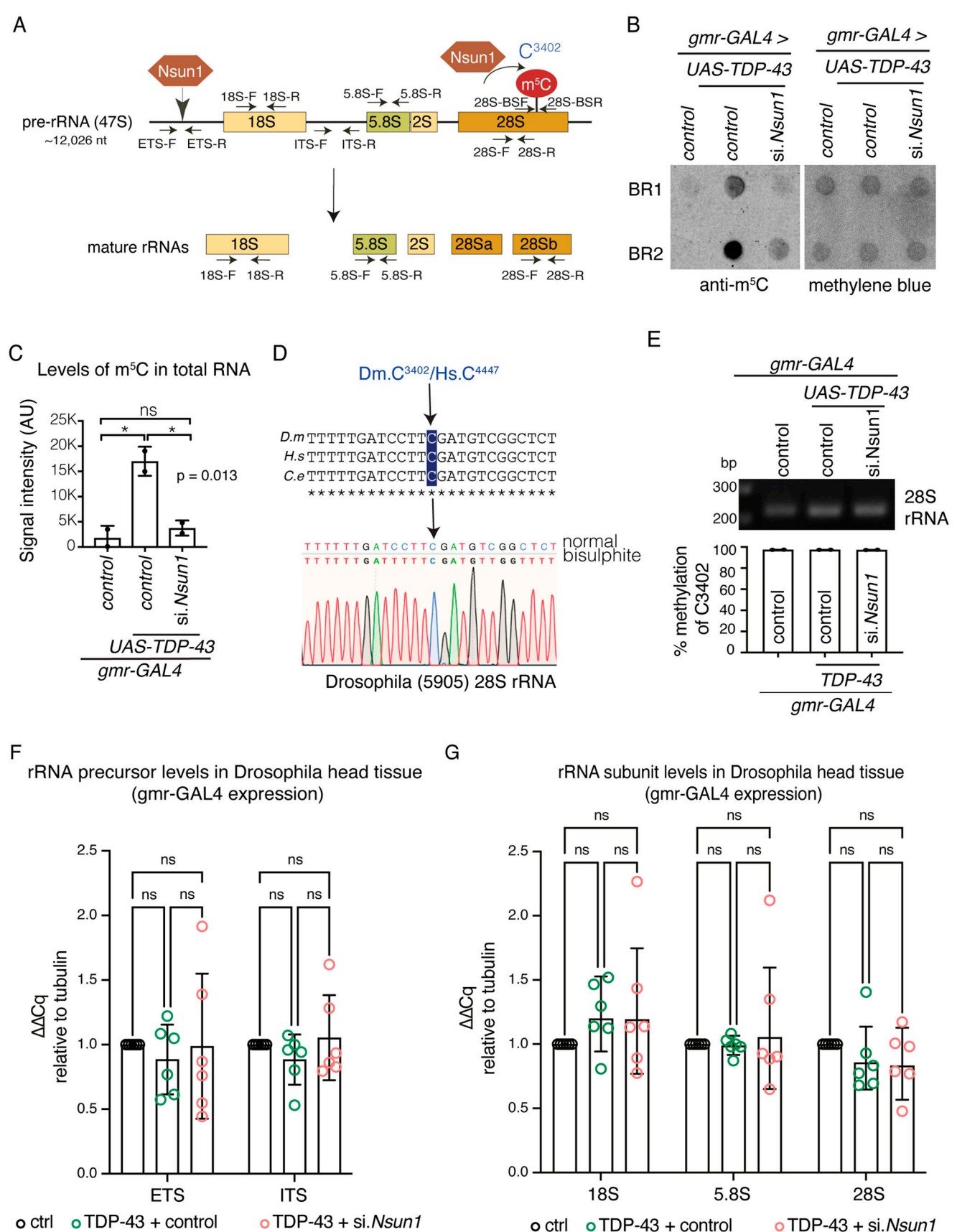

Using the N-terminal antibody, we immunoprecipitated endogenous NSUN1 from human U20S cells and immunoblotted for TDP-43. This demonstrated that NSUN1 consistently co-immunoprecipitated with endogenous TDP-43 (Fig 6D), validating our mass spectrometry data (Fig 1, Table S1). Conversely, immunoprecipitation of endogenous TDP-43 revealed a selective interaction with the less abundant NSUN1 isoform 3 in U20S and SH-SY5Y cells (Fig 6E and F). To determine whether this interaction is RNA-dependent, we expressed an RNA-binding–deficient mutant of TDP-43 (TDP-43-5F-L-YFP) (Buratti & Baralle, 2001), and assessed its ability to co-immunoprecipitate with NSUN1 compared with the WT form of TDP-43 (TDP-43-WT-YFP). TDP-43, lacking RNA-binding capacity, still co-immunoprecipitated NSUN1 in both U2OS and SH-SY5Y cells (Fig 6G and H), indicating that the interaction between NSUN1 and TDP-43 is not dependent upon RNA and may represent a protein–protein interaction. Furthermore, consistent with our data in Drosophila (Fig S6C–E), TDP-43 harbouring the disease-causing Q331K mutation co-immunoprecipitated with NSUN1 (Fig 6G), indicating that the interaction is maintained despite the pathogenic mutation. Collectively, these data show that TDP-43 and NSUN1 isoform 3 form a protein complex that is independent of RNA and that this interaction is resistant to the Q331K disease-causing mutation in TDP-43.

### TDP-43 and NSUN1 interact in the nucleolus and nucleoplasm, and their interaction increases upon nucleolar breakdown

To determine when and where TDP-43 and NSUN1 interact, we examined their localisation in U2OS cells (Fig S7A and B). As anticipated, NSUN1 co-labelled with the nucleolar proteins Fibrillarin (FIB) and Nucleophosmin (NPM1) (Fig S7A–C). By contrast, TDP-43 was diffusely nucleoplasmic and excluded from nucleoli (Fig S7D and E). Because TDP-43 localisation changes under stress, we tested actinomycin D—which inhibits rRNA transcription, induces nucleolar breakdown, and triggers TDP-43 translocation to the cytoplasm (Haaf & Ward, 1996; Ayala et al, 2008; Ederle et al, 2018; Duan et al, 2022). Upon actinomycin D treatment, the intensity, size, and number of NSUN1 nucleoli were reduced, whereas nucleoplasmic NSUN1 increased (Fig S7D and F–J), indicating that NSUN1 redistributes to the nucleoplasm after nucleolar breakdown. Consistent with nucleolar disruption–induced TDP-43 translocation to the cytoplasm, nuclear TDP-43 levels decreased (Fig S7J) and although nucleolar TDP-43 intensity remained largely

unchanged, the ratio of nucleolar to nuclear TDP-43 increased (Fig S7K–M). The redistribution of NSUN1 to the nucleoplasm suggests enhanced potential for NSUN1–TDP-43 interaction. To test this, we used proximity ligation assay (PLA), which detects protein–protein interactions within <40 nm. Under basal conditions, TDP-43 and NSUN1 formed a protein complex in the nucleus but not in the cytoplasm, compared with negative controls (Figs 7A–C and S8A and B). Actinomycin D treatment increased both the number and intensity of TDP-43–NSUN1 PLA foci, without affecting their size (Fig 7B). To determine the subnuclear localisation of these PLA foci, cells were counterstained with the nucleolar marker (Nucleolar Green). Approximately 75% of TDP-43–NSUN1 PLA puncta co-labelled with the nucleolus under normal conditions, and surprisingly this was not significantly altered actinomycin D treatment (Fig 7D and E), indicating that TDP-43 and NSUN1 function in both the nucelolus and in the nucleoplasm under normal and nucleolar stress conditions. By comparison, the co-immunoprecipitation between NSUN1 and TDP-43 remained largely unchanged after actinomycin D treatment (Fig 7E), which may reflect either the detection limits of the co-IP assay or that the PLA-detected increase in foci number and intensity represents closer proximity of the proteins rather than an increase in overall interaction. Together, these data suggest that TDP-43 and NSUN1 interact in the nucleolus as well as the nucleoplasm under basal conditions, and that this interaction is enhanced upon inhibition of RNA transcription and nucleolar breakdown.

### NSUN1 levels are altered in patient postmortem tissue

Our data showing that Nsun1 down-regulation mitigates TDP-43 toxicity in Drosophila (Figs 3, 4, 5, and S6) suggest that human NSUN1 may contribute to TDP-43–mediated pathology. This prompted us to examine whether NSUN1 undergoes pathological changes in ALS/FTD postmortem tissue. To test this, we immunolocalised NSUN1 in spinal cord sections from 10 individuals without neurological disease and 10 individuals diagnosed with FTD/ALS (Table S2). Consistent with our human cell data (see Fig S7), NSUN1 was enriched in the nucleolus of motor neurons in control tissue, with additional diffuse nucleoplasmic and cytoplasmic staining (Fig 8A). In FTD/ALS motor neurons, NSUN1 retained its nucleolar localisation and diffuse nucleoplasmic and cytoplasmic distribution, without forming cytoplasmic aggregates (Fig 8A, Table S3), suggesting that NSUN1 does not co-aggregate with pathological TDP-43.

**Figure 5. Expression of TDP-43 increases m5C in total RNA in a Nsun1-dependent manner without affecting rRNA processing.**
**(A)** The pre-47S pre-rRNA is depicted; the two Nsun1 proteins show sites of the rRNA that are regulated by Nsun1. The 5′ end shows the region of Nsun1 regulation of rRNA processing and the 3′ end shows the region of Nsun1 cytosine methylation. The primer positions for real-time PCR and bisulphite sequencing are presented. **(B, C)** TDP-43 expression causes a Nsun1-dependent increase in 5-methylcytosine ($m^5C$) in total RNA. Top blot is immunoblotted for $m^5C$ and lower blot is stained with methylene blue. BR, biological repeat. **(C)** TDP-43 expression causes a Nsun1-dependent increase in 5-methylcytosine ($m^5C$). **(C)** Graph is mean (±SD) of $m^5C$ levels in RNA relative to methylene blue (C) from ~100 heads per genotype for each of the two biological repeats, a one-way ANOVA and a Tukey's test is presented. *$P < 0.05$; ns, not significant. **(D)** The Nsun1-target cytosine and its methylation is conserved in Drosophila. Total RNA from WT Drosophila was bisulphite treated and sequenced to show the methylated cytosine (C3402). Dm, *Drosophila melanogaster*; Hs, *Homo sapiens*; Ce, *Caenorhabditis elegans*. **(E)** Bisulphite followed by Illumina sequencing showed that C3402 is nearly always methylated and is unaltered by TDP-43, or TDP-43 with si.*Nsun1*. Graph is mean (±SD), a one-way ANOVA, and a Tukey's test is presented; ns, not significant. **(F)** The levels of the 47S rRNA precursor are unaltered by TDP-43 expression or by TDP-43 with reduction in *Nsun1* (si.*Nsun1*). Graph is mean (±sem) of six biological repeats, one-way ANOVA, and a Tukey's test. Each data point is the mean of three technical replicates. **(G)** The levels of the 18S, 28S and 5.8S subunits are unaltered by TDP-43 expression or by TDP-43 with reduction in *Nsun1* (si.*Nsun1*), measured by real-time PCR. Graph is mean (±sem) of six biological repeats, one-way ANOVA, and a Tukey's test. Genotypes are gmr-GAL4 control: +/+; gmr-GAL4 (YH3)/si.mCherry$^{35785}$, gmr-GAL4 + TDP-43 + control: *TDP-43 (37M)/+; gmr-GAL4 (YH3)/si.mCherry$^{35785}$*, gmr-GAL4 + TDP-43 + si.Nsun1: *TDP-43 (37M)/si.Nsun1$^{TRiP.HMC04440}$; gmr-GAL4 (YH3)/+.*
Source data are available for this figure.

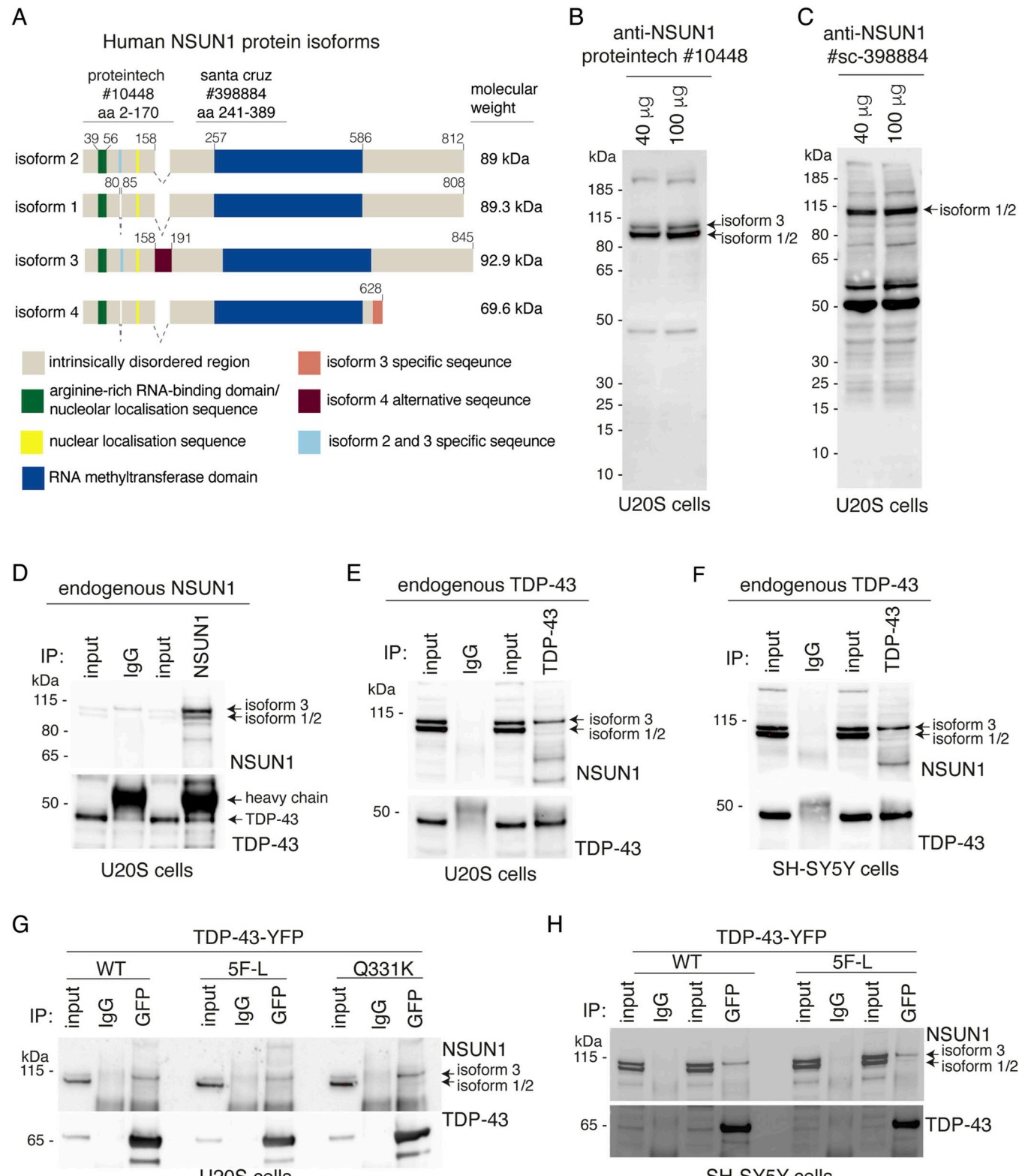

**Figure 6. TDP-43 selectively interacts with NSUN1 isoform 3, and the interaction persists in the absence of RNA and in the presence of the TDP-43 Q331K disease-causing mutation.**
**(A)** NSUN1 has a central RNA methyltransferase domain and an arginine-rich RNA-binding motif. There are four NSUN1 isoforms listed in NCBI and ensembl. These are isoform 1 (NCBI: NP_001028886.1 and ensembl: Nop2-212-i2), isoform 2 (NCBI: isoform 4, NP_001245239.1 and ensembl Nop2-218-i3), isoform 3 (NCBI: NP_001245238.1 and ensembl: Nop2-202-i4) and isoform 4 (NCBI: NP_001245239.1 and ensembl: Nop2-218-i3). The predicted molecular weight and conserved protein domains are listed. **(B)** The rabbit polyclonal antibody to NSUN1 detected two major NSUN1 protein bands at the predicted molecular weight in U2OS protein lysate. The antigen for the

Next, we analysed NSUN1 solubility in postmortem frontal cortex from controls and ALS/FTD patients. Proteins were sequentially extracted in buffers of increasing denaturing strength and were first immunoblotted for TDP-43 (Fig S9). As expected, the control cohort was negative for pathological forms of TDP-43—the C25 fragment and high molecular weight smear (Neumann et al, 2006) (Fig S9). In our ALS/FTD cohort, four of five samples were positive for TDP-43 pathology (Fig S9). Patient #13 was negative for frontal cortex TDP-43 pathology by Western blot; this patient was diagnosed with frontotemporal lobar degeneration (FTLD)-TDP type C, associated with the semantic variant of primary progressive aphasia (svPPA) (Borghesani et al, 2020). Unlike other FTLD sub-types, atrophy in type C begins in the anterior temporal lobe rather than the frontal cortex, with TDP-43 aggregates appearing as compact neuronal cytoplasmic inclusions in the temporal lobe with dystrophic inclusions in the frontal cortex (Neumann et al, 2021). It is possible that in patient 13, disease progression from the anterior temporal lobe to the frontal cortex was slow, explaining the absence of TDP-43 pathology in the frontal cortex. We next immunoblotted each biochemical fraction from all patients for NSUN1 and observed that in most patients (control and ALS/FTD), NSUN1 remained predominantly in the most soluble (low-salt) fraction (Fig 8C), consistent with the absence of insoluble aggregation with TDP-43 in ALS spinal cord. Immunoblotting of the low-salt fractions detected three NSUN1 isoforms: upper (isoform 3), middle (isoforms 1/2), and lower (isoform 4). Quantification of NSUN1 levels in ALS/FTD with confirmed TDP-43 pathology showed that NSUN1 isoforms 1/2 were significantly reduced (Fig 8D–G), whereas isoform 3—the NSUN1 isoform that preferentially interacts with TDP-43 (Fig 6)—and isoform 4 were unaffected (Fig 8D–G). Analysis including patient #13 is shown in Fig S10. The persistence of isoform 3 is particularly notable as it suggests that the isoform capable of interacting with TDP-43 remains available in disease, potentially enabling continued pathological interactions. Collectively, these data indicate that although NSUN1 does not co-aggregate with TDP-43, its isoform composition and potentially functional balance are altered in ALS/FTD postmortem tissue.

## Discussion

Using a global proteomics approach, we discovered that proteins that bind to the RNA modification m5C are among the top enriched molecular functions in our native TDP-43 interactome. This led us to a model for TDP-43 disease toxicity, whereby TDP-43 activates Nsun1-mediated methylation of cytosines in RNA, resulting in cytoplasmic accumulation of TDP-43, neurodegeneration, and lifespan deficits. Importantly, our data show that the TDP-43–induced increase in m5C does not occur at the canonical Nsun1-target cytosine in 28S rRNA; rather, TDP-43 may activate Nsun1 methylation at other cytosines, the identities of which remain to be determined. In humans, TDP-43 physically and selectively interacts with the longer NSUN1 isoform (isoform 3) independently of RNA in human neuronal-like cells, and this occurs in the nucleolus and nucleoplasm. In ALS/FTD postmortem frontal cortex, the stoichiometry of NSUN1 isoforms is significantly altered, with the shorter isoform (isoform 1/2) down-regulated and the longer isoform 3 persisting. The persistence of isoform 3, which selectively interacts with TDP-43, suggests that a pool of NSUN1 capable of contributing to pathological interactions remains in disease. These findings suggest that TDP-43 redirects NSUN1 activity toward novel RNA substrates and that NSUN1 isoform composition is reshaped in ALS/FTD.

RNA m5C methylation is an adaptive cellular response to stressors such as oxidative stress, heat shock, and nutrient deprivation (Chan et al, 2010, 2012; Schaefer et al, 2010; Blanco et al, 2014; Schosserer et al, 2015; Aguilo et al, 2016; Heissenberger et al, 2020). For example, in *S. cerevisae*, NSUN2 methylates tRNA[Leu] during oxidative stress to promote translation of TTG-codon-enriched mRNAs, whereas NSUN7 methylates enhancer RNAs during starvation to regulate transcription (Chan et al, 2010, 2012; Schaefer et al, 2010; Blanco et al, 2014; Aguilo et al, 2016). Levels of RNA methyltransferases, including NSUN1 and NSUN5 homologues in Drosophila and in *C. elegans*, respectively, are critical for propagating these stress responses and for lifespan regulation (Schosserer et al, 2015; Heissenberger et al, 2020). Consistent with this and upon TDP-43 expression, m5C in total RNA is significantly up-regulated in a Nsun1-dependent manner. Importantly, we show that this increase does not occur at the canonical Nsun1-target cytosine in 28S rRNA, suggesting that TDP-43 activates noncanonical Nsun1 methylation at other cytosines. Because antibody-based detection has limited sensitivity, future work using mass spectrometry or long-read direct RNA sequencing will better define the true rRNA methylation status with and without TDP-43, and it would have the potential to uncover additional sites across the transcriptome. Our data also show Nsun1 is essential during development but dispensable in adults, suggesting that its activation is context specific. These data suggest that pathological triggers, such as TDP-43 overexpression, can activate Nsun1 to methylate noncanonical cytosines in RNA. Thus, we suggest that specific triggers, such as TDP-43–induced degeneration, activate Nsun1 causing an increase in methylation at noncanonical, and still to be discovered, cytosines and target RNAs. This Nsun1-mediated methylation may reinforce TDP-43 cytoplasmic retention, providing a mechanistic link between RNA modification and the nuclear–cytoplasmic mislocalisation described in ALS/FTD.

rabbit polyclonal antibody was raised against the N-terminal region of NSUN1. We selected this antibody for our studies. **(C)** The mouse monoclonal antibody to NSUN1 detected one single high molecular weight band and several low molecular weight bands in U2OS protein lysate. The antigen for the mouse monoclonal antibody to NSUN1 spans the RNA methyltransferase domain (aa 241–389). **(D)** Endogenous TDP-43 co-immunoprecipitated with endogenous NSUN1 in U2OS cells. Top panel immunoblotted for NSUN1. Lower panel immunoblotted for TDP-43. **(E)** Endogenous NSUN1 co-immunoprecipitated with endogenous TDP-43 in U2OS cells. Top panel immunoblotted for NSUN1. Lower panel immunoblotted for TDP-43. **(F)** Endogenous NSUN1 co-immunoprecipitated with endogenous TDP-43 in SH-SY5Y cells. Top panel immunoblotted for NSUN1. Lower panel immunoblotted for TDP-43. **(G)** Endogenous NSUN1 co-immunoprecipitated with human TDP-43-WT-YFP, human TDP-43-5F-L-YFP and human TDP-43-Q331K-YFP in U2OS cells. Top panel immunoblotted for NSUN1. Lower panel immunoblotted for TDP-43. **(H)** Endogenous NSUN1 co-immunoprecipitated with human TDP-43-WT-YFP and human TDP-43-5F-L-YFP in SH-SY5Y cells. Top panel immunoblotted for NSUN1. Lower panel immunoblotted for TDP-43. Source data are available for this figure.

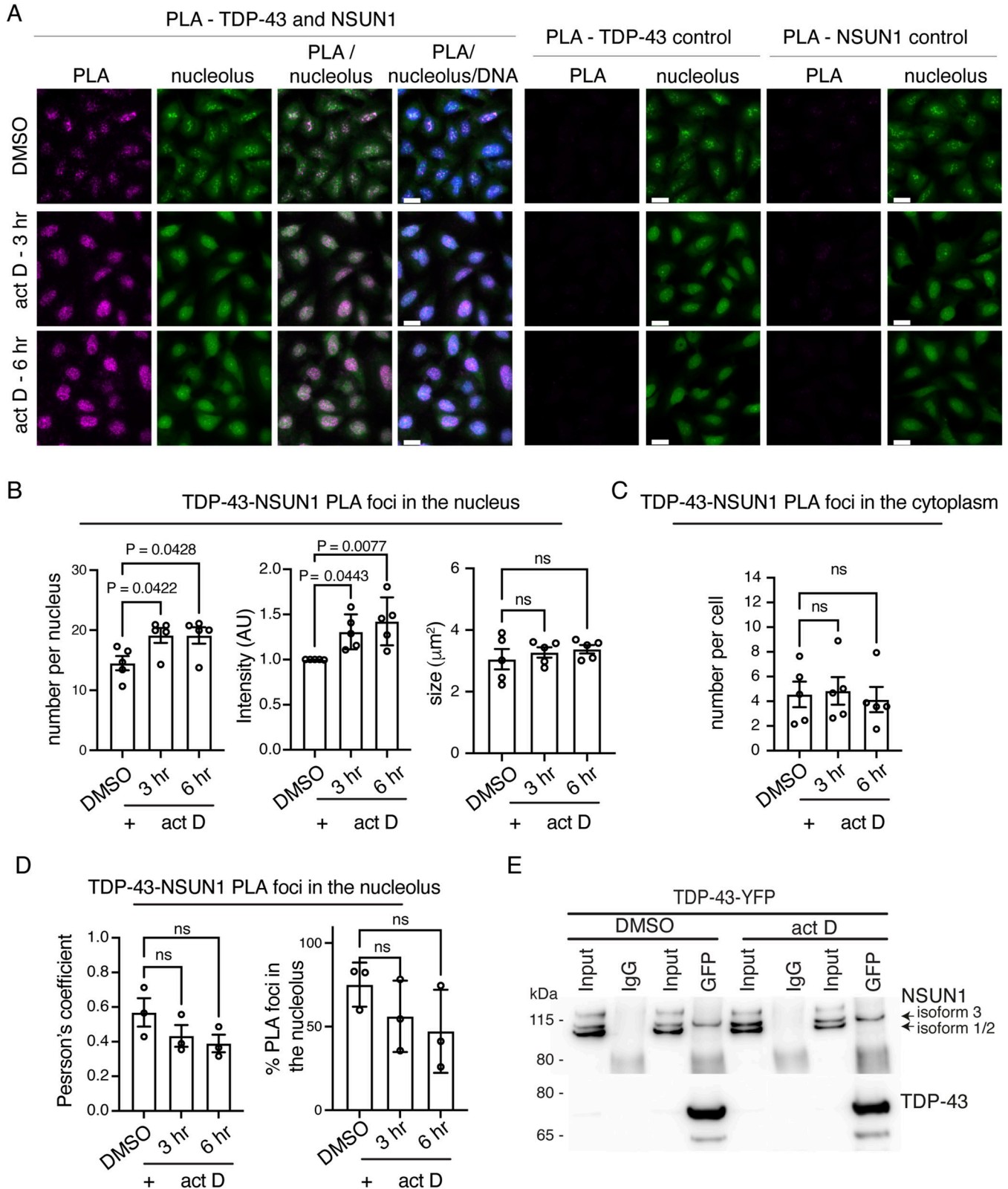

**Figure 7. NSUN1 and TDP-43 interact in the nucleolus and nucleoplasm.**
**(A)** PLA shows that TDP-43 and NSUN1 interact in the nucleolus and nucleoplasm and that the interaction increases upon 1 µg/ml actinomycin D (act D) treatment. Micrographs of U20S cells showing the PLA signal (magenta) counterstained with Nucleolus Bright Green and Hoechst. Scale bars are 25 µm. **(B)** Actinomycin D (act D) treatment (1 µg/ml) caused an increase in TDP-43-NSUN1 PLA puncta number and intensity but had no effect on TDP-43-NSUN1 PLA puncta size. Graph is mean (±sem) of five separate experiments, one-way ANOVA, and a Dunnet's test; ns, not significant. **(C)** TDP-43-NSUN1 PLA puncta in the cytoplasm were unaffected by actinomycin D (act

Previous transcriptomic studies show that TDP-43 and NSUN1 target largely divergent RNAs: TDP-43 binds mostly mRNAs with less than 2% being rRNA, whereas NSUN1 binds almost exclusively rRNA (98.32%) and very little mRNA (0.01%) (Polymenidou et al, 2011; Tollervey et al, 2011; Liao et al, 2022). This is consistent with our overall immunofluorescence observations in human neuronal-like cells, where TDP-43 and NSUN1 occupy mostly separate nuclear compartments. However, our proximity ligation assay (PLA) data reveal that under basal conditions, TDP-43 and NSUN1 form a nuclear protein complex, with ~75% of interactions localising to the nucleolus. These PLA results indicate that even when largely separated, a subset of TDP-43 and NSUN1 molecules are in proximity, potentially poised for functional interaction. Furthermore, upon nucleolar stress induced by actinomycin D, which inhibits RNA transcription, NSUN1 redistributes to the nucleoplasm. Concurrently, the number and intensity of TDP-43–NSUN1 complexes increase, and these complexes are present in both the remaining nucleoli and in the nucleoplasm, suggesting that TDP-43 and NSUN1 may function together in RNA regulation when transcription is inhibited. In ALS/FTD postmortem tissue, stress-induced nucleolar changes occur upstream of TDP-43 pathology (Haeusler et al, 2014; Mizielinska et al, 2017; Aladesuyi Arogundade et al, 2021), supporting a model for increased crosstalk between nucleolar proteins and TDP-43 in the propagation of disease. Finally, there are examples of potential converging pathways for TDP-43 and NSUN1. For instance, TDP-43 that is phosphorylated at threonine 153 (pT153) and tyrosine 155 (pY155) is only found in the nucleolus, this is enhanced by heat shock, and detectable only with a T153/pY155-specific antibody (Li et al, 2017). In addition, despite small RNAs being a lower-abundance target compared with their main RNA targets, both proteins bind them: NSUN1 associates with box C/D small nucleolar RNAs (snoRNAs), and TDP-43 binds small Cajal RNAs (scaRNAs), which promotes 2′-O-methylation of U1 and U2 small nuclear RNAs to regulate splicing (Izumikawa et al, 2019; Liao et al, 2022). This raises the intriguing possibility that TDP-43 may direct NSUN1 methylation via small nuclear or nucleolar RNAs. NSUN1 can also function in the nucleoplasm to methylate mRNAs in contexts such as kidney and ovarian cancer (Yang et al, 2023; Tian et al, 2024). Together, these observations hint that the noncanonical $m^5C$ methylation we detect in Drosophila may reflect aspects of a conserved pathway that underlies shared TDP-43– NSUN1 interactions.

Beyond $m^5C$, several RNA modifications have been implicated in regulating TDP-43 localisation and function. RNA modifications broadly influence RNA stability, splicing, and translation, fine-tuning cellular responses to stress and maintaining RNA homeostasis (Zaccara et al, 2019; Delaunay et al, 2024). Emerging evidence suggests that TDP-43 mislocalisation in ALS/FTD postmortem tissue correlates with increased $m^6A$-RNA levels, which may disrupt the stability of target RNAs (McMillan et al, 2023).

Elevated $m^6A$-RNA can also sequester reader proteins, as exemplified by cytoplasmic accumulation of the $m^6A$ reader YTHDF2 in motor neurons of ALS/FTD patients (McMillan et al, 2023). Similarly, methylation of CAG-trinucleotide repeats with $m^1A$ promotes sequestration of TDP-43 into cytoplasmic stress granules (Jiang et al, 2021). Although it remains unclear whether modified RNA itself or associated reader proteins regulate TDP-43, evidence supports a direct role of RNA modifications, as $m^6A$ and m1A enhance TDP-43 binding to RNA through indirect and direct mechanisms, respectively (Jiang et al, 2021; McMillan et al, 2023). Collectively, these studies support the idea that RNA modifications are critical regulators of TDP-43 localisation and disease-associated toxicity. Our work builds on this framework by showing that the $m^5C$ pathway is up-regulated by TDP-43 expression and promotes TDP-43 accumulation in the cytoplasm. We hypothesize that NSUN1, $m^5C$-RNA and/or the $m^5C$-RNA reader proteins identified in our native TDP-43 interactome (ALYREF, YBX1, YTHDF2, and C1QBP) may act as upstream triggers of TDP-43 mislocalisation. Finally, our data demonstrate that NSUN1 protein levels are altered in postmortem ALS/FTD frontal cortex. The shorter isoforms (1/2) are significantly reduced, altering NSUN1 isoform stoichiometry and potentially affecting its overall functionality. In contrast, isoform 4 and isoform 3—the latter being the isoform that preferentially interacts with TDP-43—persist, potentially enabling continued pathological interactions. This shift in isoform balance suggests a dual effect: loss of certain NSUN1 isoforms, coupled with maintenance of the TDP-43–interacting isoform that may propagate disease mechanisms. Although down-regulation of Drosophila Nsun1 is protective, these postmortem changes reflect end-stage disease, where affected neurons have degenerated and only resilient cells remain. Collectively, these findings indicate that multiple RNA modifications converge to influence TDP-43 localisation and toxicity, with our work highlighting NSUN1 and $m^5C$-RNA as previously unrecognized contributors to ALS/FTD pathogenesis.

# Materials and Methods

### Plasmids

Human TDP-43-WT-YFP and TDP-43-5F-L-YFP both in pcDNA3.2 were described previously (Elden et al, 2010). TDP-43-Q331K was made by performing site-directed mutagenesis using QuikChange II XL mutagenesis kit (#200521; Agilent) using primers listed in Table S3. The Drosophila plasmid pJFRC5-5XUAS-IVS-mCD8::GFP was obtained from addgene (#26218) and the mCD8-GFP was replaced with human TDP-43 from TDP-43-WT-YFP (pcDNA3.2).

### Drosophila stocks and maintenance

Drosophila stocks are described in Table S4. Transgenic strains for TDP-43 were previously described (Elden et al, 2010; Kim et al, 2014).

D) treatment (1 μg/ml). Graph is mean (±sem) of 5 separate experiments, one-way ANOVA, and a Dunnet's test; ns, not significant. **(D)** TDP-43-NSUN1 PLA puncta form in the nucleolus and nucleoplasm and this is unaltered by actinomycin D (act D) treatment (1 μg/ml). Graphs are mean (±sem) of three separate experiments, one-way ANOVA and a Dunnet's test; ns, not significant. **(E)** Endogenous Nsun1 isoform 3 co-immunoprecipitated with TDP-43-YFP in U20S cells and this is unchanged by actinomycin D (act D) treatment (1 μg/ml).
Source data are available for this figure.

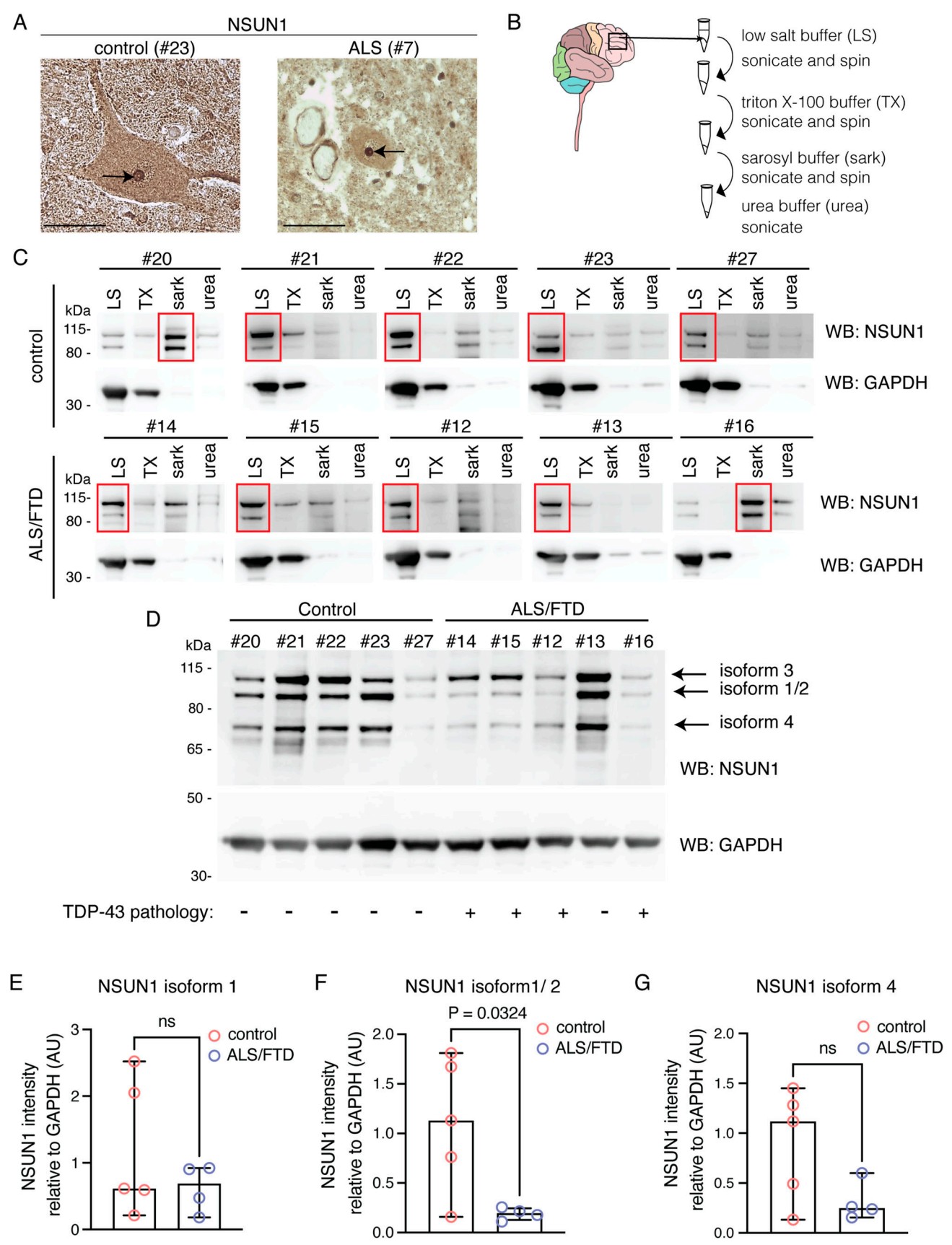

The siRNA, loss of function and GAL4 strains were obtained from the Bloomington stock centre, Indiana, USA. All experiments were carried out at 25°C in Bloomington Formulations (#FLY1004; Scientific Laboratories), or in Molasses Formulation: 27.03 g/liter inactive yeast (FLY1062; Scientific Laboratory Supplies), 72.13 g/liter yellow cornmeal (FLY1076; Scientific Laboratory Supplies), 90 ml/liter Molasses (FLY1296; Scientific Laboratory Supplies), 0.81 g/16.27 ml Tegosept (FLY1046; Scientific Laboratory Supplies), and 5.63 ml/liter Propionic Acid (402907; Sigma-Aldrich), unless otherwise stated. All fly strains are listed in Tables S4 and S5. The pJFRC5-5XUAS-IVS-TDP-43 plasmid was inserted into attP40 in *vas-int; attP40* (13-20; Stock) by the University of Cambridge Department of Genetics Fly Facility. This latter line was used for detecting the TDP-43 puncta on cryosections.

### External Drosophila eye imaging, paraffin sectioning and quantification

Female Drosophila aged 2–3 d were imaged with a Leica Z16 Apo A microscope, DFC420 camera and 2.0x planapochromatic objective, 0.034–0.224 nA, as described (François-Moutal et al, 2022). The area of eye degeneration was measured using ImageJ and compared with the total area of the eye. This was scored over three independent biological repeats. Control and experimental matings were propagated together at 25°C. For paraffin sections, female Drosophila heads were fixed, processed and quantified as previously described (McGurk et al, 2021). Sections were imaged on a Zeiss Axio imager A1 with a 20x objective with a 0.5 numerical aperture. The length of the internal retina was quantified using ImageJ (Rueden et al, 2017) from paraffin sections at the same anatomical position from one section per animal in three to five female heads from each genotype for each repeat.

### Drosophila lifespan

For lifespan analysis, >100 (1–2 d-old) Drosophila males were separated into groups of 20–25 and aged in vials of standard Bloomington formula made with 10% extra water. Drosophila food vials were inoculated with 100 µl of 2 mg/ml RU486. Dead Drosophila were scored every 2 d and survivors were tipped into a fresh vial inoculated with 100 µl of 2 mg/ml RU486. Survival curves and a log-rank test for trend were performed using GraphPad prism 6. The lifespan analysis was repeated at different times throughout the year.

### Nuclear and cytoplasmic extraction from Drosophila head tissue

Nuclear–cytoplasmic fractionation was performed as previously described (McGurk et al, 2018a). Briefly, the NER-PER nuclear and cytoplasmic extraction kit (# 78833; Thermo Fisher Scientific) was used according to manufacturer's instructions with the following modifications: 10 male heads were homogenised in 110 µl of ice-cold CERI for 1 min, the sample was vortexed on the highest setting for 30 s, incubated on ice for 10 min, 5.5 µl of ice-cold CER II was added. And the sample was vortexed for 10 s on the highest setting. The samples were incubated on ice for 2 min, vortexed on the highest setting for 15 s and centrifuged at maximum speed for 10 min at 4°C. The supernatant (cytoplasmic extract) was transferred to a clean pre-chilled tube and centrifuged at maximum speed for 10 min at 4°C. The supernatant (cytoplasmic extract) was transferred to a pre-chilled tube and stored at –20°C. The nuclear pellet was washed in 50 µl of ice-cold CERI and centrifuged at maximum speed for 10 min at 4°C. The supernatant was discarded, and the pellet was resuspended in 55 µl of ice-cold NER and vortexed on the highest setting for 10 s every 10 min for a total of 40 min and centrifuged at maximum speed for 10 min at 4°C. The supernatant (nuclear fraction) was transferred to pre-chilled tube. Samples for immunoblotting were made up in 1× LDS Sample Buffer (# NP0007; Thermo Fisher Scientific) with 5% β-mercaptoethanol (#M3148; Sigma-Aldrich), heat denatured at 95°C, chilled on ice for 5 min and centrifuged at 2,300g, 5 min at 4°C. Protein samples were electrophoresed on a 4–12% Bis-Tris gel (#NP0323; Thermo Fisher Scientific) with NuPAGE MOPS buffer (#NP0001; Thermo Fisher Scientific) and transferred onto 0.45 µm nitrocellulose by wet transfer (20 V for 75 min) and in SDS NuPAGE Transfer Buffer (# NP0006; Thermo Fisher Scientific). Blots were blocked in 5% milk in TBST (TBS with 0.05% TWEEN-20) for 1 h at RT with gentle rocking, and in primary antibody made up in TBST overnight at 4°C with gentle rocking. Blots were washed in TBST for 5 min at RT with rocking (four times), incubated in secondary antibody made up in TBST for 1 h at RT with rocking, washed in TBST for 5 min at RT with rocking (four times) and signal was detected using ECL Select Western blotting (Cat # RPN2235; Amersham). For nuclear fractions, the equivalent of one head was electrophoresed and for cytoplasmic fractions the equivalent of 0.5 head was electrophoresed.

Total TDP-43 or β-galactosidase protein levels were measured as described previously (McGurk et al, 2018a). Briefly, protein was extracted from 10 female heads in 100 µl of 2X Laemelli buffer, 5% (vol/vol) β-mercaptoethanol, denatured at 95°C for 5 min, chilled on ice for 5 min and centrifuged at 2,300g for 5 min at 4°C. The

**Figure 8. In ALS/FTD, NSUN1 remains nucleolar but NSUN1 isoform stoichiometry is altered.**
**(A)** Representative micrographs of control and ALS spinal cord postmortem tissue immunolabeled for NSUN1 and counterstained with hematoxylin. Arrow indicates nucleolus in a motor neuron. Scale bar is 100 µm. **(B)** Schematic of the biochemical fractionation of frontal cortex tissue. **(C)** Protein isolated from frontal cortex tissue by sequential biochemical fractionation and immunoblotted for NSUN1 showed that in four of five control samples and four of five ALS/FTD samples NSUN1 was mostly in the soluble fractions (red boxes). LS, low-salt buffer; TX, Triton X-100 buffer; sark, sarkosyl buffer; urea, urea buffer. **(D)** The low-salt protein fraction isolated from control and ALS/FTD frontal cortex immunoblotted for NSUN1 and GAPDH. **(E)** The levels of NSUN1 isoform 3 is unaltered in ALS/FTD frontal cortex. NSUN1 is quantified relative to the GAPDH control on the same blot. Graph is the mean (±SD), t test; ns, not significant. **(F)** The levels of NSUN1 isoform 1/2 is significantly reduced in ALS/FTD frontal cortex. NSUN1 is quantified relative to the GAPDH control on the same blot Graph is the mean (±SD) and a t test. **(G)** The levels of NSUN1 isoform 4 are unaltered in ALS/FTD frontal cortex. NSUN1 is quantified relative to the GAPDH control on the same blot. Graph is the mean (±SD), t test; ns, not significant.
Source data are available for this figure.

equivalent of half a Drosophila head (5 $\mu$l) was electrophoresed for each sample. Primary antibodies were made up in TBST and were: TDP-43 (1 in 10,000, 10782-2-AP; Proteintech), Tubulin-HRP (1 in 5,000, #9099S; Cell Signalling Technology), and mouse Lamin C (1 in 1,000 # ADL101-s; Developmental Studies hybridoma bank) and $\beta$-galactosidase (1 in 5,000, #Z378A; Promega). HRP-coupled secondary antibodies used: goat antibody to rabbit (1 in 5,000, #AP307P; EMD Millipore) and goat antibody to mouse (1 in 10,000 #ab6789; Abcam). All experiments were carried out on three or more biological replicates, blots were quantified with ImageJ (Rueden et al, 2017). The area under the curve for each band was measured using ImageJ for both the experimental protein (TDP-43/$\beta$-galactosidase) and the loading control (Tubulin/LaminC). Each protein band was calculated as a percentage of the total intensity across the entire data set on the gel (this was performed for each protein), and the experimental band was made relative to the appropriate loading controls. For nuclear and cytoplasmic blots, nuclear TDP-43 was made relative to the Lamin C, whereas cytoplasmic TDP-43 was made relative to Tubulin. GraphPad prism 9 software was used to calculate statistical significance. For all blots, the control protein (Tubulin/LaminC) was detected on the same blot as TDP-43 or $\beta$-galactosidase.

### Drosophila immunofluorescence

Cryosections and immunostaining of Drosophila heads were performed as previously described (McGurk & Bonini, 2012). Briefly, female Drosophila were fed 100 $\mu$l of 2 mg/ml RU486 for the indicated times and were transferred to fresh RU486-innoculated food every 2 d, for 4 or 7 d. The heads were embedded in O.C.T. (Tissue-Tek), 12 $\mu$m serial sections were cut and mounted on slides. Tissue sections were warmed on a hot plate (37°C for 30 min), fixed in 0.4% PFA in PBS for 30 min and at 37°C, stained overnight in mouse TDP-43 (1 in 200, # 60019-2-IG, Proteintech), (4°C), followed by secondary antibody (1 in 500, donkey anti mouse Alexa Fluor 594) for 3 h at RT and finally Hoechst (0.5 $\mu$g/ml, 15 min). Images were acquired on a Leica SP8 confocal microscope using identical settings across genotypes and used an HCPL APO 20X objective with a numerical aperture of 0.75. ImageJ was used to quantify TDP-43 positive puncta and a GraphPad prism was used to test for significance.

### RNA isolation and real-time PCR

The levels of the m5C-RNA methyltransferases were measured as described previously (François-Moutal et al, 2022). For *Nsun1* knockdown by *DaGS-GAL4*, 10–12 adult males were aged for 5 d on Bloomington standard cornmeal medium inoculated with 100 $\mu$l of 4 mg/ml RU486 at 25°C. Drosophila were transferred on to fresh RU486-innoculated food every 2 d. All other methyltransferases were mated to *Da-GAL4* and were collected on day 1. For all matings, the abdomen was removed, and the remaining tissue homogenised in 200 $\mu$l of TRIzol. For rRNA measurements, 10 female heads were homogenised in 200 $\mu$l of TRIzol. A further 200 $\mu$l of TRIzol was added, and the samples were vortexed for 30 s at RT. The sample was resuspended five times with a syringe (21Gx1 needle, 1 ml syringe), and 150 $\mu$l of chloroform was

subsequently added. Samples were shaken by hand, incubated at RT for 3 min, and centrifuged at 12,000g, at 4°C for 15 min. The upper aqueous phase was purified using the RNA clean and Concentrator –25 (#R1017; Zymo Research) and included in-column DNase I treatment. First-strand cDNA was synthesized using Superscript III (# 18080-051; Thermo Fisher Scientific). For measurements of the methyltransferases, 500 ng of RNA was used in the cDNA reaction, whereas for the rRNA 100 ng of RNA was used. In all cases, RNA was synthesized into cDNA using random primers. Luna real-time PCR mix (NEB, cat # M3003) was used with either the Bio-Rad CFX Connect Real-Time PCR Detection System or Quant-Studio 6 Flex Real-Time PCR System, the latter used ROX Reference Dye (cat# 12223012; Invitrogen) for normalisation. Each experiment was carried out on Drosophila adults prepared from three independent crosses, unless otherwise stated in the legends. Primers are listed in Table S6. GraphPad prism was used to test for significance.

### RNA dot blot and bisulphite RNA sequencing

The RNA from ~100 male and female heads were homogenised in 200 $\mu$l TRIzol (Thermo Fisher Scientific) and subsequently made up to 1 ml with 800 $\mu$l of TRIzol. Samples were vortexed for 30 s, passed through a 20G1/2 needle and 1 ml syringe five times, and incubated at RT for 5 min. RNA was chloroform extracted using 200 $\mu$l chloroform, followed by vigorous shaking for 30 s and incubation at RT for 3 min. Samples were centrifuged and 12,000g for 15 min at 4°C. A second chloroform extraction was performed using 200 $\mu$l chloroform. RNA was precipitated overnight in ethanol and 3 M sodium acetate pH 5.2 (Thermo Fisher Scientific) at –80°C. Samples were centrifuged at maximum speed at 4°C for 30 min. The RNA pellet was washed in 70% ethanol and centrifuged at maximum speed at 4°C for 15 min. The pellet was resuspended in 50 $\mu$l RNase free water. DNase was digested using turbo DNase (Thermo Fisher Scientific). For the dot blot, RNA was denatured at 95°C for 5 min, chilled on ice 5 min, and centrifuged at 1,300g for 4 min at 4°C. 3 $\mu$g of RNA prepared in the TBE-Urea Sample Buffer was spotted onto a nylon membrane. Membranes were UV cross-linked at 0.120 J, washed in TBST (0.1% tween) for 10 min at RT with rotation. The membrane was blocked in 5% milk made up in TBST (0.1% tween) with rotation at RT 1 h. Blots were incubated in primary antibody overnight at 4°C with rotation, followed by 3 × 5 min washes in TBS + 0.02% tween and incubation in secondary antibody at RT for 2 h with rotations. Membranes were washed 4 × 5 min in TBS + 0.02% tween. Signal was detected using ECL select with accumulating capture of 1 min for total of 10 min. Antibodies used were anti-rabbit m5C (1 in 300, ab214727, RRID:AB_2802117; Abcam) and goat anti-rabbit HRP (1:10,000, Cat# 115-035-146, RRID: AB_11212848; EMD Millipore).

For bisulphite sequencing, 1,000 ng of total RNA was bisulphite treated according to the EZ RNA methylation kit (R5001; Zymo Research) and eluted in 13 $\mu$l of water by incubating water on column for 4 min at RT before elution by centrifugation. 100 ng of the resulting RNA underwent cDNA first-strand synthesis using random primers and Superscript III (Thermo Fisher Scientific) followed by RNase digestion with 10 $\mu$l (100 U) RNase I for 5 min at RT. The region surrounding the Nsun1-target cytosine was

amplified by PCR from 4 µl of cDNA using the KAPA2G robust PCR system (KK5004; Merck), with buffer A and with the addition of an extra 0.25 mM $MgCl_2$ and double the recommended amount of Taq. The primers were deaminated and are listed in Table S6. The PCR programme used was as follows:

95°C 3 min
95°C 15 s*
55°C 15 s*
72°C 1 min*
72°C 4 min
*40 cycles

For Sanger sequencing, the PCR product was subcloned into TOPO TA (Thermofisher TOPO TA Cloning Kit for Sequencing # 450030). For Illumina sequencing, four reactions for each genotype were pulled and purified with QIAGEN's PCR purification kit (#28104; QIAGEN) and eluted in 13 µl pre-warmed (55°C) RNase/DNase free water. 25 µl at a concentration of 20 µg/µl underwent Amplicon-EZ sequencing at Azenta, with 2 × 250 bp coverage and ~50,000 reads per sample.

## Methylation analysis

To assess potential multimapping issues, the sequenced reads were first mapped to the *D. melanogaster* genome BDGP6.32, downloaded from Ensembl (release 107), using *bowtie2* version 2.4.5. The 28S sequence pdb|3J3E|5 from genbank NCBI was added to the reference. All C nucleotides were converted to T before mapping, to account for the bisulfite conversion in data. Five regions in the genome were identified with mapping depth of at least 30, albeit not in canonical chromosomes, but in additional scaffolds and rDNA sequences. Bismark version 0.23.1 was used to detect methylation sites. 10 sites were found in the CpG context in the pdb|3 J3E|5 sequence, although none of them was methylated except for the locus of interest, C3402, which was found to be highly methylated in all samples. The code for this analysis (Snakemake file and the R code for downstream processing) is available from https://github.com/bartongroup/MG_Methyl28S.

## Mammalian cells and culture details

U20S cells and SH-SY5Y cells were a kind gift from Dr Adrien Rousseau (MRC-PPU, University of Dundee) and Prof. Mirital Muqit (MRC-PPU, University of Dundee), respectively. U20S and SH-SY5Y cells were routinely grown in DMEM containing high glucose, L-glutamine and sodium pyruvate (#41966052; Thermo Fisher Scientific), 10% filter-sterile FBS (#10270106; Thermo Fisher Scientific) and antibiotic-antimycotic (#15240062; Thermo Fisher Scientific). Cells were grown at 37°C with 5% $CO_2$, with a water bath for humidification. Cells were washed with Dulbecco's PBS without calcium or magnesium (#14190169; Thermo Fisher Scientific) and trypsinized in trypsin with 0.25% EDTA (# 25200056; Thermo Fisher Scientific).

## Co-immunoprecipitation and immunoblotting

TDP-43 was co-immunoprecipitated from SH-SY5Y and U20S cells as previously described (McGurk et al, 2020), but with some minor alterations. Briefly, for all co-immunoprecipitations, 12 µg of antibody or normal IgG was bound to 50 µl of dynabeads protein G (#10004D; Thermo Fisher Scientific). For co-immunoprecipitation of endogenous TDP-43 and NSUN1 (including the mass spectrometry studies), a T75 flask was seeded the day before with 4 × 10[6] U20S cells or 14 × 10[6] SH-SY5Y cells. Cells were grown overnight under standard conditions. One fully confluent T75 flask of U20S and SH-SY5Y cells (equivalent to 9 × 10[6] cells and 28 × 10[6] cells, respectively) were trypsinized in trypsin with 0.25% EDTA, pelleted, and washed in Dulbecco's PBS without calcium and without magnesium. The cell pellet was resuspended in 1 ml ice-cold Pierce IP Lysis Buffer (#8778; Thermo Fisher Scientific) with Halt Protease Inhibitor Single-Use Cocktail (#78443; Thermo Fisher Scientific) incubated on ice (10 min), passed three times through a 21G1 1/2 needle attached to a 1 ml syringe and transferred to centrifuge tubes, rotated at 15 rpm at 4°C (15 min) on a rotary mixer. The cell lysate was centrifuged at 16,873 g for 10 min at 4°C and the supernatant was collected, 25 µl of lysate was removed for input. 500 µl of the 1 ml lysate was either incubated with IgG beads or the antibody-bound beads and reaction volumes were made up to 1 ml with lysis buffer containing protease inhibitor. Co-immunoprecipitations were incubated at 4°C with 15 rpm rotation (18 h) on a rotary mixer. The beads were washed three times in 500 µl lysis buffer containing protease inhibitor. To elute, beads were incubated at 95°C for 5 min in 1× LDS Sample Buffer (# NP0007; Thermo Fisher Scientific) containing 5% β-mercaptoethanol. Input samples were denatured at 95°C for 5 min in 1× LDS Sample Buffer containing 5% β-mercaptoethanol. Protein was electrophoresed on NuPAGE 4–12% Bis-Tris gels in NuPAGE MOPS buffer and transferred onto 0.45 µm nitrocellulose in the SDS transfer buffer using the XCell II Blot Module at 30 V for 90 min. Membranes were incubated in TBS with 0.05% Tween-20 (TBST) containing 5% non-fat dry milk with rocking for 1 h at RT and in primary antibody in TBST with rocking at 4°C overnight. Membranes were washed 5 × 5 min in TBST, incubated in secondary antibody in TBST for 1 h at RT, washed 5 × 5 min in TBST and incubated in Amersham ECL Select Western blotting system for 5 min in the dark.

For transfected TDP-43-YFP, a T75 flask was seeded with 4 × 10[6] U20S cells or 7 × 10[6] SHSY57 cells and incubated overnight in DMEM with high L-glutamine, sodium pyruvate, 10% filter-sterile FBS and antibiotic-antimycotic for ~20 h. The following day, the media were replaced with DMEM with high glucose and L-glutamine and 10% FBS. Cells were inoculated with the transfection mixture consisting of 8 µg of plasmid DNA, 8 µl PLUS reagent and 28 µl LTX (#15338100; Thermo Fisher Scientific) in 1,600 µl OPTIMEM (# 31985062; Thermo Fisher Scientific) and incubated for at 37°C for 21 h upon which time they were lysed in 1 ml Pierce lysis buffer (#87787; Thermo Fisher Scientific) as described above. Beads were incubated with 1 ml of protein lysate obtained from one T75 flask. Co-immunoprecipitation and immunoblotting were carried out as described above. Antibodies for co-immunoprecipitation were mouse anti-TDP-43 (#60019-2-IG, RRID: AB_2200520; Proteintech), sheep GFP (#S268B; MRC-PPU), mouse GFP (#A-11120, RRID:AB_221568; Thermo Fisher Scientific), rabbit NSUN1 (10448-1-AP, RRID:AB_2282772; Proteintech), normal mouse IgG (#sc-2025; Santa Cruz Biotechnology), normal sheep IgG (#12-515; EMD Millipore), normal rabbit IgG (#2729s; Cell Signaling Technology). Antibodies for immunoblotting were rabbit anti-TDP-43 (1:10,000, Cat# 10782-2-AP, RRID: AB_615042; Proteintech), rabbit anti-NSUN1 (1:5,000 #10448-1-AP; Proteintech) and goat anti-rabbit HRP (1:10,000, Cat# 115-035-146, RRID:AB_11212848; EMD Millipore).

## Sample processing and mass spectrometry analysis

Samples were electrophoresed on a 1D SDS–PAGE for 15 min; gels were stained with Quick Coomassie Stain and the whole area of stained protein was excised and subjected to in-gel digestion with 1 mg/ml Trypsin (Thermo Fisher Scientific) at a final concentration of 12.5 $\mu$g/ml. Digested peptides were run on a Q-Exactive Plus (Thermo Fisher Scientific) coupled to a Dionex Ultimate 3000 HPLC system (Thermo Fisher Scientific). A 2–35% B gradient comprising eluent A (0.1% formic acid) and eluent B (80% acetonitrile/0.1% formic acid) was used to run a 120-min gradient. Each fragment was run consecutively, and blanks used between samples. The top 10 most intense peaks from a mass range of 350–1,600 m/z in each MS1 scan with a resolution of 70,000 were then taken for MS2 analysis at a resolution of 17,500. Spectra were fragmented using higher-energy C-trap dissociation (HCD). Label-free analysis was performed in MaxQuant version 1.6.6.0 using the RAW files generated with data manipulation in Microsoft Excel Office 365 to provide comparisons between control and treated samples.

## Immunofluorescence

Immunofluorescence was carried out as previously described (McGurk et al, 2020). Briefly, glass coverslips, in a 24 well plate, were seeded with 110,000 U20S cells and were cultured as described above. The next day, cells were fixed in 4% PFA for 15 min and permeabilized three times in PEM-T buffer (100 mM PIPES, 1 mM MgCl$_2$, 10 mM EGTA pH 6.8% and 0.1% Triton X-100), for 3 min each. Cells were blocked in 10% normal donkey serum (Sigma-Aldrich) in TBS with 0.05% Tween-20 (TBST) and primary antibodies in TBST were applied overnight at 4°C in a humidified chamber. Cells were washed three times in PEM-T buffer (3 min each); then, secondary antibody in TBST for 45 min at RT and in the dark. Cells were washed three times in TBST (3 min each), stained with 1 $\mu$g/ml Hoechst 33342 (Thermo Fisher Scientific) for 15 min, washed in deionized H$_2$O and mounted in ProLong Diamond (Thermo Fisher Scientific). Images were acquired on a Leica SP8 confocal microscope using an HCPL APO 20X objective with a numerical aperture of 0.75. Images were quantified using Image J software and data plotted in GraphPad Prism. Primary antibodies used were mouse TDP-43 (1 in 500, # 60019-2-IG; Proteintech), rabbit anti-NSUN1 (1:1,000 #10448-1-AP; Proteintech), Nucelophosmin (1 in 500, #ab86712, RRID:AB_10675692; AbCam) and Fibrillarin (1 in 500, #ab4566, RRID:AB_304523; Abcam). Secondary antibodies used were goat anti-rabbit Alexa Fluor 488; goat anti mouse Alexa Fluor 488; donkey anti-mouse Alexa Fluor 594; goat anti-rabbit Alexa Fluor 594; goat anti-rabbit Alexa Fluor 647; goat anti-mouse Alexa Fluor 647 (All 1 in 500, #A11008, #A10680, #A32744, #A11012, #A21244, #A21235; all Thermo Fisher Scientific).

## Proximity ligation assay and high content image acquisition and quantification

A total of 20,000 U20S cells were seeded onto a 96-well plate; the next day, cells were treated with either DMSO or 1 $\mu$g/ml actinomycin D for the indicated time. Cells were fixed in 4% PFA for 15 min,

permeabilized three times in TBS with 0.1% Triton X-100, for 3 min for each. Cells were blocked and processed using the proximity ligation assay from Navinci (#60025; NaveniFlex Cell Red). Antibodies used were rabbit anti-NSUN1 (1:2,000 #10448-1-AP; Proteintech) and mouse TDP-43 (1 in 500, # 60019-2-IG; Proteintech). Cells were counterstained with 0.5 $\mu$mol/liter Nucleolus Bright Green (N511-10; Insight Biotechnology) for 5 min at RT. Controls were used as advised for the NaveniFlex Cell Red, which removes one antibody form the incubation. Cells were imaged on an Operetta CLS1601 (Revvity): imaging was performed using a 20x objective (0.4 NA) with 2x binning. The acquisition settings used for each fluorophore were: Hoechst 10% power, 20 ms exposure with 355–385 Excitation filter (Ex) and 430–500 Emission filter (Em); AlexaFluor488 50% power, 50 ms exposure with 460–490 Ex and 500–550 Em; and AlexaFluor594 50% power, 300 ms exposure with 560–575 Ex and 570–650 Em. 12 fields of view were acquired per well of a 96-well plate. Image analysis was performed using CellProfiler (4.2.80002). Eight non-overlapping images at 20X were taken for each well and used for quantification; ~2,000 cells were quantified per well. An individual experiment contained all conditions in duplicate or triplicate and was performed on separate weeks. An Otsu thresholding method was used to identify all objects: nuclei were identified by the Hoechst image; nucleoli on a processed (enhanced and edged) AlexaFluor488 image; PLA puncta on a processed (enhanced) AlexaFluor594 image. Puncta with a aax-imum intensity below an experimentally determined value were discarded. Retained puncta were related to both nuclei and nucleoli to determine whether they were located inside the larger objects. Measurements were made by CellProfiler using the MeasureObjectSizeShape, MeasureObjectIntensity, and MeasureColocalization modules. Image intensity and threshold values (Final Threshold, Sum of Entropies, and Weighted Variance) determined by CellProfiler were used to identify and discard images of poor quality.

## Human samples and data

To ensure that experiments involving human subjects conformed to the principles of the WMA Declaration of Helsinki and the Department of Health and Human Services Belmont Report, ethical approval for the human tissue work was reviewed and granted by the access committee at the London Neurodegenerative Disease Brain Bank at King's College London. Informed consent was given from all donors and all the experiments conformed to the principles set out in the WMA Declaration of Helsinki and the Department of Health and Human Services Belmont Report. Frozen frontal cortex and paraffin-embedded spinal cord tissue fixed in 10% formalin were provided by the London Neurodegenerative Diseases Brain Bank (Institute of Psychiatry, King's College London, UK). Table S2 provides summaries for each patient studied.

## Human neuropathology

Tissue was examined by routine neuropathologic diagnostic methods, as described previously (McGurk et al, 2014, 2018b). Briefly, spinal cord tissue was fixed in 10% neutral buffered and

7-μm-thick paraffin sections were prepared onto glass microscope slides. After dewaxing and rehydration endogenous peroxidases were quenched in 30% hydrogen peroxide made up in methanol (30 min) and washed in running tap water (10 min). Slides were incubated in Tris–EDTA buffer pH9 in a pressure cooker using the standard operating procedure for the 2100 Retriever. Slides were cooled in the pressure cooker (at least 1 h), washed in 0.1 M Tris pH 7.6 and blocked in 0.1 M Tris pH 7.6 with 2% FBS. Primary antibodies, in 0.1 M Tris pH 7.6 with 2% FBS, were applied to each slide overnight at 4°C in a humidified chamber. Slides were washed in 0.1 M Tris pH 7.6, blocked in Tris pH 7.6 with 2% FBS, and incubated with biotinylated IgG from rabbit (1 in 1,000 #BA-2000; Vector labs) in a humidified chamber for 1 h at RT. Slides were washed in 0.1 M Tris pH 7.6 then 0.1 M Tris pH 7.6 with 2% FBS and incubated with an avidin-conjugated horseradish peroxidase (#PK-6100; Vectastain ABC kit) made up in Tris pH 7.6 with 2% FBS (1 h at RT). Slides were washed in Tris pH 7.6 and developed with Diaminiobenzidine (DAB) solution (SK-4105; Vector labs) for 8 min at RT. Slides were counterstained with Harris hematoxylin (30 s), washed in running tap water (10 min) dehydrated, cleared in xylene and mounted in cytoseal XYL (#8312–4; Thermo Fisher Scientific). All Tris-based washes were 5 min. Primary antibodies used were rabbit anti-phosphorylated (pS409/410) TDP-43 (1 in 5,000 #80007-I-RR; Proteintech) and rabbit anti-NSUN1 (1:5,000 #10448-1-AP; Proteintech).

### Biochemical fractionation of human postmortem frontal cortex

Biochemical fractionation was performed as described for TDP-43 (Neumann et al, 2006; Laferriere et al, 2019; Arseni et al, 2022). Here, 100 mg of frozen human frontal cortex was finely diced with a sterile scalpel and homogenised in 500 μl low-salt buffer (10 mmol/liter Tris pH 7.5, 5 mmol/liter EDTA, 1 mmol/liter DTT, 10% sucrose) and Halt Protease Inhibitor Cocktail (Thermo Fisher Scientific) using a Dounce homogenizer. The sample was transferred to a 1 ml Open-Top Thickwall 11 × 34 mm polycarbonate tube (Cat # 343778; Beckman) and centrifuged at 25,000g for 30 min at 4°C. The supernatant (the low-salt fraction) was stored at –80°C. The pellet was sonicated in 500 μl Triton-X buffer (10 mmol/liter Tris pH 7.5, 5 mmol/liter EDTA, 1 mmol/liter DTT, 10% sucrose, 0.5 mol/liter NaCl, 1% Triton X-100, and Halt Protease Inhibitor Cocktail) at 50% amplitude, 5 s on and 2 s off for 40 s (or until fully resuspended) using Sonics Ultrasonic Vibra-Cell processor. The sample was centrifuged at 180,000g for 30 min at 4°C. The supernatant (the Triton X-100 fraction) was stored at –80°C. The pellet was sonicated in 500 μl Sarkosyl buffer at 50% amplitude, 5 s on and 2 s off for 40 s or until fully resuspended and incubated at RT for 1 h on a platform wave shaker. The sample was centrifuged at 180,000g for 30 min at RT and the supernatant (sarkosyl fraction) was stored at –80°C. The pellet was sonicated in 100 μl urea buffer (7 mol/liter urea, 2 mol/liter thiourea, 4% CHAPS, 30 mmol/liter, Tris–HCl, pH 8.5) and centrifuged at 25,000g for 30 min at RT. The supernatant (urea fraction) was stored at –80°C. Protein was quantified using Pierce 660 nm Protein Assay with ionic detergent compatibility reagent (cat #: 22663; Pierce) following the Microplate Procedure protocol. All buffers expect the urea buffer were kept on ice.

Samples were prepared for immunoblotting by adding 10 μg protein (except for the urea fraction which used 3 μg of protein) to SDS sample buffer (10 mmol/liter Tris pH 6.8, 1 mmol/liter EDTA, 40 mmol/liter DTT, 1% SDS, and 10% sucrose) in a final volume of 16 μl. Samples were incubated at 95°C for 5 min, chilled on ice for 5 min and centrifuged at 2,300g for 5 min at 4°C. 15 μl of the 16 μl sample was electrophoresed on a 4–12% Bis-Tris gel with NuPAGE MOPS buffer and transferred onto 0.45-micron nitrocellulose by wet transfer (30 V for 75 mins). Blots were blocked in 5% milk in TBST (TBS with 0.05% TWEEN-20) for 1 h at RT and incubated in primary antibody in TBST overnight at 4°C. Membranes were washed 4 × 5 min in TBST at RT, incubated in secondary antibody for 1 h in TBST at RT, and signal detected using ECL Select Western blotting (Cat # RPN2235; Amersham). Antibodies used were rabbit anti-phosphorylated (pS409/410) TDP-43 (#80007-I-RR; Proteintech), rabbit anti-NSUN1 (1:5,000, #10448-1-AP; Proteintech), and rabbit anti GAPDH (1:1,000, #2118T, RRID:AB_561053; Cell Signalling Technology). Secondary use was goat anti-rabbit HRP (1: 10,000, Cat# 115-035-146; EMD Millipore). Blots were quantified with ImageJ and Graphpad prism 9 software was used to calculate statistical significance. Experimental protein levels were quantified relative to the GAPDH loading control, which was imaged on the same membrane as NSUN1.

### Statistical analysis

Statistics were performed using GraphPad prism software (GraphPad software). Significance was set at $P < 0.05$; values for asterisks are in the legends. All graphs presented represent the mean (± SD, or ± sem), unless otherwise stated. n is the number of biological replicates and is indicated in the figure legends. Significance was set at $P < 0.05$.

# Data Availability

Illumina sequencing datasets are available in BioStudies (www.ebi.ac.uk/biostudies), accession: S-BSST2213). Mass spectrometry datasets are available on PRIDE (www.ebi.ac.uk/pride), identifier: PXD070093.

# Supplementary Information

# Acknowledgements

We thank all the people and families who were involved in this study. We thank Prof. Kees Weijer (School of Life Sciences, University of Dundee) for the use of the widefield fluorescent microspore; the Dundee Imaging Facility (School of Life Sciences, University of Dundee) for the use of their paraffin embedder, embedding station, microtome and Leica SP8; the Dundee fingerprints facility (School of Life Sciences, University of Dundee) for mass spectrometry and analysis; and the National Phenotypic Screening Centre for high content imaging of the PLA signal (School of Life Sciences,

University of Dundee). We thank Dr Charlotte Hurst (School of Life Sciences, University of Dundee) for the initial help with PLA experiments, and Daniel Wright and Greg McNeil (Life Science Undergraduate programme, School of Life Sciences, University of Dundee) for quantification of the Q331K Drosophila eye and generation of the Q331K point mutation in TDP-43, pcDNA3.2, respectively. We thank the MRC-PPU Reagents and Services for subcloning pJFRC5-5XUAS-IVS-TDP-43 and the University of Cambridge Department of Genetics Fly Facility for generating the TDP-43 transgenic. This work was funded by grants awarded to LM (University of Dundee start-up grant, Academy of Medical Sciences Springboard Award (SBF005\1080), Alzheimer's Research UK Early Career Bridging Award (ARUK-ECRBF2022A-019), Royal Society Research grant (RG/R2/232203), Alzheimer's Society Dementia Research Leader's Fellowship (596) and Target ALS New Investigator Award (NI-2023-NAI-S4)).

## Author Contributions

M Parra-Torres: conceptualization, data curation, formal analysis, supervision, validation, investigation, visualization, methodology, project administration, and writing—review and editing.
K Dissanayake: formal analysis and investigation.
JA Gray: investigation.
AJ Langlands: formal analysis and investigation.
R Kucuk: validation.
M Gierlinski: formal analysis.
C Troakes: resources.
A King: resources and supervision.
L McGurk: conceptualization, resources, data curation, formal analysis, funding acquisition, investigation, visualization, methodology, project administration, and writing—original draft, review, and editing.

## Conflict of Interest Statement

The authors declare that they have no conflict of interest.

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
