## [Reviewer comments · Life Science Alliance]

Aberrant NSUN1 Activity Connects m5C RNA Modification to TDP-43 Neurotoxicity in ALS/FTD

Melissa Parra-Torres, Kumara Dissanayake, James Gray, Alistair Langlands, Ridvan Kucuk, Marek Gierlinski, Claire Troakes, Andrew King, and Leeanne McGurk

DOI: <https://doi.org/10.26508/lsa.202503297>

Corresponding author(s): Leeanne McGurk, University of Dundee

Review Timeline:

Submission Date:	2025-03-07
Editorial Decision:	2025-04-25
Revision Received:	2025-09-09
Editorial Decision:	2025-09-18
Revision Received:	2025-10-13
Accepted:	2025-10-16

Scientific Editor: Sarita Hebbbar

Transaction Report:

April 24, 2025

Re: Life Science Alliance manuscript #LSA-2025-03297-T

Dr. Leeanne McGurk
University of Dundee
School of Life Sciences
MSI/WTB complex
Dundee DD1 5EH
United Kingdom

Dear Dr. McGurk,

Thank you for submitting your manuscript entitled "NSUN1 methylation of RNA underpins TDP-43 pathogenesis in ALS/FTD" to Life Science Alliance. The manuscript was assessed by three expert reviewers, whose comments are appended to this letter.

Overall, the three reviewers find this work of value to the community. That said, we agree with the reviewers that several significant points in the manuscript must be addressed for publication at LSA. These are:

1. TDP-43 interactome
 - a. Address points from Reviewer 1, points 1-2.
 - b. Corroborate TDP-43 and NSUN1 interaction with PLA (Reviewer 2, results-points 1-2)
2. TDP-43 interaction specifically with NSUN1 in flies:
 - a. address the concern of Reviewer 1 in point 3 on the identity of fly NSUN5/7
 - b. address the concern about off-target effects (Reviewer 3, point 1)
 - c. Validate Knockdown effects of NSUN1 at protein level (Reviewer 1, point 4, Reviewer 2, results-point 3 and Reviewer 3, point 1).
 - d. Validate reduction in RNA (NSUN2, NSUN3, NSUN4, NSUN5, NSUN6, and MT2) for the respective TRIP lines crossed with drivers (Reviewer 1, point 4)
 - e. Immunofluorescence to verify TDP-43 in controls and NSUN1 Knockdowns (Reviewer 2, results-point 4 and Reviewer 3, point 2)
3. NSUN1 in AD
 - a. specific interaction of TDP-43 with isoform 3 under stress conditions (Reviewer 2, results-point 13, Reviewer 3, point 4)

In view of these recommendations, we invite you to submit a revised manuscript addressing the reviewers' comments. When submitting the revision, please include a letter addressing all the reviewers' comments point by point. While a rebuttal must respond to all points in some form, additional data to resolve these points (other than ones indicated above) is not required.

Thank you for this interesting contribution to Life Science Alliance. We are looking forward to receiving your revised manuscript.

Sincerely,

Sarita Hebbar, PhD
Scientific Editor
Life Science Alliance
<http://www.lsjournal.org>

B. MANUSCRIPT ORGANIZATION AND FORMATTING:

Reviewer #1 (Comments to the Authors (Required)):

The manuscript by Parra Torres and colleagues investigates the network of native proteins interacting with TDP-43 in human neuronal-like SH-SY5Y cells, revealing a strong enrichment of pathways associated with RNA modification. Notably, proteins involved in binding to RNA 5-methylcytosine (m5C) and 6-methyladenine (m6A) modifications were among the most enriched. The authors demonstrate that elevated TDP-43 levels in *Drosophila melanogaster* led to increased m5C RNA modifications, a process dependent on the m5C methyltransferase NSUN1. This nucleolar enzyme is shown to be essential for TDP-43-driven neurodegeneration in *Drosophila*.

In human cells, TDP-43 was found to specifically interact with the long isoform of NSUN1, which has been found to remain prevalent in post-mortem frontal cortex tissue from neurodegenerative disease patients, while other isoforms are notably reduced. These findings suggest that TDP-43 influences RNA methylation pathways and that excessive m5C modification may contribute to TDP-43-associated neurodegeneration, highlighting a previously unrecognized role of TDP-43 in regulating RNA methylation.

This study is of clear interest to LSA readers. However, I have a few suggestions and comments below that the authors should consider for improving the manuscript and enhance its suitability for publication.

Major points:

- Unfortunately, Table S1 has not been included, yet it is a crucial component for analyzing the TDP-43 native interactome. The absence of Table S1 significantly impairs the ability to validate findings, replicate results, and derive meaningful insights from the interactome analysis. Providing access to this table is vital for ensuring transparency, reproducibility, and the advancement of research in this field.
- Figure 1B, in its current form, is completely unusable because the resolution is too low, making all component labels unreadable. This lack of clarity severely limits the ability to interpret the data accurately. An alternative, and potentially more effective, approach would be to apply a more stringent confidence score. This could help refine the data presentation, making key interactions clearer, more distinguishable, and more reliable. The full dataset will still be reported in Table S1, when it will be included, ensuring that all information remains accessible despite applying a more stringent confidence score for clarity. Improving these aspects will enhance the figure's overall usefulness in supporting the analysis.

- It is unclear in Figures 2D, 2E, and 2F whether NSUN7 has been confused with NSUN5. NSUN7 is not conserved in *Drosophila*, as also indicated in Figure 2B, and there are not RNAi stocks available at the Bloomington Stock Center for silencing NSUN7. The stock reported in Table S3 ($y[1] \text{ sc}^* v[1] \text{ sev}[21]; P\{y[+t7.7]v[+t1.8]=\text{TRiP.HMS00438}\}\text{attP2/TM3, Sb}[1]\text{Bloomington \# 32440}$) corresponds to the NSUN5 TRIP line rather than NSUN7. This inconsistency is quite confusing for the reader. I recommend clarifying this point to ensure accuracy and avoid potential misinterpretations.
- There is no direct evidence that the TRIP lines used in the rescue experiments shown in Figure 2D effectively silence the corresponding RNA and consequently reduce protein levels. While Figures 2G and S1D clearly demonstrate a significant reduction of NSUN1 RNA in both *gmr-GAL4*> and *daGS-GAL4* flies, supporting that the rescue of TDP-43-dependent retinal neurodegeneration is due to NSUN1 depletion. The same validation is missing for NSUN2, NSUN3, NSUN4, NSUN5, NSUN6, and MT2. To substantiate the claim that this effect is specific to NSUN1, it is essential to provide, in absence of specific antibodies, RNA level measurements for these genes in the respective TRIP lines crossed with the employed drivers.

Minor points:

- General comment: It would be very helpful for readers if the figures were mentioned and described at least once in the main text. For example, figure 1A and 2A are never mentioned or described. Similarly, panel A, B and C of Figure S4 are not described in the text.
- Results, second paragraph: Cytosines in RNA are methylated by the m5C-RNA methyltransferases there are the seven NOL1/NOP2/SUN domain (NSUN) enzymes (NSUN1-7) and the DNA methyltransferase DNMT2a, should be "Cytosines in RNA are methylated by the m5C-RNA methyltransferases there which include seven NOL1/NOP2/SUN domain (NSUN) enzymes (NSUN1-7) and the DNA methyltransferase DNMT2a."
- Results, second paragraph: This revealed that *Drosophila* has six of the eight m5 methyltransferases. The word that is missing.
- Results, fifth paragraph: To define the physical underpinnings that control the TDP-43 and NSUN1 interaction...should be changed in: To define the molecular basis of the TDP-43 and NSUN1 physical interaction...
- Figure 5A lateral labelling is missing.
- Please check carefully whether the figure numbers mentioned in the main text correspond to the correct figures. For example, in the third paragraph, Figure S2 has been swapped with Figure S3. Similarly, in the fourth paragraph, Figure 4D,E should be Figure 4B,C.
- The legend of Figure S4 describes panels A-B-C-E-F, that should be A-B-C-D-E
- Legend Table S2: Scale 0 - no TDP-43 pathology or no motor neuron loss; 1 - mild TDP-43 pathology or no motor neuron loss; 2 - moderate TDP-43 pathology or no motor neuron loss; 3 - severe TDP-43 pathology or no motor neuron loss. I believe that the word "no" should be replaced with "mild," "moderate," and "severe" respectively. If this is incorrect, I had difficulty understanding the rationale behind the numbers in relation to phenotype severity.
- In the abstract: In *Drosophila melanogaster* that recapitulate TDP-43 disease pathology, should be changed in: In a *Drosophila melanogaster* model that

Reviewer #2 (Comments to the Authors (Required)):

In this manuscript, Parra Torres et al highlight an intriguing connection between RNA methylation and TDP-43, an RNA binding protein with fundamental ties to several neurodegenerative diseases. In performing an unbiased screen of proteins that interact with endogenous TDP-43 from a human neuroblastoma cell line (SHSY-5Y), the authors found significant enrichment for factors that are involved in RNA methylation, including 5-methylcytosine (5mC) and 6-methyladenosine (m6A). Follow up studies confirmed RNA-independent interactions between TDP-43 and NSUN1, a 5mC methyltransferase acting on rRNA. Knockdown of the *Drosophila* homologue of NSUN1 prevented TDP-43 mediated phenotypes in fly overexpression models, and the authors detected a subtle shift in NSUN1 isoform expression in post-mortem samples from patients with ALS and FTD, two neurodegenerative conditions featuring TDP-43 pathology.

The manuscript touches on an important topic that could have significant ramifications for our understanding of how TDP-43 affects RNA homeostasis, both in healthy cells and in disease. The initial studies of TDP-43 interactions may initially seem redundant given the many previous studies of the same topic, but the authors focus on endogenous TDP-43, in contrast to many prior investigations that use overexpressed TDP-43. Genetic interactions between TDP-43 and NSUN1 in flies strengthen the manuscript, but still leave many open questions.

Specifically, the nature of the experiments (IP-MS) raises concern that the interaction between TDP-43 and nucleolar factors such as NSUN1 represents an artifact of compartmental mixing. Findings in the *Drosophila* TDP-43 OE model are interesting, but should be validated in loss-of-function models if possible, and the specificity of the m5A phenotype should be explored further. The human data, while exceptionally helpful for tying the observations to disease pathogenesis, are also incomplete. Specific concerns are discussed below.

Abstract:

- some words are missing (ie "but" in the first sentence, and "models" in the third sentence)
- Because of prior work on TDP-43 and m6A, the authors need to be more specific when referencing "RNA methylation"

Introduction:

- 1st paragraph: TDP-43 pathology in AD is actually comorbid LATE. Usually this is referred to as AD with LATE-NC (neuropathological changes)
- 1st paragraph: "Disease-inducing" changes is vague as written
- 3rd paragraph, first sentence: run on, consider splitting in two

Results:

- why is it expected that TDP-43 interacts with nucleolar proteins? Typically, TDP-43 is excluded from the nucleolus. The fact they the authors detect an interaction suggests that this may be an artifact of cell lysis and mixing of distinct compartments. Studies using actinomycin D suggests that NSUN1 and TDP-43 overlap when the nucleolus dissolves, but the relevance of this is unclear. Follow up studies with PLA may be more convincing than simple co-localization.
- PLA could also be used to strengthen the authors argument that TDP-43 interacts with NSUN1 in disease states
- Relatively large effects of NSUN1 KD contrast with mild reduction in NSUN1 abundance by qRT-PCR (Fig 2G). This is also very different than results in Sup Fig 1D, showing very effective NSUN1 KD. Do these effects persist at the protein level?
- The relatively high levels of cytoplasmic TDP-43 in control flies (Fig 3D) is unusual, making the effect of NSUN1 KD difficult to interpret. Can the authors verify this effect using a different method (ie immunofluorescence?)
- Second section, first sentence and last sentence: run on sentences, consider splitting in two or adding more punctuation
- In addition to the over expression models, the authors are encouraged to examine the effects of NSUN1 depletion in a TDP-43 loss of function model
- Second paragraph, first sentence, "TDP-43 promotes NSUN1 methylation of RNA in *Drosophila*" section: run on sentence, consider splitting in two
- What is NSUN1 methylating, if not rRNA?
- If m5A is a stress response, is the regulation of m5A in TDP-43 OE flies (Fig 4B, C) a response to stress? Are similar changes observed in other situations or models?
- There are >1 NSUN1 bands in immunoblots in Fig 5A-E. Please indicate which bands corresponds to which isoform in the figure
- The fact that TDP-43 binds the long NSUN1 isoform and not the short isoform is intriguing. This provides an opportunity to map out the domains required for interaction. The extra 4 kDa in isoform 3 is clearly important, but is it sufficient for TDP-43 binding? What is required in TDP-43 for binding NSUN1, if not RNA recognition?
- What does it mean that isoform 3 (which selectively interacts with TDP-43) is the only one NOT reduced in human ALS/FTD tissue?
- In Fig 6E, the most appropriate comparisons include individual isoforms in controls vs ALS/FTD (ie isoform 3 in controls vs isoform 3 in ALS/FTD), rather than isoform 3 vs other isoforms in controls, and separately in ALS/FTD (as is currently shown)
- The panel title for Fig 6E should be "protein" and not "proein"

Reviewer #3 (Comments to the Authors (Required)):

In this study, Parra Torres et al. investigated the interaction between TDP-43 and the 5-methylcytosine (m5C) RNA methyltransferase NSUN1 and the functional implications of this interaction in the context of amyotrophic lateral sclerosis (ALS) and frontotemporal dementia (FTD). Using co-immunoprecipitation experiments of endogenous TDP-43 from SH-SY5Y cells, the interactome was identified by mass spectroscopy, revealing an enrichment of RNA-based processes and functions. The authors focused on a possible functional link between NSUN1 and TDP-43. Using a *Drosophila* model of TDP-43 neurotoxicity, they show that downregulation of NSUN1, but not other methyltransferases, reduces lesions in the outer eye and retina of flies. In addition, downregulation of NSUN1 in adult neurons partially rescued the lifespan shortening caused by TDP-43 neurotoxicity. Dot blot analysis revealed increased m5C in RNA upon TDP-43 overexpression, whereas bisulfite analysis of the rRNA target cytosine showed no effect. Similarly, rRNA processing was unaffected by TDP-43 overexpression and NSUN1 knockdown, which is surprising given the critical role of NSUN1 in regulating rRNA processing. The authors then switch to human cells and show by co-immunoprecipitation that TDP-43 preferentially interacts with a specific NSUN1 isoform in an RNA-independent manner. From immunofluorescence experiments in the presence or absence of nucleolar stress conditions, the authors conclude that NSUN1 and TDP-43 colocalise during stress. Finally, in an effort to relate these findings to ALS/FTD disease, NSUN1 protein levels were examined in post-mortem tissue from healthy and diseased individuals, and differences in the isoform composition of NSUN1 were found.

Overall, this is a potentially interesting study that suggests the involvement of aberrant NSUN1-dependent RNA methylation in TDP-43 pathology. The study shows some intriguing observations, such as the partial rescue of TDP-43 effects by siNSUN1 and the specific interaction with a particular isoform, but does not go into detail on the molecular mechanisms. Furthermore, some of the interpretations and claims, in particular the link between m5C and TDP-43 accumulation in the cytoplasm, are not clearly supported by the data presented. The following issues need to be addressed prior to publication:

Major issues

1. Although the observed phenotypic rescue of TDP-43 effects by NSUN1 knockdown seems clear from the data shown in Figures 2 and 3, it is puzzling that this strong effect can be achieved by only 20% reduction (Figure 2G) of mRNA. The magnitude of the reduction is not shown in Figure 3. Is there a greater reduction in NSUN1 protein levels? A cursory look at flybase suggests that the dsRNA may also target the *dgt5* gene, which shares the NSUN1 locus. Could there be an off-target effect?
2. Figure 3D, E: Just looking at the western blot, it is not clear that NSUN1 KD significantly affects the cytoplasmic localisation of TDP-43. Please explain how the relative levels of TDP-43 for the nucleus and cytoplasm were calculated. Is it the ratio to laminC or tubulin?
3. The analysis of m5C is also problematic. Firstly, dot blot analysis is a very crude and unreliable method for this, in particular because the available antibodies are suboptimal for use with m5C in RNA (PMID: 32636310). Furthermore, the absence of any m5C signal in the control spot is worrying. There should be a fairly significant signal derived from rRNA and tRNA. Therefore, to support one of the main claims of the manuscript, that TDP-43 toxicity involves aberrant RNA methylation, the authors need to use additional methods such as mass spectrometry. Figure 4D shows a Sanger sequencing result, but the text claims that Illumina sequencing was used - which is correct?
4. The text refers to the different NSUN1 western blot bands as different isoforms, and the schematic in Fig. S4 suggests that they result from alternative splicing. Is there any evidence for this? In other words, could the bands correspond to differently modified proteins rather than spliceforms? Assuming that these bands are indeed from different isoforms, it would have been interesting to see more experiments characterising the specific interaction of TDP-43 with isoform 3.
5. Fig. 6E: It is difficult to really see the stress-dependent co-localisation of TDP-43 and NSUN1. In fact, it looks more like they are mutually exclusive in some of the more prominent foci in the ActD-treated cells. It would be better to show the merge without the Hoechst stain to allow better differentiation.
6. The conclusions drawn from Figure 7 regarding the changes in NSUN1 isoform composition in ALS brains are rather weakly supported by the data shown. First, the number of samples is small, and second, the variability introduced by Western blot detection seems to be quite large (e.g. compare #21, 22) in panels C and D. Also, it is unclear to this reviewer why the statistical tests were performed between the different isoforms of one condition and not between the same isoform in co and ALS patients?
7. Figure 7C,D and Figure S4D: Why is the size of the bands identified as isoform 4 so different in the blots shown in the two figures?

Minor points

1. The rationale for choosing RNA methyltransferases to study is not clear when most enriched GO terms were RNA modification binding proteins.
2. The editing of the manuscript is rather sloppy. There are many grammatical errors in the main text (e.g. missing punctuation) which do not help clarity. The captions are very minimalistic. Please include more experimental details, especially for Figure 3 (how was the relative level calculated in E,F) and Figure 5 (Cells in B are misidentified). The labelling of Western blots is sometimes inadequate, e.g. Fig. 5A, D (several bands are not identified). Mislabelling of *Drosophila* genotypes in the caption of Fig. 3 (NSUN2 is shown instead of NSUN1). In Fig. S4, panels E and F are mislabelled.
3. Graphical summary is not very informative and contains typos (disese, sam).

We would like to thank the Reviewers and Editors for their time and constructive suggestions, which have helped strengthen the conclusions of the manuscript. We have addressed the Editors' and Reviewers' comments by providing additional data, editing the manuscript, and expanding figures. Specifically, we have validated all knockdown strains, further demonstrated that NSUN1 downregulation mitigates TDP-43 cytoplasmic accumulation using immunofluorescence and microscopy, confirmed the direct interaction between TDP-43 and NSUN1 via proximity ligation assays (PLA) showing that this occurs in both the nucleolus and nucleoplasm. We also further expanded the pathological analysis of ALS/FTD postmortem tissue to validate that NSUN1 isoforms 1/2 are selectively reduced in disease. These revisions are described below as we address each Reviewer comment individually. We hope that the Reviewers and Editors agree that these changes have strengthened the manuscript, demonstrating that NSUN1-mediated m⁵C methylation is a critical pathway regulating TDP-43 localization and toxicity in motor neuron disease.

We have uploaded the manuscript with figure legends placed after the Methods section. Figures are uploaded separately for clarity, and we have also provided a combined figure file with legends underneath for ease of review. The numbering of the legends in the document below corresponds to the figure numbers in the manuscript.

Reviewer #1 (Comments to the Authors (Required)):

The manuscript by Parra Torres and colleagues investigates the network of native proteins interacting with TDP-43 in human neuronal-like SH-SY5Y cells, revealing a strong enrichment of pathways associated with RNA modification. Notably, proteins involved in binding to RNA 5-methylcytosine (m⁵C) and 6-methyladenine (m⁶A) modifications were among the most enriched.

The authors demonstrate that elevated TDP-43 levels in *Drosophila melanogaster* led to increased m⁵C RNA modifications, a process dependent on the m⁵C methyltransferase NSUN1. This nucleolar enzyme is shown to be essential for TDP-43-driven neurodegeneration in *Drosophila*.

In human cells, TDP-43 was found to specifically interact with the long isoform of NSUN1, which has been found to remain prevalent in post-mortem frontal cortex tissue from neurodegenerative disease patients, while other isoforms are notably reduced. These findings suggest that TDP-43 influences RNA methylation pathways and that excessive m⁵C modification may contribute to TDP-43-associated neurodegeneration, highlighting a previously unrecognized role of TDP-43 in regulating RNA methylation.

This study is of clear interest to LSA readers. However, I have a few suggestions and comments below that the authors should consider for improving the manuscript and enhance its suitability for publication.

Major points:

- Unfortunately, Table S1 has not been included, yet it is a crucial component for analyzing the TDP-43 native interactome. The absence of Table S1 significantly impairs the ability to validate findings, replicate results, and derive meaningful insights from the interactome analysis. Providing access to this table is vital for ensuring transparency, reproducibility, and the advancement of research in this field.

We apologise for this oversight. Table S1 will be uploaded as supplemental data, and in addition includes the PANTHER molecular function analysis.

Updated text:

Results, line 181: Gene ontology (GO) analysis of molecular function revealed that proteins that bind to methylated cytosine (5-methylcytosine/ m⁵C) in RNA was the enriched pathway (Figure 1C and Table S1).

- Figure 1B, in its current form, is completely unusable because the resolution is too low, making all component labels unreadable. This lack of clarity severely limits the ability to interpret the data accurately. An alternative, and potentially more effective, approach would be to apply a more stringent confidence score. This could help refine the data presentation, making key interactions clearer, more distinguishable, and more reliable. The full dataset will still be reported in Table S1, when it will be included, ensuring that all information remains accessible despite applying a more stringent confidence score for clarity. Improving these aspects will enhance the figure's overall usefulness in supporting the analysis.

We apologise for the lack of clarity in image 1B, the image is now high resolution, and all names are visible upon magnification. Figures will also be uploaded individually as well as in the word document for better resolution. The network was produced using the most stringent confidence score of > 0.9 and is based only on text mining and evidence, so we cannot reduce the complexity by increasing confidence. We are using the image to highlight the main biological areas of the TDP-43 interactome: RNA metabolism, translational control, DNA maintenance expression and cellular trafficking. To support reader examination of the interactome, we have presented each cluster individually in the supplemental data (Fig S1-S5). We have clarified stringent confidence details in the main text, and this is shown below. In addition, we repeated the panther analysis for GO molecular function on the most recent database, this still shows the same enrichment value for C5-methylcytidine RNA reader activity. The order has changed to reflect FDR score, this places C5-methylcytidine RNA reader activity as the top hit. To support this, we have added three data to Table S1 with enrichment scores as well as the FDR. The graph in panel 1C has changed to reflect this.

Updated text:

Results, line 175: STRING interaction analysis was used to generate a high-confidence interaction network (interaction score >0.9), which revealed four main clusters enriched for proteins involved in ribosome biogenesis, RNA splicing/transport/degradation, DNA replication and repair, and protein transport (Figure 1B and S1-4).

- It is unclear in Figures 2D, 2E, and 2F whether NSUN7 has been confused with NSUN5. NSUN7 is not conserved in *Drosophila*, as also indicated in Figure 2B, and there are not RNAi stocks available at the Bloomington Stock Center for silencing NSUN7. The stock reported in Table S3 ($y[1] sc[*] v[1] sev[21]; P\{y[+t7.7]v[+t1.8]=TRiP.HMS00438\}attP2/TM3, Sb[1]Bloomington \# 32440$) corresponds to the NSUN5 TRIP line rather than NSUN7. This inconsistency is quite confusing for the reader. I recommend clarifying this point to ensure accuracy and avoid potential misinterpretations.

We apologise for this oversight this has been amended in all figures, legends and main text.

- There is no direct evidence that the TRIP lines used in the rescue experiments shown in Figure 2D effectively silence the corresponding RNA and consequently reduce protein levels. While Figures 2G and S1D clearly demonstrate a significant reduction of NSUN1 RNA in both *gmr-GAL4* and *daGS-GAL4* flies, supporting that the rescue of TDP-43-dependent retinal neurodegeneration is due to

NSUN1 depletion. The same validation is missing for NSUN2, NSUN3, NSUN4, NSUN5, NSUN6, and MT2. To substantiate the claim that this effect is specific to NSUN1, it is essential to provide, in absence of specific antibodies, RNA level measurements for these genes in the respective TRIP lines crossed with the employed drivers.

We thank the reviewer for this suggestion. We measured the expression of the corresponding genes upon global knock down (with Daughterless GAL4). These data demonstrate that with the siRNAs for NSUN2, 5, 6, and Mt2, as well as the NSUN4 loss-of-function allele there is significant reduction of the target genes. Data is presented in Figure 2 beside the phylogram.

Updated text:

Results, line: 202: We obtained Drosophila siRNA strains for Nsun1, Nsun2, Nsun5, Nsun6 and Mt2 and a loss of function strain for NSUN4. To confirm that the silencing strategy for each gene was effective the siRNAs were expressed ubiquitously with either daughterless-geneswitch (daGS)-GAL4 (NSUN1) or daughterless (da)-GAL4 (Nsun2, Nsun4, Nsun5, Nsun6, and Mt2). This analysis showed that all strains to the m⁵C-RNA methyltransferases significantly reduced the expression of their respective m⁵C-RNA methyltransferase targets (Figure 2D-E).

Minor points:

- General comment: It would be very helpful for readers if the figures were mentioned and described at least once in the main text. For example, figure 1A and 2A are never mentioned or described. Similarly, panel A, B and C of Figure S4 are not described in the text.

We have gone through the manuscript and ensured all panels are mentioned.

- Results, second paragraph: Cytosines in RNA are methylated by the m⁵C-RNA methyltransferases there are the seven NOL1/NOP2/SUN domain (NSUN) enzymes (NSUN1-7) and the DNA methyltransferase DNMT2a, should be "Cytosines in RNA are methylated by the m⁵C-RNA methyltransferases there which include seven NOL1/NOP2/SUN domain (NSUN) enzymes (NSUN1-7) and the DNA methyltransferase DNMT2a."

Corrected.

- Results, second paragraph: This revealed that Drosophila has six of the eight m⁵ methyltransferases. The word that is missing.

Corrected.

- Results, fifth paragraph: To define the physical underpinnings that control the TDP-43 and NSUN1 interaction...should be changed in: To define the molecular basis of the TDP-43 and NSUN1 physical interaction...

Corrected.

- Figure 5A lateral labelling is missing.

Corrected.

- Please check carefully whether the figure numbers mentioned in the main text correspond to the correct figures. For example, in the third paragraph, Figure S2 has been swapped with Figure S3. Similarly, in the fourth paragraph, Figure 4D,E should be Figure 4B,C.

Corrected.

- The legend of Figure S4 describes panels A-B-C-E-F, that should be A-B-C-D-E
Corrected.

- Legend Table S2: Scale 0 - no TDP-43 pathology or no motor neuron loss; 1 - mild TDP-43 pathology or no motor neuron loss; 2 - moderate TDP-43 pathology or no motor neuron loss; 3 - severe TDP-43 pathology or no motor neuron loss. I believe that the word "no" should be replaced with "mild," "moderate," and "severe" respectively. If this is incorrect, I had difficulty understanding the rationale behind the numbers in relation to phenotype severity.

In the sections from two post mortem tissue samples there were motor neurons that had no phosphorylated TDP-43 aggregates, these would be classified as 0. However they instead had non-neuronal TDP-43 pathology. Thus these samples should be 0 for MN pathology and then 1 or more for non-neuronal TDP-43 pathology. We have amended the table to clearly state non-neuronal TDP43 pathology. We agree with the reviewer that the motor neuron loss should not be 0 – it should be mild. This has been edited in the Table S3 and associated legend have added the following text to the legend of the Table S3:

Updated text:

File S2-S5, Table S3, line 37: TDP-43 pathology: 0 = no TDP-43 aggregates; 1= mild TDP-43 aggregates; 2= moderate TDP-43 pathology; 3=severe TDP-43 pathology .
Motor neuron loss: 1= mild motor neuron loss; 2=moderate motor neurons loss; 3= severe motor neuron loss.

- In the abstract: In *Drosophila melanogaster* that recapitulate TDP-43 disease pathology, should be changed in: In a *Drosophila melanogaster* model that

Corrected.

Reviewer #2 (Comments to the Authors (Required)):

In this manuscript, Parra Torres et al highlight an intriguing connection between RNA methylation and TDP-43, an RNA binding protein with fundamental ties to several neurodegenerative diseases. In performing an unbiased screen of proteins that interact with endogenous TDP-43 from a human neuroblastoma cell line (SHSY-5Y), the authors found significant enrichment for factors that are involved in RNA methylation, including 5-methylcytosine (5mC) and 6-methyladenosine (m6A). Follow up studies confirmed RNA-independent interactions between TDP-43 and NSUN1, a 5mC methyltransferase acting on rRNA. Knockdown of the *Drosophila* homologue of NSUN1 prevented TDP-43 mediated phenotypes in fly overexpression models, and the authors detected a subtle shift in NSUN1

isoform expression in post-mortem samples from patients with ALS and FTD, two neurodegenerative conditions featuring TDP-43 pathology.

The manuscript touches on an important topic that could have significant ramifications for our understanding of how TDP-43 affects RNA homeostasis, both in healthy cells and in disease. The initial studies of TDP-43 interactions may initially seem redundant given the many previous studies of the same topic, but the authors focus on endogenous TDP-43, in contrast to many prior investigations that use overexpressed TDP-43. Genetic interactions between TDP-43 and NSUN1 in flies strengthen the manuscript, but still leave many open questions.

Specifically, the nature of the experiments (IP-MS) raises concern that the interaction between TDP-43 and nucleolar factors such as NSUN1 represents an artifact of compartmental mixing. Findings in the *Drosophila* TDP-43 OE model are interesting, but should be validated in loss-of-function models if possible, and the specificity of the m5A phenotype should be explored further. The human data, while exceptionally helpful for tying the observations to disease pathogenesis, are also incomplete. Specific concerns are discussed below.

Abstract:

- some words are missing (ie "but" in the first sentence, and "models" in the third sentence)
- Because of prior work on TDP-43 and m6A, the authors need to be more specific when referencing "RNA methylation"

We have corrected these errors and ensured that RNA methylation is described appropriately throughout the text.

Introduction:

- 1st paragraph: TDP-43 pathology in AD is actually comorbid LATE. Usually this is referred to as AD with LATE-NC (neuropathological changes)

We have amended this sentence.

- 1st paragraph: "Disease-inducing" changes is vague as written

Updated text:

Intro, line 118: Aggregation is accompanied by nuclear depletion and loss of TDP-43 function, leading to widespread RNA processing defects such as aberrant cryptic exon inclusion. These alterations disrupt reading frames and protein expression, driving neurotoxicity in disease (Brown *et al.*, 2022; Klim *et al.*, 2019; Ling *et al.*, 2015; Ma *et al.*, 2022).

- 3rd paragraph, first sentence: run on, consider splitting in two

Updated text:

Intro, line 142: There are over one hundred types of RNA modifications, deposited by writer proteins, interpreted by readers, and removed by erasers (Delaunay *et al.*, 2024; Zaccara *et al.*, 2019). Among the most abundant are N6-methyladenosine (m⁶A) and N1-methyladenosine (m¹A), both of which enhance TDP-43 binding, with m1A additionally driving cytoplasmic accumulation (Dominissini *et al.*, 2016; Li *et al.*, 2016; McMillan *et al.*, 2023; Nguyen *et al.*, 2023; Sun *et al.*, 2023; Wei *et al.*, 1976; Wei & Moss, 1977).

Results:

- why is it expected that TDP-43 interacts with nucleolar proteins? Typically, TDP-43 is excluded from the nucleolus. The fact they the authors detect an interaction suggests that this may be an artifact of cell lysis and mixing of distinct compartments. Studies using actinomycin D suggests that NSUN1 and TDP-43 overlap when the nucleolus dissolves, but the relevance of this is unclear. Follow up studies with PLA may be more convincing than simple co-localization.

We thank the Reviewer for this suggestion. While TDP-43 is largely excluded from the nucleolus under basal conditions and with standard immunofluorescent approaches, our proximity ligation assay (PLA) data demonstrate that a subset of TDP-43 and NSUN1 molecules form a nuclear protein complex with ~60–75% of PLA punctae co-localizing with the nucleolus under normal conditions (Figures 7A–D). Importantly, PLA detects protein–protein interactions at distances <40 nm, indicating close proximity rather than mere co-localization. Upon actinomycin D–induced nucleolar stress, PLA shows increased TDP-43–NSUN1 PLA punctae number and intensity. These findings provide in situ evidence that the TDP-43–NSUN1 interaction occurs in intact cells and is not an artifact of lysis or compartment mixing. Therefore, the PLA data strongly support a biologically relevant interaction between TDP-43 and NSUN1, which can be modulated by nucleolar stress. These data are in Figure 7 and Figure S8, with the original IFC data in Figure S7. We also added detail on TDP-43 in the nucleolus from published studies in the discussion.

Updated text:

Results, line 364: To test this, we employed proximity ligation assay (PLA), which detects protein–protein interactions within <40 nm. Under basal conditions, TDP-43 and NSUN1 formed a nuclear protein complex, but not in the cytoplasm, compared to negative controls (Figure 7A–C, Figures S8A–B). Actinomycin D treatment increased both the number and intensity of TDP-43–NSUN1 PLA foci, without affecting their size (Figure 7B). To determine the subnuclear localization of these PLA foci, cells were counterstained with the nucleolar marker Nucleolar Green. Approximately 75% of TDP-43–NSUN1 PLA puncta co-labelled with the nucleolus under normal conditions, and this remained unaltered by actinomycin D treatment (Figure 7D–E), indicating that TDP-43 and NSUN1 function in both the nucleoplasm and nucleolus under normal and nucleolar stress conditions. By comparison, the co-immunoprecipitation between NSUN1 and TDP-43 remained largely unchanged following actinomycin D treatment, which may reflect either the detection limits of the co-IP assay or that the PLA-detected increase in foci number and intensity represents closer proximity of the proteins rather than an increase in overall binding. Together, these data suggest that TDP-43 and NSUN1 form a nuclear protein complex under basal conditions, and that this interaction is enhanced upon inhibition of RNA transcription and nucleolar breakdown.

Discussion, line 455: Previous transcriptomic studies show that TDP-43 and NSUN1 target largely divergent RNAs: TDP-43 binds mostly mRNAs with less than 2% rRNA, whereas NSUN1 binds almost exclusively rRNA (98.32%) and very little mRNA (0.01%) (Liao *et al.*, 2022; Polymenidou *et al.*, 2011; Tollervey *et al.*, 2011). This is consistent with our overall immunofluorescence observations in human neuronal-like cells, where TDP-43 and NSUN1 occupy mostly separate nuclear compartments. However, our proximity ligation assay (PLA) data reveal that under basal conditions, TDP-43 and NSUN1 form a nuclear protein complex, with ~75% of interactions localizing to the nucleolus. These PLA results indicate that even when largely separated, a subset of TDP-43 and NSUN1 molecules are in proximity,

potentially poised for functional interaction. Furthermore, upon nucleolar stress induced by actinomycin D, which inhibits RNA transcription, NSUN1 redistributes to the nucleoplasm. Concurrently, the number and intensity of TDP-43–NSUN1 complexes increase, and these complexes are present in both the nucleolus and nucleoplasm, suggesting that TDP-43 and NSUN1 may function together in RNA regulation when transcription is inhibited.

Discussion, line 468: Finally, there are examples of potential converging pathways for TDP-43 and NSUN1. For instance, In TDP-43 phosphorylated at threonine 153 (pT153) and tyrosine 155 (pY155) localizes to the nucleolus upon heat shock, and is detectable only with a T153/pY155-specific antibody (Li *et al*, 2017). Additionally, despite small RNAs being a lower-abundance target compared to their main RNA targets, both proteins bind them: NSUN1 associates with box C/D small nucleolar RNAs (snoRNAs), and TDP-43 binds small Cajal RNAs (scaRNAs), which promotes 2'-O-methylation of U1 and U2 small nuclear RNAs to regulate splicing (Izumikawa *et al*, 2019; Liao *et al.*, 2022). This raises the intriguing possibility that TDP-43 may direct NSUN1 methylation via small nuclear or nucleolar RNAs. NSUN1 can also function in the nucleoplasm to methylate mRNAs in contexts such as kidney and ovarian cancer (Tian *et al*, 2024; Yang *et al*, 2023). Together, these observations hint that the noncanonical m5C methylation we detect in *Drosophila* may reflect aspects of a conserved pathway that underlies shared TDP-43–NSUN1 interactions, offering a subtle window into potential functional convergence without specifying the exact targets.

Methods, line 789:

Proximity Ligation Assay and High Content Image Acquisition and Quantification

20,000 U2OS cells were seeded onto a 96-well plate, the following day cells were treated with either DMSO or 1 ug/ml actinomycin D for the indicated time. Cells were fixed in 4% paraformaldehyde for 15 min, permeabilized three times in TBS with 0.1% triton X100, for 3 min for each. Cells were blocked and processed using the proximity ligation assay from Navinci (NaveniFlex Cell Red #60025). Antibodies used were rabbit anti-NSUN1 (1:2,000; Proteintech #10448-1-AP) and mouse TDP-43 (1 in 500, Proteintech, # 60019-2-IG). Cells were counterstained with 0.5–5 µmol/l Nucleolus Bright Green (Insight Biotechnology, N511-10) for 5 min at room temp. Controls were used as advised for the NaveniFlex Cell Red, which remove one antibody form the incubation. Cells were imaged on an Operetta CLS1601 (Revvity): imaging was performed using a 20x objective (0.4NA) with 2x binning. The acquisition settings used for each fluorophore were: Hoechst 10% power, 20ms exposure with 355-385 Excitation filter (Ex) and 430-500 Emission filter (Em); AlexaFluor488 50% power, 50ms exposure with 460-490 Ex and 500-550 Em; and AlexaFluor594 50% power, 300ms exposure with 560-575 Ex and 570-650 Em. 12 fields of view were acquired per well of a 96 well plate. Image analysis was performed using CellProfiler (4.2.80002). Eight non-overlapping images at 20X were taken for each well and used for quantification, ~2000 cells were quantified per well. An individual experiment contained all conditions in duplicate or triplicate and were performed on separate weeks. An Otsu thresholding method was used to identify all objects: nuclei were identified by the Hoechst image; nucleoli on a processed (enhanced and edged) AlexaFluor488 image; PLA puncta on a processed (enhanced) AlexaFluor594 image. Puncta with a Maximum intensity below an experimentally determined value were discarded. Retained puncta were related to both nuclei and nucleoli to determine whether they were located inside the larger objects. Measurements were made by CellProfiler using the MeasureObjectSizeShape, MeasureObjectIntensity, and MeasureColocalization modules. Image intensity and threshold values (Final Threshold, Sum of Entropies, and Weighted Variance) determined by CellProfiler were used to identify and discard images of poor quality.

- PLA could also be used to strengthen the authors argument that TDP-43 interacts with NSUN1 in disease states

This a great a suggestion and although out of scope for this manuscript we will consider PLA in postmortem tissue in the future.

- Relatively large effects of NSUN1 KD contrast with mild reduction in NSUN1 abundance by qRT-PCR (Fig 2G). This is also very different than results in Sup Fig 1D, showing very effective NSUN1 KD. Do these effects persist at the protein level?

Unfortunately, there are no antibodies to *Drosophila* NSUN1, or any of the other methyl transferases so we are lited to mRNA assessments. We would like to highlight our approaches in qPCR. We use global expression of the siRNA to gain information on how well the siRNA functions. This is because in assessing transcript levels in whole heads when the siRNA is only expressed in the eye will underestimate the degree of eye-specific silencing because signals from other tissues, such as brain and exoskeleton, can dilute the measurement. We have highlighted this in the main text.

Updated text:

Results, line 227: Notably, expression of *Nsun1* was significantly reduced by siRNA-mediated knockdown in the eye (Figure 3D), although measuring transcript levels in whole heads will underestimate the extent of eye-specific silencing due to signal dilution from other tissues (e.g., brain and exoskeleton).

- The relatively high levels of cytoplasmic TDP-43 in control flies (Fig 3D) is unusual, making the effect of NSUN1 KD difficult to interpret. Can the authors verify this effect using a different method (ie immunofluorescence?)

The presence of TDP-43 in the cytoplasmic fraction is expected, as these flies express human TDP-43, and protein levels were assessed using an antibody specific to human TDP-43. By day 14, TDP-43 expression leads to cytoplasmic accumulation, as shown in the figure. We have updated the Results section to clarify this. Additionally, using immunofluorescence, we confirmed that inducible TDP-43 expression in the nervous system leads to TDP-43 aggregate formation and this is reduced upon *Nsun1* silencing, supporting our immunoblotting data. These data gave been added to Figure 4.

Updated text:

Results, line 260: To assess how *Nsun1* downregulation affects TDP-43 accumulation in adult neurons, we measured TDP-43 protein levels in nuclear, cytosolic, and insoluble fractions from *Drosophila* head tissue. As expected, expression of TDP-43 with *elav-GS-GAL4* resulted in detectable protein in all three compartments (Figure 4D-E).

Results, line 269: Consistent with these biochemical findings, microscopic analysis of the adult brain revealed large cytoplasmic TDP-43 puncta in neurons expressing TDP-43 with *elav-GS-GAL4* (Figure 4F-G). Strikingly, *Nsun1* downregulation significantly reduced both the number and size of TDP-43 cytoplasmic puncta (Figure 4G). Collectively, these results

demonstrate that Nsun1 downregulation in adult neurons promotes nuclear retention of TDP-43, reduces cytoplasmic accumulation, and alleviates TDP-43–induced lifespan deficits.

- Second section, first sentence and last sentence: run on sentences, consider splitting in two or adding more punctuation

Corrected.

- In addition to the over expression models, the authors are encouraged to examine the effects of NSUN1 depletion in a TDP-43 loss of function model

We agree with the Reviewer that loss of function effects is the next stage for understanding the full effects of TDP-43 disease mechanisms. While this is out the scope of this initial study, it will be taken forward to future work.

- Second paragraph, first sentence, "TDP-43 promotes NSUN1 methylation of RNA in Drosophila" section: run on sentence, consider splitting in two

Corrected.

- What is NSUN1 methylating, if not rRNA?

We think this is a great question. There is evidence to suggest that it could be small regulatory RNAs, but it could be mRNAs as NSUN1 has been shown in the context of cancer methylate mRNA. Perhaps it is other rRNA sites. We touch on this in the discussion.

Updated text:

Discussion, line 473; Additionally, despite small RNAs being a lower-abundance target compared to their main RNA targets, both proteins bind them: NSUN1 associates with box C/D small nucleolar RNAs (snoRNAs), and TDP-43 binds small Cajal RNAs (scaRNAs), which promotes 2'-O-methylation of U1 and U2 small nuclear RNAs to regulate splicing (Izumikawa *et al*, 2019; Liao *et al.*, 2022). This raises the intriguing possibility that TDP-43 may direct NSUN1 methylation via small nuclear or nucleolar RNAs. NSUN1 can also function in the nucleoplasm to methylate mRNAs in contexts such as kidney and ovarian cancer (Tian *et al*, 2024; Yang *et al*, 2023). Together, these observations hint that the noncanonical m⁵C methylation we detect in Drosophila may reflect aspects of a conserved pathway that underlies shared TDP-43–NSUN1 interactions.

- If m5A is a stress response, is the regulation of m5A in TDP-43 OE flies (Fig 4B, C) a response to stress? Are similar changes observed in other situations or models?

It is possible that TDP-43 overexpression triggers NSUN1 activation as part of a stress response, and we discuss this possibility in the Discussion. However, we do not believe this represents a general response to neurodegenerative toxicity, because NSUN1 downregulation does not suppress toxicity in other neurodegenerative disease models (Figure S6).

Updated text

Discussion, line 435: RNA m⁵C methylation is an adaptive cellular response to stressors such as oxidative stress, heat shock, and nutrient deprivation (Aguilo *et al*, 2016; Blanco *et al*, 2014; Chan *et al*, 2010; Chan *et al*, 2012; Heissenberger *et al*, 2020; Schaefer *et al*, 2010; Schosserer *et al*, 2015).

Discussion, line 443: Consistent with this, we detect little to no m⁵C in total RNA from control *Drosophila* head tissue, whereas TDP-43 expression significantly upregulates m⁵C in an NSUN1-dependent manner.

Discussion Line 448: These data suggest that pathological triggers, such as TDP-43 overexpression, can activate NSUN1 to methylate noncanonical cytosines in RNA. Thus, we suggest that under conditions when RNA m⁵C levels are low, specific triggers, such as TDP-43-induced degeneration, activates NSUN1 causing an increase in methylation at non-canonical, and still to be discovered, cytosines and target RNAs.

- There are >1 NSUN1 bands in immunoblots in Fig 5A-E. Please indicate which bands corresponds to which isoform in the figure

Corrected.

- The fact that TDP-43 binds the long NSUN1 isoform and not the short isoform is intriguing. This provides an opportunity to map out the domains required for interaction. The extra 4 kDa in isoform 3 is clearly important, but is it sufficient for TDP-43 binding? What is required in TDP-43 for binding NSUN1, if not RNA recognition?

We think these are these are interesting questions that will tease a part how TDP-43 and NSUN1 function in future studies.

- What does it mean that isoform 3 (which selectively interacts with TDP-43) is the only one NOT reduced in human ALS/FTD tissue?

The persistence of NSUN1 isoform 3 in ALS/FTD tissue is particularly noteworthy because it is the isoform that selectively interacts with TDP-43. While the shorter isoforms (1/2) are reduced, the continued presence of isoform 3 suggests that a pool of NSUN1 capable of contributing to pathological TDP-43 interactions remains in disease.

Updated text:

Abstract, line 59: In ALS/FTD post-mortem frontal cortex, NSUN1 isoform 3 persists, while the shorter isoform is reduced, suggesting that a pool of NSUN1 capable of contributing to pathological TDP-43 interactions remains in disease.

Results, line 414; Collectively, these data indicate that while NSUN1 does not co-aggregate with TDP-43, its isoform composition and potentially functional balance are altered in ALS/FTD postmortem tissue.

Discussion, line 427: TDP-43 physically and selectively interacts with the longer NSUN1 isoform (isoform 3) independently of RNA in human neuronal-like cells. In ALS/FTD postmortem frontal cortex, the stoichiometry of NSUN1 isoforms is significantly altered, with the shorter isoforms downregulated and the longer isoform 3 persisting. The persistence of isoform 3, which selectively interacts with TDP-43, suggests that a pool of NSUN1 capable of contributing to pathological interactions remains in disease

- In Fig 6E, the most appropriate comparisons include individual isoforms in controls vs ALS/FTD (ie

isoform 3 in controls vs isoform 3 in ALS/FTD), rather than isoform 3 vs other isoforms in controls, and separately in ALS/FTD (as is currently shown).

We have presented the data relative to the GAPDH control (Figure 8 D-G). Additionally, we show the pathological findings for each patient by western blot, including one patient who lacks TDP-43 pathology (Figure S9). This patient was excluded from the main data analysis (Figure 8D–G) but is included in the analysis presented in Figure S10.

Updated text:

Results, line 392: Next, we analysed NSUN1 solubility in postmortem frontal cortex from controls and ALS/FTD patients. Proteins were sequentially extracted in buffers of increasing denaturing strength and were first immunoblotted for TDP-43 (Figure S9). As expected, the control cohort were negative for pathological forms of TDP-43 -the C25 fragment and high molecular weight smear (Neumann *et al.*, 2006) (Figure S9). In our ALS/FTD cohort, four out of five samples were positive for TDP-43 pathology (Figure S9). Patient #16 was negative for frontal cortex TDP-43 pathology by western blot; this patient was diagnosed with frontotemporal lobar degeneration (FTLD)-TDP type C, associated with the semantic variant of primary progressive aphasia (svPPA) (Borghesani *et al.*, 2020). Unlike other FTLD subtypes, atrophy in type C begins in the anterior temporal lobe rather than the frontal cortex, with TDP-43 aggregates appearing as compact neuronal cytoplasmic inclusions in the temporal lobe versus dystrophic inclusions in the frontal cortex (Neumann *et al.*, 2021). It is possible that in patient 13, disease progression from the anterior temporal lobe to the frontal cortex was slow, explaining the absence of TDP-43 pathology in the frontal cortex.

Results, Line 407: Immunoblotting of the low-salt fractions revealed the three NSUN1 isoforms: upper (isoform 3), middle (isoforms 1/2), and lower (isoform 4). Quantification of NSUN1 levels in ALS/FTD with confirmed TDP-43 pathology showed that NSUN1 isoforms 1/2 were significantly reduced (Figure 8D–G), while isoform 3—the NSUN1 isoform that preferentially interacts with TDP-43 (Figure 6)—and isoform 4 were unaffected (Figure 8D–G). Analysis including Patient #16 is shown in Figure S10.

- The panel title for Fig 6E should be "protein" and not "proein"
Corrected.

Reviewer #3 (Comments to the Authors (Required)):

In this study, Parra Torres et al. investigated the interaction between TDP-43 and the 5-methylcytosine (m5C) RNA methyltransferase NSUN1 and the functional implications of this interaction in the context of amyotrophic lateral sclerosis (ALS) and frontotemporal dementia (FTD). Using co-immunoprecipitation experiments of endogenous TDP-43 from SH-SY5Y cells, the interactome was identified by mass spectroscopy, revealing an enrichment of RNA-based processes and functions. The authors focused on a possible functional link between NSUN1 and TDP-43. Using a *Drosophila* model of TDP-43 neurotoxicity, they show that downregulation of NSUN1, but not other methyltransferases, reduces lesions in the outer eye and retina of flies. In addition, downregulation of NSUN1 in adult neurons partially rescued the lifespan shortening caused by TDP-43 neurotoxicity. Dot blot analysis revealed increased m5C in RNA upon TDP-43 overexpression, whereas bisulfite analysis of the rRNA target cytosine showed no effect. Similarly, rRNA processing was unaffected by TDP-43 overexpression and NSUN1 knockdown, which is surprising given the critical role of NSUN1 in regulating rRNA processing. The authors then switch to human cells and show by co-immunoprecipitation that TDP-43 preferentially interacts with a specific NSUN1 isoform in an RNA-

independent manner. From immunofluorescence experiments in the presence or absence of nucleolar stress conditions, the authors conclude that NSUN1 and TDP-43 colocalise during stress. Finally, in an effort to relate these findings to ALS/FTD disease, NSUN1 protein levels were examined in post-mortem tissue from healthy and diseased individuals, and differences in the isoform composition of NSUN1 were found.

Overall, this is a potentially interesting study that suggests the involvement of aberrant NSUN1-dependent RNA methylation in TDP-43 pathology. The study shows some intriguing observations, such as the partial rescue of TDP-43 effects by siNSUN1 and the specific interaction with a particular isoform, but does not go into detail on the molecular mechanisms. Furthermore, some of the interpretations and claims, in particular the link between m5C and TDP-43 accumulation in the cytoplasm, are not clearly supported by the data presented. The following issues need to be addressed prior to publication:

Major issues

1. Although the observed phenotypic rescue of TDP-43 effects by NSUN1 knockdown seems clear from the data shown in Figures 2 and 3, it is puzzling that this strong effect can be achieved by only 20% reduction (Figure 2G) of mRNA. The magnitude of the reduction is not shown in Figure 3. Is there a greater reduction in NSUN1 protein levels? A cursory look at flybase suggests that the dsRNA may also target the *dgt5* gene, which shares the NSUN1 locus. Could there be an off-target effect?

Assessing transcript levels in whole heads when the siRNA is only expressed in the eye will underestimate the degree of eye-specific silencing because signals from other tissues, such as brain and exoskeleton, where the siRNA is not expressed dilutes the measurement. Nevertheless, we observed a significant reduction in NSUN1 expression upon siRNA induction in the eye (Figure 3D). We have now emphasised this in the main text. The *dgt5* gene is on the opposite strand to *Nsun1*, and *Nsun1* is in the first intron of *Dgt5*, making it impossible for the siRNA directly work on *Dgt5*. We have however, measured *Dgt5* levels to ensure that *Nsun1* has no regulatory effect on the *Dgt5* gene. This has shown that *Dgt5* is unaffected by si.*Nsun1* (Figure S6A-B).

Updated text:

Results, line 227: Notably, expression of *Nsun1* was significantly reduced by siRNA-mediated knockdown in the eye (Figure 3D), although measuring transcript levels in whole heads will c the extent of eye-specific silencing due to signal dilution from other tissues (e.g., brain and exoskeleton).

Results line 209: To exclude potential off-target effects of the siRNA to *Nsun1* on *Dgt5*, we measured the *Dgt5* mRNA levels upon NSUN1 silencing (Figure S6A-B). This confirmed that the siRNA was specific to *Nsun1* and did not alter *Dgt5* expression.

2. Figure 3D, E: Just looking at the western blot, it is not clear that NSUN1 KD significantly affects the cytoplasmic localisation of TDP-43. Please explain how the relative levels of TDP-43 for the nucleus and cytoplasm were calculated. Is it the ratio to laminC or tubulin?

For nuclear TDP-43 we made protein levels relative to Lamin C, and for cytoplasmic TDP-43 the protein is relative to tubulin. We have removed the ratio value as we think this is leading to confusion. We have additionally added the insoluble fraction to this blot. These data have bene made clear in the main text, legend and methods. Additionally, using

immunofluorescence, we confirmed that inducible TDP-43 expression in the nervous system leads to TDP-43 aggregate formation, and this is reduced upon Nsun1 silencing, supporting our immunoblotting data. These data have been added to Figure 4.

Updated text:

Results, line 260: To assess how Nsun1 downregulation affects TDP-43 accumulation in adult neurons, we measured TDP-43 protein levels in nuclear, cytosolic, and insoluble fractions from *Drosophila* head tissue. As expected, expression of TDP-43 with elav-GS-GAL4 resulted in detectable protein in all three compartments (Figure 4D-E). Concomitant Nsun1 silencing, however, increased nuclear TDP-43 levels (0.2 ± 0.08 (SD) vs 0.6 ± 0.1 (SD), control vs si.Nsun1, respectively), decreased cytoplasmic TDP-43 (0.82 ± 0.1 (SD) vs 0.45 ± 0.03 (SD), control vs siNSUN1, respectively) and had no effect on the relative levels of insoluble TDP-43 (0.60 ± 0.57 (SD) vs 0.53 ± 0.19 (SD), control vs siNSUN1, respectively) (Figure 4D-E). For the insoluble fraction, Lamin C was used as a loading control because it was present in this fraction; however, we cannot rule out that this may not fully reflect true loading as TDP-43 may impact Lamin C, so the insoluble fraction data should be interpreted with caution.

Results, line 272: Consistent with these biochemical findings, microscopic analysis of the adult brain revealed large cytoplasmic TDP-43 puncta in neurons expressing TDP-43 with elav-GS-GAL4 (Figure 4F-G). Strikingly, Nsun1 downregulation significantly reduced both the number and size of TDP-43 cytoplasmic puncta (Figure 4G). Collectively, these results demonstrate that Nsun1 downregulation in adult neurons promotes nuclear retention of TDP-43, reduces cytoplasmic accumulation, and alleviates TDP-43-induced lifespan deficits.

3. The analysis of m5C is also problematic. Firstly, dot blot analysis is a very crude and unreliable method for this, in particular because the available antibodies are suboptimal for use with m5C in RNA (PMID: 32636310). Furthermore, the absence of any m5C signal in the control spot is worrying. There should be a fairly significant signal derived from rRNA and tRNA. Therefore, to support one of the main claims of the manuscript, that TDP-43 toxicity involves aberrant RNA methylation, the authors need to use additional methods such as mass spectrometry.

We thank the reviewer for these suggestions and the citation. For clarification the citation does not show any evidence that the 5mC antibody used is suboptimal - they in fact showed that their antibody to 5mC worked very well. The antibody that was suboptimal was to an RNA modification called pseudouridine. Furthermore, 5mC is a modification that is widely studied in DNA, and 5mC antibodies have been widely tested and used in detection of 5mC vs unmethylated cytosine. We are using an excellent antibody for the detection of 5mC in RNA in a dot blot assay which is one of the gold standards for measuring changes in RNA modifications {McMillan, 2023 #2791}. Contrary to what the Reviewer anticipates, there is very little RNA with the m5C modification with estimations of only 0.3% of cytidines being methylated in total RNA {Fu, 2015 #3138}. Furthermore, 5mC is considered a stress activated modification – again suggesting that it is normally at low levels.

Updated text:

Discussion, line 435: RNA m⁵C methylation is an adaptive cellular response to stressors such as oxidative stress, heat shock, and nutrient deprivation (Aguilo *et al*, 2016; Blanco *et al*, 2014; Chan *et al*, 2010; Chan *et al*, 2012; Heissenberger *et al*, 2020; Schaefer *et al*, 2010; Schosserer *et al*, 2015). For example, NSUN2 methylates tRNA^{Leu} during oxidative stress to promote translation of TTG-codon-enriched mRNAs, while NSUN7 methylates enhancer

RNAs during starvation to regulate transcription (Aguilo *et al.*, 2016; Blanco *et al.*, 2014; Chan *et al.*, 2010; Chan *et al.*, 2012; Schaefer *et al.*, 2010). Levels of RNA methyltransferases, including NSUN1 and NSUN5, are critical for propagating these stress responses and for lifespan regulation in *Drosophila* and *C.elegans*, (Heissenberger *et al.*, 2020; Schosserer *et al.*, 2015). Consistent with this, we detect little to no m⁵C in total RNA from control *Drosophila* head tissue, whereas TDP-43 expression significantly upregulates m⁵C in an NSUN1-dependent manner

Figure 4D shows a Sanger sequencing result, but the text claims that Illumina sequencing was used - which is correct?

Both are correct, Sanger sequencing of a cloned PCR product was used to show that methylation was detected and Illumina was used to measure the levels of the modified base.

Updated text:

Results, line 297: To measure m⁵C levels of C3402, total RNA isolated from adult *Drosophila* was treated with sodium bisulphite, reverse transcribed, and the region surrounding C3402 was PCR amplified. Sanger sequencing of the subcloned PCR product showed that the conserved NSUN1 cytosine (C3402) is methylated in adult *Drosophila* (Figure 5D). To determine how TDP43 expression influenced methylation at this site, m⁵C levels of C3402 were quantified by illumina sequencing of the PCR product, comparing control, TDP-43, and TDP-43 and si.Nsun1. This showed that the NSUN1-target cytosine in *Drosophila* was constitutively methylated under control conditions (Figure 5E) and was unaltered by TDP-43 expression with or without NSUN1 downregulation (Figure 5E).

Methods, Line 684: For sanger sequencing the PCR product was subcloned into TOPO TA (Thermofisher TOPO™ TA Cloning™ Kit for Sequencing # 450030).

4. The text refers to the different NSUN1 western blot bands as different isoforms, and the schematic in Fig. S4 suggests that they result from alternative splicing. Is there any evidence for this? In other words, could the bands correspond to differently modified proteins rather than spliceforms? Assuming that these bands are indeed from different isoforms, it would have been interesting to see more experiments characterising the specific interaction of TDP-43 with isoform 3.

We agree with the reviewer that there are exciting follow-up experiments to explore the roles of NSUN1 isoform 3, isoform 2, and the region specific to NSUN1 isoform 3. While we cannot yet rule out that these isoforms may undergo post-translational modifications, we have expressed HA-tagged NSUN1 isoform 2 and isoform 3, and both migrate at the expected molecular weights corresponding to the endogenous proteins. Unfortunately, further optimization is required for additional co-immunoprecipitation studies, as the HA tag may affect subcellular localization—although nucleolar localization is still observed, the overall distribution appears broader. These experiments will form the basis of our next studies. We clarify the use of the n-terminal epitope of the NSU1 antibody.

Updated text:

Results, line 322: Human NSUN1 exists as four isoforms, all of which share a central, highly conserved methyltransferase domain flanked by N-terminal and C-terminal intrinsically disordered regions (Figure 6A). The N-terminal region contains both a nuclear localisation sequence and a nucleolar localisation sequence, which also functions as an arginine-rich RNA-binding domain (Gustafson *et al.*, 1998). To establish an effective co-

immunoprecipitation approach, we compared two NSUN1 antibodies with distinct epitopes: one targeting the N-terminal region, which spans sequence differences among isoforms 1, 2, and 3, and a second targeting the central methyltransferase domain (Figure 6A). The N-terminal antibody recognized two major splice variants—isoforms 1 and 2 (89 kDa and 89.3 kDa, respectively)—and the less abundant isoform 3 (93 kDa) (Figure 6B). By contrast, the methyltransferase-domain antibody detected only a single NSUN1 isoform (Figure 6C). These findings were expected, as the N-terminal antibody targets a region that differs among NSUN1 splice variants, enabling recognition of multiple isoforms.

5. Fig. 6E: It is difficult to really see the stress-dependent co-localisation of TDP-43 and NSUN1. In fact, it looks more like they are mutually exclusive in some of the more prominent foci in the ActD-treated cells. It would be better to show the merge without the Hoechst stain to allow better differentiation.

We have added further data to support and strengthen these findings. Our proximity ligation assay (PLA) data demonstrate that a subset of TDP-43 and NSUN1 molecules form a nuclear protein complex with ~60–75% of PLA punctae co-localizing with the nucleolus under normal conditions (Figures 7A–D). Importantly, PLA detects protein–protein interactions at distances <40 nm, indicating close proximity rather than mere co-localization. Upon actinomycin D–induced nucleolar stress, PLA shows increased TDP-43–NSUN1 PLA punctae number and intensity. These findings provide in situ evidence that the TDP-43–NSUN1 interaction occurs in intact cells and is not an artifact of lysis or compartment mixing. Therefore, the PLA data strongly support a biologically relevant interaction between TDP-43 and NSUN1, which can be modulated by nucleolar stress. These data are in Figure 7 and Figure S8, with the original IFC data in Figure S7. We also added detail on TDP-43 in the nucleolus from published studies in the discussion.

Updated text:

Results, line 364: To test this, we employed proximity ligation assay (PLA), which detects protein–protein interactions within <40 nm. Under basal conditions, TDP-43 and NSUN1 formed a nuclear protein complex, but not in the cytoplasm, compared to negative controls (Figure 7A–C, Figures S8A–B). Actinomycin D treatment increased both the number and intensity of TDP-43–NSUN1 PLA foci, without affecting their size (Figure 7B). To determine the subnuclear localization of these PLA foci, cells were counterstained with the nucleolar marker Nucleolar Green. Approximately 75% of TDP-43–NSUN1 PLA puncta co-labelled with the nucleolus under normal conditions, and this remained unaltered by actinomycin D treatment (Figure 7D–E), indicating that TDP-43 and NSUN1 function in both the nucleoplasm and nucleolus under normal and nucleolar stress conditions. By comparison, the co-immunoprecipitation between NSUN1 and TDP-43 remained largely unchanged following actinomycin D treatment, which may reflect either the detection limits of the co-IP assay or that the PLA-detected increase in foci number and intensity represents closer proximity of the proteins rather than an increase in overall binding. Together, these data suggest that TDP-43 and NSUN1 form a nuclear protein complex under basal conditions, and that this interaction is enhanced upon inhibition of RNA transcription and nucleolar breakdown.

6. The conclusions drawn from Figure 7 regarding the changes in NSUN1 isoform composition in ALS

brains are rather weakly supported by the data shown. First, the number of samples is small, and second, the variability introduced by Western blot detection seems to be quite large (e.g. compare #21, 22) in panels C and D. Also, it is unclear to this reviewer why the statistical tests were performed between the different isoforms of one condition and not between the same isoform in co and ALS patients?

We have presented the data for each NSUN1 isoform relative to the GAPDH control (Figure 8 D-G). Additionally, we show the pathological findings for each patient by western blot, including one patient who lacks TDP-43 pathology (Figure S9). This patient was excluded from the main data analysis (Figure 8D–G) but is included in the analysis presented in Figure S10. We have stated clearly what statistical tests were performed in the legends.

Updated text:

Results, line 392: Next, we analysed NSUN1 solubility in postmortem frontal cortex from controls and ALS/FTD patients. Proteins were sequentially extracted in buffers of increasing denaturing strength and were first immunoblotted for TDP-43 (Figure S9). As expected, the control cohort were negative for pathological forms of TDP-43 -the C25 fragment and high molecular weight smear (Neumann *et al.*, 2006) (Figure S9). In our ALS/FTD cohort, four out of five samples were positive for TDP-43 pathology (Figure S9). Patient #16 was negative for frontal cortex TDP-43 pathology by western blot; this patient was diagnosed with frontotemporal lobar degeneration (FTLD)-TDP type C, associated with the semantic variant of primary progressive aphasia (svPPA) (Borghesani *et al.*, 2020). Unlike other FTLD subtypes, atrophy in type C begins in the anterior temporal lobe rather than the frontal cortex, with TDP-43 aggregates appearing as compact neuronal cytoplasmic inclusions in the temporal lobe versus dystrophic inclusions in the frontal cortex (Neumann *et al.*, 2021). It is possible that in patient 13, disease progression from the anterior temporal lobe to the frontal cortex was slow, explaining the absence of TDP-43 pathology in the frontal cortex.

Results, Line 407: Immunoblotting of the low-salt fractions revealed the three NSUN1 isoforms: upper (isoform 3), middle (isoforms 1/2), and lower (isoform 4). Quantification of NSUN1 levels in ALS/FTD with confirmed TDP-43 pathology showed that NSUN1 isoforms 1/2 were significantly reduced (Figure 8D–G), while isoform 3—the NSUN1 isoform that preferentially interacts with TDP-43 (Figure 6)—and isoform 4 were unaffected (Figure 8D–G). Analysis including Patient #16 is shown in Figure S10.

Figure S9 Legend (main manuscript) line 1359: **Figure S9**: TDP-43 immunoblots of human frontal cortex tissue. Postmortem frontal cortex tissue from control and ALS/FTD cohorts immunoblotted for TDP-43 (upper panel) and total protein with PonceauS (lower blot). Tissue was sequentially extracted in protein solubilisation buffer of increasing denaturing strength. The presence of TDP-43 smear (*) and the C25 TDP-43 fragment (***) in the urea fraction indicates TDP-43 pathology is present. TDP-43 at expected molecular weight is indicated (**). Patient 13 in the ALSFTD cohort lacks TDP-43 pathology. Fraction labels are: LS= low salt buffer, TX= Triton X 100 buffer, SARK= sarkosyl buffer, UREA= urea buffer. TDP-43 immunoblot of diseased patients showed high molecular weight smear of ubiquitinated TDP-43 (*), ~45kDa band (**), and pathologic truncated TDP-43-C25 (***)

Figure S10 Legend (main manuscript) line 1376: **Figure S10**: NSUN1 protein levels in frontal cortex with inclusion of the FTD patient that lacked TDP-43 pathology.

A-C: NSUN1 isoforms quantified relative to GAPDH. Graphs are the mean (\pm SD) and a T test. ns: not significant. The data point with the *symbol is the patient that lacks TDP-43 pathology.

Legend (Figure 8), line 1143: E: In ALS/FTD with TDP-43 pathology, NSUN1 isoforms 1/2 are significantly reduced, while isoform 4 and isoform 3 are unaltered. NSUN1 is quantified relative to the GAPDH control. Graph is the mean (\pm SD), and a T test, ns: not significant.

Methods, line 877: Experimental protein levels were quantified relative to the GAPDH loading control.

7. Figure 7C,D and Figure S4D: Why is the size of the bands identified as isoform 4 so different in the blots shown in the two figures?

We have removed isoform 4 from the (S4D, now 5B) as we only had this there as an indicator of where it could be. We do not see the smaller band clearly in U2OS cells.

Minor points

1. The rationale for choosing RNA methyltransferases to study is not clear when most enriched GO terms were RNA modification binding proteins.

We chose the methyltransferases because they target specific RNA species unlike the RNA binding proteins – we reasoned that this would help narrow the mechanistic area. We have amended the figure to highlight the different RNA targets of each methyltransferase and we slightly edited the associated main text.

Results, line 190: Cytosines in RNA are methylated by the s-adenosylmethionine (sam) dependent m⁵C-RNA methyltransferases (Figure 2A), which include the seven NOL1/NOP2/SUN domain (NSUN) enzymes (NSUN1-7) and the DNA methyltransferase DNMT2a. Each m⁵C-RNA methyltransferase methylates specific types of RNAs e.g., NSUN2 methylates mRNA, NSUN2/3/6 methylates tRNA, NSUN1/4/5 methylates rRNA and Mt2 methylates tRNA (Bohnsack *et al*, 2019). Thus, we set out to define which of the m⁵C -RNA methyltransferase(s) functionally interact with TDP-43, hypothesizing that this would reveal the specific RNA methylation pathway and RNA type engaged by TDP-43 through its association with modified RNAs.

2. The editing of the manuscript is rather sloppy. There are many grammatical errors in the main text (e.g. missing punctuation) which do not help clarity.

We have improved the text for clarity and flow

The captions are very minimalistic. Please include more experimental details, especially for Figure 3 (how was the relative level calculated in E,F)

We improved captions and made clear how relative levels in 3EJ now 4D-E

Updated text:

Figures 4 Legend (main file) line 1010: E: TDP-43 protein levels quantified relative to the appropriate control (Lamin C for the nucleus, Tubulin for the cytoplasm and Lamin C for

insoluble protein. Graphs is the mean (\pm SD) of 3 biological repeats and multiple T tests, ns: not significant.

Methods, line 601: All experiments were carried out on three or more biological replicates, blots were quantified with ImageJ (Rueden *et al.*, 2017). Area under the curve for each band was measured using ImageJ for both the experimental protein (TDP-43/b-galactosidase) and the loading control (tubulin/laminC). Each protein band was calculated as percentage of the total intensity across the entire data set on the gel (this was done for each protein), and the experimental band was made relative to the appropriate loading controls. For nuclear and cytoplasmic blots, nuclear TDP-43 was made relative to the lamin C while cytoplasmic TDP-43 was made relative to Tubulin. GraphPad prism 9 software was used to calculate statistical significance.

and Figure 5 (Cells in B are misidentified).

Corrected

The labelling of Western blots is sometimes inadequate, e.g. Fig. 5A, D (several bands are not identified).

Corrected

Mislabelling of Drosophila genotypes in the caption of Fig. 3 (NSUN2 is shown instead of NSUN1). In Fig. S4, panels E and F are mislabelled.

Corrected

3. Graphical summary is not very informative and contains typos (disese, sam).

We removed the graphical summary as this has not been requested.

Figure 5 (Cells in B are misidentified).

Corrected

Fig. 5A, D (several bands are not identified)

Corrected

September 18, 2025

RE: Life Science Alliance Manuscript #LSA-2025-03297-TR

Dr. Leeanne McGurk
University of Dundee
School of Life Sciences
MSI/WTB complex
Dundee DD1 5EH
United Kingdom

Dear Dr. McGurk,

Thank you for submitting your revised manuscript entitled "Aberrant NSUN1 Activity Connects m5C RNA Modification to TDP-43 Neurotoxicity in ALS/FTD". Your manuscript was re-evaluated by the original reviewers whose comments are appended below.

As you will note, the reviewers have acknowledged that the revised manuscript is significantly improved. Reviewer 3 has reiterated a comment on the m5C antibody and contradiction in the lack of signal under control conditions. We recommend that you rephrase the conclusion for the detection of 5mC from fly head total RNA, by including a point to reflect the reviewer's original comment on the use of other approaches to validate this.

We would be happy to publish your paper in Life Science Alliance pending final revisions necessary to meet our formatting guidelines.

-In the methods section, please provide the following:

-a citation or composition for standard molasses formulation for fly food used.

-Bloomington Drosophila Stock Center ID and/or original citation for the line GMR-GAL4 (YH3) line.

-A scale bar for Fig. S7A and S7E.

-size information for scale bar in Fig. 7A, Fig S7 panels.

-details for microscopes, objectives (N.A. and magnification) and excitation/emission filters used for fluorescence microscopy for both fly tissues and for mammalian cells.

-For human subjects, we encourage you to follow our guidelines, "For experiments involving human subjects, the authors must identify the committee approving the experiments, provide a statement that informed consent was obtained from all subjects and that the experiments conformed to the principles set out in the WMA Declaration of Helsinki and the Department of Health and Human Services Belmont Report."

-For Western blots in the methods, we encourage you to specify if GAPDH serves as a loading control (the same gel) or a reference protein. Likewise for TDP-43 normalisation to LaminC or Tubulin.

-Please conduct a thorough spell check on the manuscript.

-We encourage you to use appropriate nomenclature for fly gene/protein names.

-We thank you for providing the raw western blot images. We urge you to confirm that all the blots are identified correctly, for example: Figure 6 and Figure 7, the blot names do not match with the images shown in the panels.

-Please consult our manuscript preparation guidelines <https://www.life-science-alliance.org/manuscript-prep> and make sure your manuscript sections are in the correct order.

-Please add your main, supplementary figure, and table legends to the main manuscript text after the references section.

-Please add an Author Contributions section to your main manuscript text.

-Please include a "Data Availability" section after the Materials & Methods section. Please consult our guidelines at <https://www.life-science-alliance.org/manuscript-prep#format>

-Please add a "Conflict of Interest" statement to your main manuscript text.

-Please correct the call-out for figure S2K.

-We encourage you to revise the figure legends for figures 7 and 8 such that the figure panels are introduced in alphabetical order.

-Please add callouts for Figures 4H-I and S8C-D to your main manuscript text.

-Please be sure that the authorship listing and order is correct

LSA now encourages authors to provide a 30-60 second video where the study is briefly explained. We will use these videos on social media to promote the published paper and the presenting author (for examples, see <https://docs.google.com/document/d/1-UWcfbE4pGcDdcgzcmiuJI2XMBJnxKYeqRvLLrLSo8s/edit?usp=sharing>). Corresponding

or first-authors are welcome to submit the video. Please submit only one video per manuscript. The video can be emailed to contact@life-science-alliance.org

A. FINAL FILES:

B. MANUSCRIPT ORGANIZATION AND FORMATTING:

Thank you for your attention to these final processing requirements. Please revise and format the manuscript and upload materials as soon as you are able.

Sincerely,

Sarita Hebbar, PhD
Scientific Editor
Life Science Alliance
<http://www.lsjournal.org>

Reviewer #1 (Comments to the Authors (Required)):

Parra Torres and colleagues investigate native protein networks interacting with TDP-43 in SH-SY5Y cells, finding strong enrichment of RNA modification pathways. Proteins binding RNA 5-methylcytosine (m5C) and 6-methyladenine (m6A) were among the most enriched. In *Drosophila*, elevated TDP-43 increased m5C RNA modifications in an NSUN1-dependent manner,

and this nucleolar methyltransferase proved essential for TDP-43-driven neurodegeneration. In human cells, TDP-43 selectively interacted with the long NSUN1 isoform, which remains abundant in post-mortem neurodegenerative cortex while others decline. These results suggest that TDP-43 modulates RNA methylation and that excessive m5C may drive TDP-43-linked neurodegeneration, highlighting a previously unrecognized role of TDP-43 in regulating RNA methylation. I believe that the revised manuscript is now suitable for publication in Life Science Alliance.

Reviewer #2 (Comments to the Authors (Required)):

The authors have addressed the majority of concerns. Although there are remaining, unanswered questions, the manuscript is significantly improved.

Reviewer #3 (Comments to the Authors (Required)):

I appreciate the effort the authors have made to address my previous comments and the improvements in clarity and readability in the revised manuscript. The inclusion of new data has certainly enhanced the overall merit of the study.

However, I would like to point out that the response to my earlier comment regarding the use of m5C antibodies in the dot blot analysis remains somewhat unconvincing. The study I cited in my initial review demonstrated that commercially available antibodies against m5C from Cell Signaling and Diagenode did not perform well in various assays. It is true that the self-made m5C antibody in that study showed good performance, yet it is important to note that this antibody was raised against an RNA nucleoside, unlike the commercial antibodies (including the one used in this manuscript), which were raised against a DNA nucleoside.

Additionally, the explanation provided for the absence of an m5C signal in total RNA from fly heads is not entirely clear. Given that rRNA and tRNA, which together constitute more than 95% of total RNA, are both constitutively methylated at several cytosine positions (as the authors themselves demonstrated for C3402), it is inaccurate to imply that there is no or very low methylation under normal conditions. The m5C signal (originating from rRNA and tRNAs) should be detectable under normal conditions. I would therefore recommend revising or rephrasing the sentence: "Consistent with this, we detect little to no m5C in total RNA from control *Drosophila* head..." to better reflect these considerations.

That said, I commend the authors for the significant improvements made to the manuscript and for the additional data, which strengthen the study overall.

As you will note, the reviewers have acknowledged that the revised manuscript is significantly improved. Reviewer 3 has reiterated a comment on the m5C antibody and contradiction in the lack of signal under control conditions. We recommend that you rephrase the conclusion for the detection of 5mC from fly head total RNA, by including a point to reflect the reviewer's original comment on the use of other approaches to validate this.

Line 454:

Because antibody-based detection has limited sensitivity, future work using mass spectrometry or 454 long-read direct RNA sequencing will better define the true rRNA methylation status with and without TDP-43 and it would have the potential to uncover additional sites across the transcriptome

We would be happy to publish your paper in Life Science Alliance pending final revisions necessary to meet our formatting guidelines.

-In the methods section, please provide the following:

-a citation or composition for standard molasses formulation for fly food used.

Line 535:

or in Molasses Formulation (27.03 g/L inactive yeast (Scientific Laboratory Supplies, FLY1062), 72.13 g/L yellow cornmeal (Scientific Laboratory Supplies, FLY1076), 90ml/l Molasses (Scientific Laboratory Supplies, FLY1296), 0.81g/16.27ml Tegosept (Scientific Laboratory Supplies, FLY1046, and 5.63 ml/l Propionic Acid (Sigma-Aldrich 402907), unless otherwise stated

-Bloomington Drosophila Stock Center ID and/or original citation for the line GMR-GAL4 (YH3) line.

Line 46 Table S4 – second entry.

-A scale bar for Fig. S7A and S7E.

Added

-size information for scale bar in Fig. 7A, Fig S7 panels.

7A in legend, S7 in all panels.

-details for microscopes, objectives (N.A. and magnification) and excitation/emission filters used for fluorescence microscopy for both fly tissues and for mammalian cells.

Fly eyes: Line 545: Leica Z16 Apo A microscope, DFC420 camera and 2.0x planapochromatic objective, 0.034 – 0.224nA, as described

Line 550 Fly section line 542: Sections were imaged on a Zeiss Axio550imager A1 with a 20x objective with a 0.5 numerical aperture.

Confocal details Drosophila Line 620: Leica SP8 confocal microscope using identical settings across genotypes and used an HCPL APO 20X objective.

Confocal details for cells Line 784 Leica SP8 confocal microscope using an HCPL APO 20X objective with a numerical aperture of 0.75

High content imaging Line 802: Cells were imaged on an

Operetta CLS1601 (Revvity): imaging was performed using a 20x objective (0.4NA) with 2x binning. The acquisition settings used for each fluorophore were: Hoechst 10% power, 20ms exposure with 355-385 Excitation filter (Ex) and 430-500 Emission filter (Em); AlexaFluor488 50% power, 50ms exposure with 460-490 Ex and 500-550 Em; and AlexaFluor594 50% power, 300ms exposure with 560-575 Ex and 570-650 Em.

-For human subjects, we encourage you to follow our guidelines, "For experiments involving human subjects, the authors must identify the committee approving the experiments, provide a statement that informed consent was obtained from all subjects and that the experiments conformed to the principles set out in the WMA Declaration of Helsinki and the Department of Health and Human Services Belmont Report."

Line 822

Human samples and data

To ensure that experiments involving human subjects conformed to the principles of the WMA Declaration of Helsinki and the Department of Health and Human Services Belmont Report, ethical approval for the human tissue work was reviewed and granted by the access committee at the London Neurodegenerative Disease Brain Bank at King's College London,. Informed consent was given from all donors and all the experiments conformed to the principles set out in the WMA Declaration of Helsinki and the Department of Health and Human Services Belmont Report.

-For Western blots in the methods, we encourage you to specify if GAPDH serves as a loading control (the same gel) or a reference protein. Likewise for TDP-43 normalisation to LaminC or Tubulin.

Flies: Line 608:

For all blots, the control protein (Tubulin/LaminC) was608 detected on the same blot as TDP-43 or β -galactosidase.

Human: Line 889

Experimental protein levels were quantified relative to the GAPDH loading control, which was889 imaged on the same membrane as NSUN1.

-Please conduct a thorough spell check on the manuscript.

done

-We encourage you to use appropriate nomenclature for fly gene/protein names.

done

-We thank you for providing the raw western blot images. We urge you to confirm that all the blots are identified correctly, for example: Figure 6 and Figure 7, the blot names do not match with the images shown in the panels.

Done and uploaded all panels in main figure

-Please consult our manuscript preparation guidelines <https://www.life-science-alliance.org/manuscript-prep> and make sure your manuscript sections are in the correct order.

done

-Please add your main, supplementary figure, and table legends to the main manuscript text after the references section.

done

-Please add an Author Contributions section to your main manuscript text.

done

-Please include a "Data Availability" section after the Materials & Methods section. Please consult our guidelines at <https://www.life-science-alliance.org/manuscript-prep#format>

Done for RNA sequencing result. I am still waiting on the proteomic upload.

-Please add a "Conflict of Interest" statement to your main manuscript text.

done

done

-Please correct the call-out for figure S2K.

done

-We encourage you to revise the figure legends for figures 7 and 8 such that the figure panels are introduced in alphabetical order.

done

-Please add callouts for Figures 4H-I and S8C-D to your main manuscript text.

done

Reviewer #3 (Comments to the Authors (Required)):

I appreciate the effort the authors have made to address my previous comments and the improvements in clarity and readability in the revised manuscript. The inclusion of new data has certainly enhanced the overall merit of the study.

However, I would like to point out that the response to my earlier comment regarding the use of m5C antibodies in the dot blot analysis remains somewhat unconvincing. The study I cited in my initial review demonstrated that commercially available antibodies against m5C from Cell Signaling and Diagenode did not perform well in various assays. It is true that the self-made m5C antibody in that study showed good performance, yet it is important to note that this antibody was raised against an RNA nucleoside, unlike the commercial antibodies (including the

one used in this manuscript), which were raised against a DNA nucleoside. Additionally, the explanation provided for the absence of an m5C signal in total RNA from fly heads is not entirely clear. Given that rRNA and tRNA, which together constitute more than 95% of total RNA, are both constitutively methylated at several cytosine positions (as the authors themselves demonstrated for C3402), it is inaccurate to imply that there is no or very low methylation under normal conditions. The m5C signal (originating from rRNA and tRNAs) should be detectable under normal conditions. I would therefore recommend revising or rephrasing the sentence: "Consistent with this, we detect little to no m5C in total RNA from control *Drosophila* head..." to better reflect these considerations. That said, I commend the authors for the significant improvements made to the manuscript and for the additional data, which strengthen the study overall.

Amended comment to:

Line 454:

Because antibody-based detection has limited sensitivity, future work using mass spectrometry or 454 long-read direct RNA sequencing will better define the true rRNA methylation status with and without TDP-43 and it would have the potential to uncover additional sites across the transcriptome

October 16, 2025

RE: Life Science Alliance Manuscript #LSA-2025-03297-TRR

Dr. Leeanne McGurk
University of Dundee
School of Life Sciences
MSI/WTB complex
Dundee DD1 5EH
United Kingdom

Dear Dr. McGurk,

Thank you for submitting your Research Article entitled "Aberrant NSUN1 Activity Connects m5C RNA Modification to TDP-43 Neurotoxicity in ALS/FTD". It is a pleasure to let you know that your manuscript is now accepted for publication in Life Science Alliance. Congratulations on this interesting work.

Your manuscript will now progress through copyediting and proofing. It is journal policy that authors provide original data upon request. Please note that there are two minor pending changes that must be completed at the proofing stage. Firstly, please provide the DOI for the deposited mass spectrometric data in the 'Data availability' statement as soon as it is available (that you have also indicated in your letter). Secondly, please include callouts for Figure S8C-D in your main manuscript text.

DISTRIBUTION OF MATERIALS:

Again, congratulations on a very nice paper. I hope you found the review process to be constructive and are pleased with how the manuscript was handled editorially. We look forward to future exciting submissions from your lab.

Sincerely,

Sarita Hebbar, PhD
Scientific Editor
Life Science Alliance
<http://www.lsajournal.org>